# S-phase PARylation of microprotein RSMC enhances the function of Sororin in sister chromatid cohesion

Meiqian Jiang[1,2,6], Jiaxin Zhang[1,6], Jiankun He[1,2], Yu Miao[3], Linhui Wang[1], Haitao Zhong[1], Yingying Gong[1], Zhen Li [ID][2], Li-Lin Du [ID][4], Xingzhi Xu [ID][1], Chunlai Chen[3], Alibek Ydyrys[5], Yisui Xia[1], Qinhong Cao [ID][2], Huiqiang Lou [ID][1✉] & Wenya Hou [ID][1✉]

## Abstract

Sororin is essential for establishing sister chromatid cohesion concurrently with DNA replication in metazoans. Although acetylation of the cohesin subunit SMC3 by ESCO1/2 is necessary for Sororin recruitment, it is by itself not sufficient. Here, we demonstrate that DNA replication-coupled Poly(ADP-Ribose) Polymerase (PARP) activity is an additional prerequisite in human cells. During normal S-phase, PARP1 PARylates a microprotein encoded by the alternative ORF *C11ORF98*, which we designate RSMC (28S rRNA/ ribosome and Sororin micro-cofactor). This PARylation strengthens the interaction of RSMC with Sororin, enhancing both chromatin recruitment and anti-Wapl activity of Sororin in concert with SMC3 acetylation. Notably, overexpression of RSMC is able to rescue cohesion defects induced by the PARP inhibitor olaparib. These findings highlight understudied microproteins as critical regulators of fundamental cellular processes, such as sister chromatid cohesion.

**Keywords** Chromatin; Cohesin; Dark Proteome; DNA Replication; Microprotein
**Subject Categories** Cell Cycle; DNA Replication, Recombination & Repair; Post-translational Modifications & Proteolysis

## Introduction

The cohesin complex tethers sister chromatids from their synthesis during DNA replication until their separation at anaphase, a process termed sister chromatid cohesion (Nasmyth et al, 2000; Peters and Nishiyama, 2012). This ring-shaped complex comprises four core subunits: SMC1 (structural maintenance of chromosomes protein 1), SMC3, a kleisin subunit RAD21 and SA1/2 (STAG1/2) (Nasmyth and Haering, 2009; Skibbens, 2009; Zheng and Yu, 2015). Beyond cohesion (Morales and Losada, 2018), cohesin shapes high-order chromatin structures such as loops (Datta et al, 2020; Davidson and Peters, 2021; Hassler et al, 2018; van Ruiten and Rowland, 2021), three-dimensional genome organization and centriole engagement (Nishiyama, 2019; Schockel et al, 2011; Yatskevich et al, 2019), and regulates DNA replication (Sherwood et al, 2010), DNA damage response/repair (Hou et al, 2022), gene expression (Perea-Resa et al, 2021), and chromosome segregation (Uhlmann, 2016). To fulfill these functions, cohesin and its association with chromatin are dynamically controlled by a plethora of regulatory factors. Mutations in the cohesin subunits or its regulators underlie cohesinopathies, including developmental disorders and cancer (Banerji et al, 2017; Mintzas and Heuser, 2019; Piche et al, 2019; Tonkin et al, 2004; Waldman, 2020).

Cohesin is loaded onto chromatin by the NIPBL-MAU2 protein complex throughout the cell cycle (Ciosk et al, 2000; Watrin et al, 2006). In the G1 phase, cohesin encircles chromatin loosely and is dynamically modulated by the cohesin-releasing factor Wapl and scaffold protein PDS5 (Gandhi et al, 2006; Goto et al, 2017; Kanke et al, 2016; Kueng et al, 2006; Sutani et al, 2009). During S phase, cohesin becomes stable (cohesive) when SMC3 acetylation (SMC3ac) and Sororin recruitment to block Wapl's activity, a process strictly coupled to DNA replication in metazoans (Ladurner et al, 2016; Marston, 2017; Nishiyama et al, 2010; Schmitz et al, 2007). SMC3 is acetylated by ESCO1 and ESCO2 (Chan et al, 2012; Rolef Ben-Shahar et al, 2008; Rowland et al, 2009; Unal et al, 2008; Zhang et al, 2008; Alomer et al, 2017; Minamino et al, 2015). The acetylase ESCO2 is recruited by several replication fork components (e.g., PCNA (Skibbens et al, 1999); CRL4-MMS22L (Sun et al, 2019; Zhang et al, 2023a; Zhang et al, 2017) and MCM (Ivanov et al, 2018; Yoshimura et al, 2021)). Post SMC3 acetylation, Sororin is recruited and bound with PDS5 to antagonize Wapl further, which is essential to establish cohesion during S phase and maintain it till mitosis (Nishiyama et al, 2010; Schmitz et al, 2007; Yamada et al, 2017; Zhou et al, 2021). Although SMC3 acetylation is essential, it alone cannot fully recruit Sororin (Nishiyama et al, 2010), suggesting missing regulators.

To identify these regulators, we conducted immunoprecipitation-coupled mass spectrometry (IP-MS) and identified an alternative

[1]Guangdong Key Laboratory for Genome Stability & Disease Prevention, School of Basic Medical Sciences, Shenzhen University Medical School, Shenzhen University, 518055 Shenzhen, China. [2]State Key Laboratory of Plant Environmental Resilience, College of Biological Sciences, China Agricultural University, 100193 Beijing, China. [3]State Key Laboratory of Membrane Biology, Beijing Frontier Research Center for Biological Structure, School of Life Sciences, Tsinghua University, 100084 Beijing, China. [4]National Institute of Biological Sciences, Beijing, China. [5]Biomedical Research Centre, Al-Farabi Kazakh National University, Al-Farabi Avenue 71, Almaty 050038, Kazakhstan. [6]These authors contributed equally: Meiqian Jiang, Jiaxin Zhang. ✉E-mail: lou@szu.edu.cn; wenya.hou@szu.edu.cn

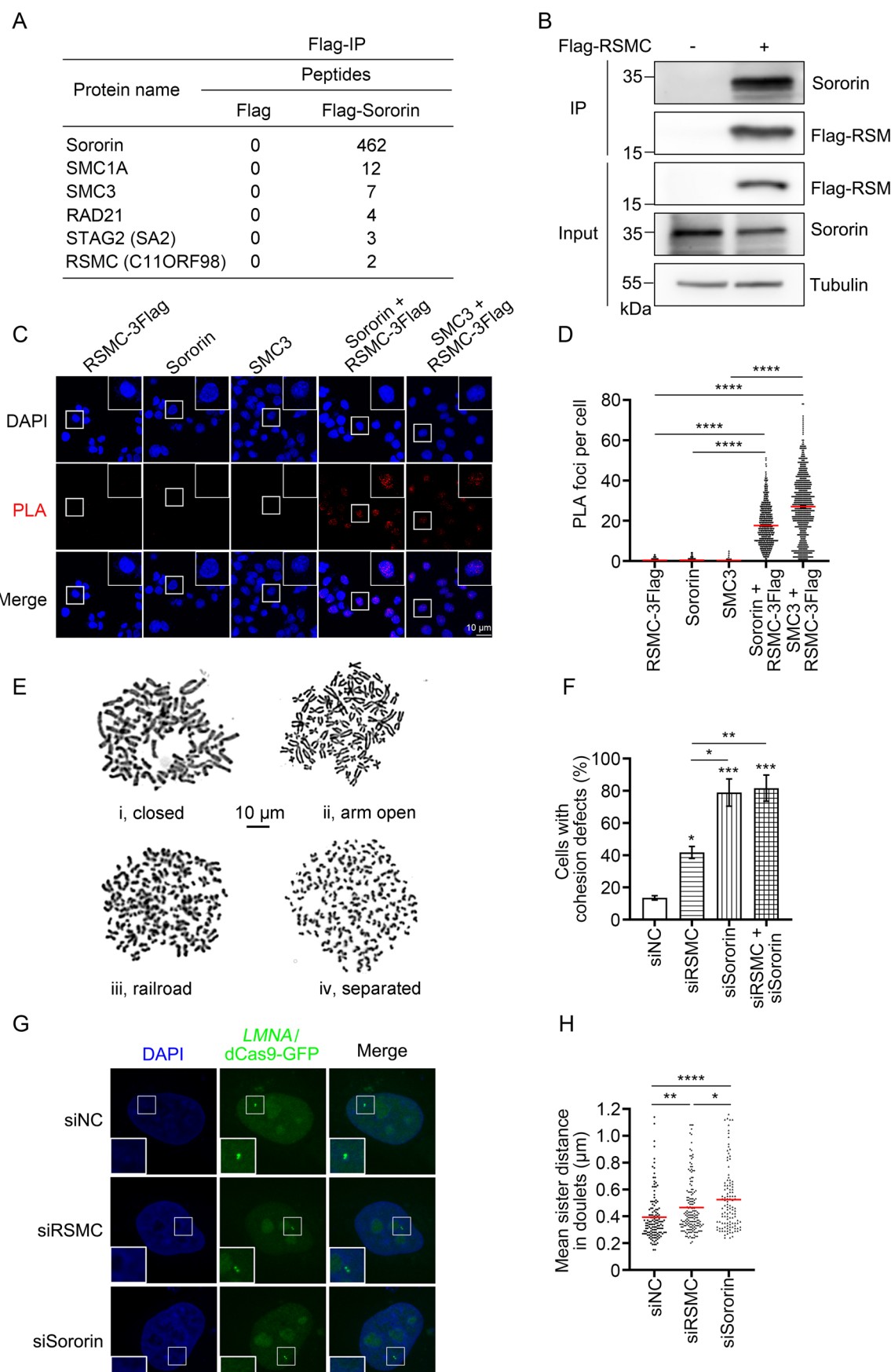

**Figure 1. Microprotein RSMC functions in sister chromatid cohesion as a Sororin partner.**

(A) Identification of RSMC as a Sororin-interacting protein by IP-MS. Flag-Sororin was overexpressed in HEK293F cells for 48 h and immunoprecipitated using anti-Flag M2 agarose. Co-immunoprecipitated proteins were identified by mass spectrometry, and the average of peptide counts from two independent replicates are shown. (B) Endogenous Sororin co-precipitates with RSMC. Flag-RSMC-overexpressing HEK293T cells were subjected to Flag IP, and endogenous Sororin in the precipitates was observed via immunoblotting with anti-Sororin antibody. In all, 3% of samples were loaded as input. (C, D) In situ interaction between RSMC and Sororin is shown by PLA. (C) RSMC-3Flag cells were subjected to PLA with anti-Flag, anti-Sororin, and anti-SMC3 antibodies. Scale bar: 10 μm. (D) Quantification of PLA foci per cell (over 700 cells from three independent assays). Data represent mean ± SEM. Significance was determined by one-way ANOVA with Tukey's post hoc test. RSMC-3Flag vs Sororin +RSMC-3Flag ****$P = 1.1712e{-8}$, RSMC-3Flag vs SMC3 + RSMC-3Flag ****$P = 1.1712e{-8}$, Sororin vs Sororin+RSMC-3Flag ****$P = 1.1712e{-8}$, SMC3 vs SMC3 + RSMC-3Flag ****$P = 1.1712e{-8}$. (E, F) RSMC KD compromises sister chromatid cohesion, although to a lesser extent than Sororin deficiency. Chromosome spreads of HEK293T cells transfected with siRNA targeting control (siNC), RSMC (siRSMC), or Sororin (siSororin). Chromosome spreads were Giemsa-stained. Scale bar: 10 μm. (F) Cohesion defects were scored from >600 cells across three biologically independent replicates. Data represent mean ± SEM. Significance was assessed by one-way ANOVA with Tukey's post hoc test. siNC vs siRSMC *$P = 0.0483$, siNC vs siSororin ***$P = 0.0003$, siNC vs siRSMC+siSororin ***$P = 0.0002$, siRSMC vs siSororin *$P = 0.0118$, siRSMC vs siRSMC+siSororin **$P = 0.0079$, siSororin vs siRSMC+siSororin $P = 0.9886$. (G, H) RSMC depletion increases sister chromatid separation. Sister chromatid distance at the *LMNA* locus (chr1q22) was measured in HeLa cells stably expressing dCas9-GFP$_{14\times}$. Cells were synchronized at G1/S (double-thymidine block), released into S phase for 4 h, and fixed with 4% paraformaldehyde (PFA). More than 100 cells were measured from at least two independent biological assays. Scale bar = 10 μm. Mean values of siNC = 0.39; siRSMC = 0.47; siSororin = 0.52. Data were mean values ± SEM, Significance was determined by one-way ANOVA with Tukey's post hoc test. siNC vs siRSMC **$P = 0.0073$, siNC vs siSororin ****$P = 2.1507e{-6}$, siRSMC vs siSororin *$P = 0.0456$. Source data are available online for this figure.

ORF (alt-ORF) *C11ORF98*-encoded microprotein (herein named RSMC, 28S rRNA/ribosome and Sororin micro-cofactor) as a Sororin partner in human cells. RSMC depletion caused cohesion defects albeit milder than Sororin knockdown (KD), and its direct association with Sororin was indispensable for cohesion. Intriguingly, this interaction was enhanced by S-phase PARP activity via RSMC PARylation, complementing SMC3 acetylation to ensure Sororin's timely chromatin recruitment and anti-Wapl function. Overexpression of RSMC rescued cohesion defects from PARP1/2 inhibition, underscoring its pivotal role. Through chromatin fractionation and in vitro competitive MST analysis, we showed that RSMC stimulates the anti-Wapl activity of Sororin. These results suggest that, together with DNA replication-coupled acetylation of SMC3, another replication-coupled activity, PARP1, exploits a microprotein RSMC to promote the recruitment and anti-Wapl activity of Sororin to establish cohesion.

## Results

### Microprotein RSMC: a key Sororin partner

Previous studies have established that Sororin is an essential factor downstream of SMC3 acetylation for both the establishment and maintenance of sister chromatid cohesion in vertebrates (Nishiyama et al, 2010; Schmitz et al, 2007). However, SMC3ac catalyzed by ESCO2 is necessary but insufficient to recruit Sororin (Nishiyama et al, 2010). To identify the hidden regulators of Sororin, we set out to look for its interaction partners in human cells through immunoprecipitation-coupled mass spectrometry (IP-MS). Flag-tagged Sororin was overexpressed in HEK293T cells and was precipitated by anti-Flag M2 beads. All core cohesin complex components (SMC1, SMC3, RAD21, STAG2) were detected in the precipitates, validating the efficacy of IP-MS (Figs. 1A and EV1A). Besides these known interactors, an alt-ORF (123 a.a., ~14 kDa), *C11ORF98*, was identified as well (Figs. 1A and EV1B). During the preparation of this manuscript, *C11ORF98* was reported to encode a nuclear factor for pre-60S ribosome assembly as a distal functional ortholog of Alb1 (Arx1 little brother) in *Saccharomyces cerevisiae* (Fig. EV1C). Nevertheless, human C11ORF98 binds 28S rRNA but not PA2G4 (Arx1 in yeast)

(Zhang et al, 2023b). Putting together its function as a Sororin cofactor described in this study, we hereafter call it RSMC (28S rRNA/ribosome and Sororin micro-cofactor). Its physical interaction with Sororin was also supported by reciprocal co-immunoprecipitation (CoIP) (Fig. 1B). To further corroborate the RSMC–Sororin association in situ, we tagged RSMC at the C-terminus by 3Flag at its genomic loci to overcome the lack of specific antibodies. Immunostaining revealed that RSMC was primarily distributed in the nucleus as Sororin (Fig. EV1D). Furthermore, we performed in situ proximity ligation assays (PLA) in this cell line and found that RSMC displayed strong signals with either Sororin or the structural subunit SMC3 (Fig. 1C,D). In addition, SMC3 co-precipitates with RSMC as well (Fig. EV1E). These data indicate that RSMC interacts with Sororin and the cohesin complex in vivo.

Next, we assessed whether RSMC is required for sister chromatid cohesion as Sororin. Human cells were treated with small interference RNAs (siRNAs) targeted to control, RSMC, or Sororin. The knockdown (KD) efficiency of RSMC was about 60% as shown by quantitative reverse transcription PCR (qRT-PCR) and immunoblotting (Fig. EV1F,G). Since *RSMC/C11ORF98* is an alternative ORF of the *LBHD1* gene (Fig. EV1B), we confirmed that these siRNAs do not alter the *LBHD1* mRNA level (Fig. EV1F) and the cohesion defects in RSMC KD cells could be complemented by reintroducing RSMC (as described later in Fig. 2C). Having established the correct KD of RSMC, we started analyzing the karyotype by chromosome spreads and Giemsa staining. Cells containing precocious separated sister chromatids (railroad or unpaired) were counted as cohesion defects (Fig. 1E). In agreement with previous studies (Nishiyama et al, 2010; Rankin et al, 2005), Sororin KD caused ~80% of cells with cohesion defects (Fig. 1F). When RSMC was depleted, ~40% of cells displayed cohesion defects. To confirm the role of RSMC in cohesion, we performed CRISPR-Cas9-mediated knockout (KO) and only obtained heterozygous RSMC KO cells with a reduced RSMC mRNA level similar to that in RSMC KD (Fig. EV1H,I). In line with it, RSMC$^{+/-}$ cells exhibited a similar cohesion defect as RSMC KD ones. Combination of siRSMC with RSMC$^{+/-}$ caused an additive cohesion loss (~67%), very close to Sororin-depleted cells (Fig. EV1J). Furthermore, the combinational depletion of RSMC and Sororin did not cause an additional cohesion defect compared with Sororin KD

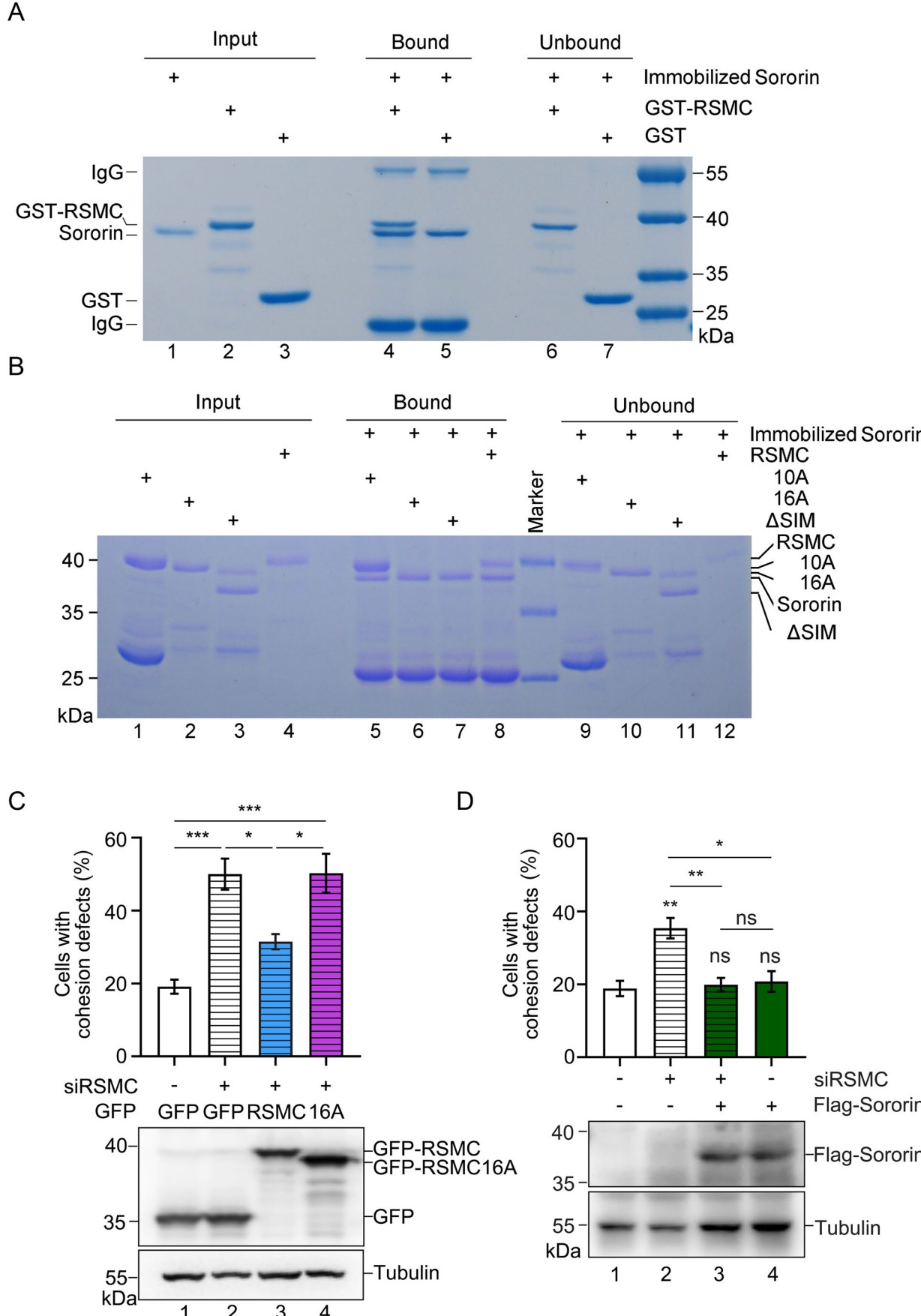

**Figure 2. RSMC participates in cohesion through direct Sororin interaction.**

(A) Direct RSMC–Sororin association confirmed by in vitro pull-down assay. Recombinant GST-RSMC or GST (negative control) was incubated with 6×His-Flag-Sororin immobilized on anti-Flag M2 agarose. Bound and unbound proteins were analyzed by SDS-PAGE and Coomassie brilliant blue (CBB) staining. 30% of the samples were loaded as input. (B) Isolation of two Sororin-binding mutants, RSMC-16A and RSMCΔSIM. In vitro pulldown was performed as described above. In total, 30% of samples were loaded as input. RSMC-10A = R30A, R31A, R33A, R35A, S65A, E87A, E89A, S98A, D113A, E115A. (C) RSMC-16A is defective in cohesion. HEK293T cells were transfected with RSMC siRNA followed by GFP, GFP-RSMC or GFP-RSMC-16A. Chromosome spreads of over four biologically independent replicates revealed RSMC, but not RSMC-16A, could rescue cohesion defects in RSMC-depleted cells. Immunoblots of GFP, GFP-RSMC, and GFP-RSMC-16A are shown below each column to confirm expression. Data represent mean ± SEM. Significance was analyzed by one-way ANOVA with Tukey's post hoc test. si(-)+GFP vs siRSMC+GFP ***$P$ = 0.0006, siRSMC+GFP vs siRSMC+RSMC *$P$ = 0.0369, siRSMC+RSMC vs siRSMC+RSMC-16A *$P$ = 0.0152, si(-)+GFP vs siRSMC+ RSMC-16A ***$P$ = 0.0003. (D) Sororin overexpression rescues cohesion defects caused by RSMC depletion. HEK293T cells stably expressing Flag or Flag-Sororin were transfected with RSMC siRNA. Cohesion defects were scored from three biologically independent replicates. Immunoblots of Flag and Tubulin validated Sororin expression. Data represent mean ± SEM. Significance determined by one-way ANOVA with Tukey's post hoc test. si(−) vs siRSMC **$P$ = 0.0059, si(−) vs siRSMC+Flag-Sororin $P$ = 0.9892, si(−) vs si(−)+Flag-Sororin $P$ = 0.9408, siRSMC vs siRSMC+Flag-Sororin **$P$ = 0.0086, siRSMC vs si(−)+Flag-Sororin *$P$ = 0.0120, siRSMC+Flag-Sororin vs si(−)+Flag-Sororin $P$ = 0.9938. Source data are available online for this figure.

(Fig. 1F), implicating that they may function in an epistatic manner. To examine whether RSMC is required for cohesion establishment during interphase, we next measured the distance between sister chromatids labeled by CRISPR/dCas9-GFP$_{14×}$ at one of the two *LMNA* loci. The distance between two *LMNA* alleles located on two homologous chromosomes was reported to be over 4.2 μm (Xu et al, 2020). We identified a pair of GFP foci with a distance less than 1.2 μm to ensure that the GFP-labeled *LMNA* locus has already replicated, in other words, the cells are in the S or G2 phases. Consistent with previous reports (Stanyte et al, 2018; Xu et al, 2020), the average distance between sister chromatids in the control cells was ~0.39 μm (Fig. 1G,H). However, the average inter-chromatid distance increased to ~0.47 and ~0.52 μm by depleting RSMC and Sororin, respectively (Fig. 1G,H). Collectively, these data indicate that microprotein RSMC is involved in sister chromatid cohesion as its partner Sororin in human cells.

## RSMC–Sororin interaction drives cohesion

Does RSMC promote sister chromatid cohesion through interaction with Sororin? To address this question, we first mapped their interaction domain by in vitro pulldown assays. Recombinant GST-RSMC and 6×His-Flag-Sororin were purified and incubated together with anti-Flag beads. About the same amount of GST-RSMC was co-purified with 6×His-Flag-Sororin, whereas GST alone did not bind at all (Fig. 2A, lanes 4 and 5). This indicates a direct association between RSMC and Sororin. Next, we took advantage of AlphaFold2 and GRAMM protein docking tools to predict the structure and interface of RSMC–Sororin. RSMC might contain two α-helixes hinged by a protruding small helix in the middle (Fig. EV2A). The protruded small helix and loose C-terminal tail seemed to form two separate interfaces (a.a. 30–65 and a.a. 87–115) of the RSMC–Sororin complex (Fig. EV2B,C). Through serial truncations and substitutions of the conserved polar side chain residues (including R, S, E, etc.), we showed that RSMC–Sororin interaction was not abolished until combinational loss of both interfaces and obtained two interaction-defective mutants called RSMCΔSIM (named hereafter S̲ororin-I̲nteracting M̲otif) (Figs. 2B, lanes 7, 8 and EV2D, lane 5) and RSMC-16A (Figs. 2B, lane 6 and EV2E, lane 2), respectively. Functionally, unlike wild-type RSMC, both RSMC-16A and RSMCΔSIM failed to rescue the cohesion defects in RSMC-depleted cells (Figs. 2C and Fig. EV2F). Thus, the role of RSMC in

sister chromatid cohesion relies on its direct interaction with Sororin. Furthermore, Sororin overexpression completely rescued the cohesion defects caused by inadequate RSMC (Fig. 2D), implying that RSMC may function as a Sororin facilitator.

## RSMC enhances Sororin recruitment during S phase

To establish sister chromatid cohesion, Sororin is recruited to chromatin-loaded cohesin after SMC3 acetylation by ESCO2. Next, we tested whether RSMC is also involved in this step by five different approaches. First, the protein levels of Sororin increased during S phase and decreased in G2/M (Rankin et al, 2005). Coincidentally, as shown by immunoblotting of the synchronized cell samples released from a double-thymidine (DT) block for the indicated time, endogenous RSMC protein phenocopied the cell-cycle-regulated pattern of Sororin (Fig. EV3A).

Second, Sororin–cohesin interaction was analyzed by co-immunoprecipitation (CoIP) through the subunit SMC3. Since their interaction is cell-cycle-regulated, we synchronized cells in early S phase by a DT block and released them into S phase for 5 h. Endogenous SMC3 was precipitated by anti-SMC3 antibody and analyzed by immunoblots. In control cells, the amount of Sororin co-precipitated with SMC3 showed a ~80% increase during S phase (Fig. 3A,B, compare lanes 3 and 1), in agreement with previous studies (Nishiyama et al, 2010). RSMC$^{+/−}$ cells had a similar basal level of SMC3-bound Sororin in G1 phase. However, no more SMC3-bound Sororin could be detected in S phase (Fig. 3A,B, compare lanes 2 and 4). Meanwhile, RSMC depletion did not change S phase progression significantly, suggesting that the reduced Sororin–cohesin interaction is unlikely due to the cell cycle delay (Fig. EV3B). These results indicate that RSMC is required for Sororin–cohesin association during S phase.

Third, since chromatin association of Sororin reflects its co-complex with cohesin as described before (Lafont et al, 2010; Nishiyama et al, 2010; Rankin et al, 2005), we then performed chromatin immunoprecipitation (ChIP) to test whether RSMC binds cohesin-associated regions (CARs) (Ladurner et al, 2016). Flag-tagged RSMC or Sororin was precipitated via anti-Flag M2 agarose. Afterward, the specific associated DNA was quantified by real-time PCR. RSMC was enriched at the same CARs (Chr2:193027129-193027415, and Chr8:134214868-134215746) as Sororin (Figs. 3C and EV3C,D), arguing that RSMC can bind CARs as well as Sororin. To evaluate whether RSMC affects the

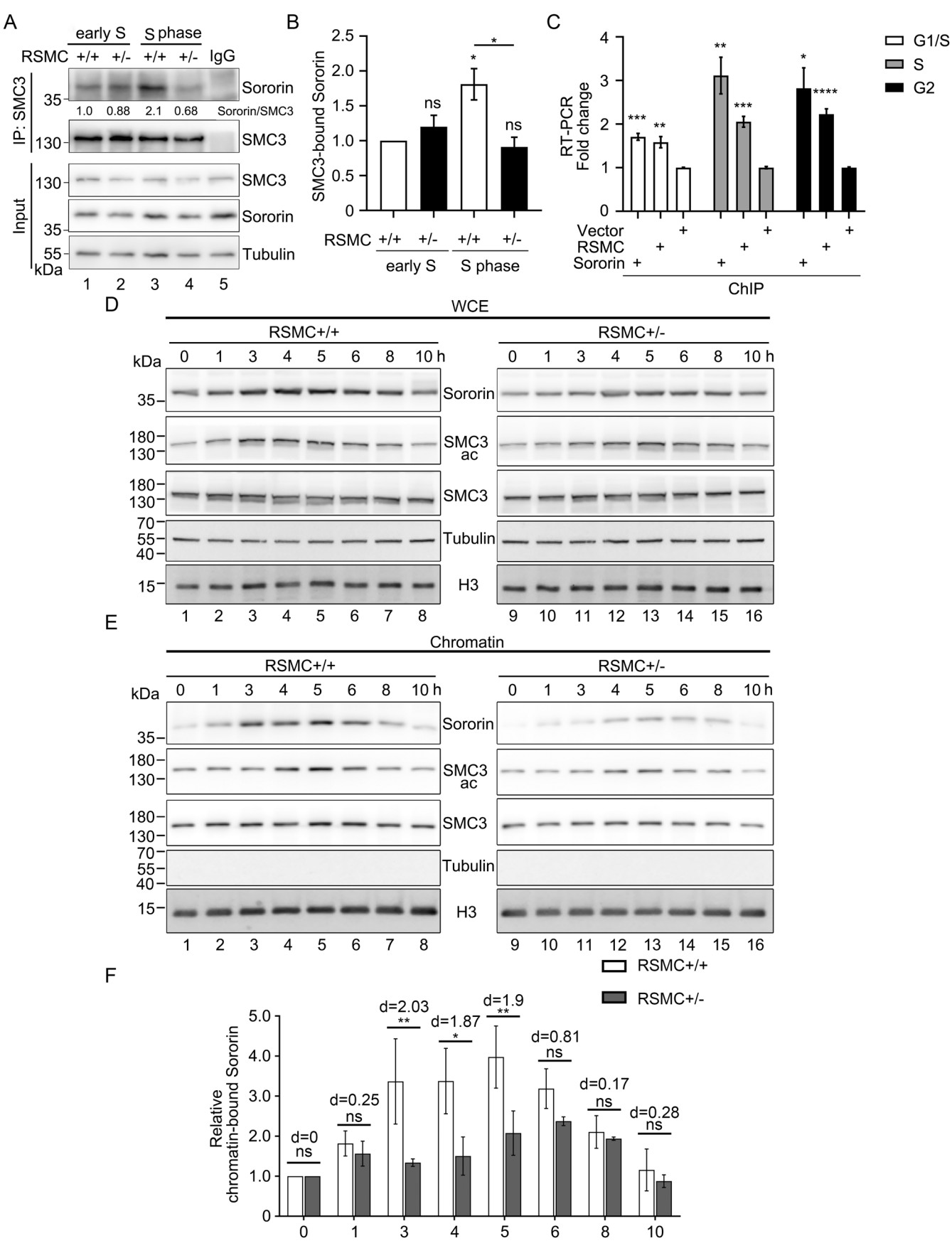

◀ **Figure 3.  RSMC promotes the recruitment of Sororin to cohesin on chromatin during the S phase.**

(A, B) RSMC is important for Sororin binding to cohesin. HEK293T RSMC$^{+/+}$ and RSMC$^{+/-}$ cells were synchronized in early (0 h post double-thymidine release) or mid-S (5 h) phases. Whole-cell lysates were immunoprecipitated with anti-SMC3 antibody or control IgG, and precipitates were immunoblotted with indicated antibodies. In total, 3% of samples were loaded as input. (B) The Sororin/SMC3 ratio in immunoprecipitates was quantified from three independent experiments using Quantity One software. Data represent mean ± SEM. The statistical significance was calculated via one-way ANOVA with Tukey's post hoc test. RSMC$^{+/+}$ (early S) vs RSMC$^{+/-}$ (early S) $P = 0.7974$, RSMC$^{+/+}$ (early S) vs RSMC$^{+/+}$ (S) *$P = 0.0257$, RSMC$^{+/+}$ (early S) vs RSMC$^{+/-}$ (S) $P = 0.9761$, RSMC$^{+/+}$ (S) vs RSMC$^{+/-}$ (S) *$P = 0.0149$. (C) RSMC, like Sororin, binds to specific cohesin-associated regions (CARs). HeLa cells transfected with 9Flag, 9Flag-RSMC, or Flag-Sororin were harvested and analyzed by ChIP using anti-Flag M2 agarose. qPCR was performed with primers targeting known Sororin/cohesin-associated CARs (Human Chr2:193027129-193027415, Chr8:134214868-134215746, see Fig. EV3C,D). Data represent mean ± SEM. The statistical significance was calculated via two-way ANOVA with Tukey's multiple comparisons test. Data from three independent experiments were shown. $P$ value from left to right: Sororin vs. Vector (G1/S) ***$P = 0.0002$, RSMC vs. Vector (G1/S) **$P = 0.0071$, Sororin vs. Vector (S) **$P = 0.0028$, RSMC vs. Vector (S) ***$P = 0.0002$, Sororin vs. Vector (G2) *$P = 0.0124$, RSMC vs. Vector (G2) ****$P = 1.3183e-5$. (D–F) RSMC is necessary for efficient Sororin recruitment to chromatin. Immunoblotting of whole-cell extracts (WCE, D) and chromatin-associated fractions (chromatin, E) from RSMC$^{+/+}$ or RSMC$^{+/-}$ cells. Histone H3 served as a loading control. (F) The relative Sororin/H3 ratio of the chromatin fractions and WCE (Fig. EV3E) was calculated from three independent assays. Data are presented as means ± SEM. The statistical significance was calculated via two-way ANOVA with Sidak's multiple comparisons test. The "d" represents the difference value of chromatin-bound Sororin in RSMC$^{+/+}$ and RSMC$^{+/-}$. $P$ value from left to right: $P > 0.9999$, $P = 0.9993$, **$P = 0.0055$, *$P = 0.0105$, **$P = 0.0094$, $P = 0.5850$, $P > 0.9999$, $P = 0.9986$. Source data are available online for this figure.

chromatin association during the S phase progression. RSMC$^{+/-}$ or control cells were synchronized in early S phase (0 h) by a DT block before releasing into fresh media to allow cells to proceed through S (1–6 h) and G2/M phases (8–10 h) (Fig. EV3B). Cells were collected at the indicated time points, whole-cell extracts (WCE) and chromatin-bound protein fractions (Chromatin) were prepared and subjected to immunoblotting. Again, the Sororin protein levels were cell cycle-regulated. Intriguingly, in RSMC$^{+/-}$, the total Sororin levels reduced by about 30% compared with wild-type during G1 and S phases (0–6 h) (Figs. 3D and EV3E). The reduction became much less apparent during G2/M, presumably due to its degradation like other cohesion factors (e.g., Sororin) at this stage. The protein levels of Sororin could be restored by treatment with MG132, a proteasome inhibitor (Fig. EV3F). Meanwhile, the mRNA levels of Sororin were not significantly changed after RSMC depletion (Fig. EV3G). These results suggest that RSMC might stabilize Sororin. To separately assess the contribution of RSMC to the chromatin recruitment of Sororin during S phase, we normalized the Sororin levels on chromatin at 0 h in RSMC$^{+/-}$ and control cells. In control, chromatin-bound Sororin increased up to fourfold from early S to late S (0–5 h) and then decreased in the G2/M phase (6–10 h) (Fig. 3E,F). However, in RSMC$^{+/-}$, the accumulation of Sororin on chromatin was not so evident, especially before 5 h. The successful chromatin fractionation was indicated by the distribution of histone H3 and Tubulin (Fig. 3D,E). These data are consistent with the RSMC-cohesin interaction shown in Fig. 3A, suggesting that RSMC contributes to the efficient and timely recruitment of Sororin to cohesin on chromatin during the S phase.

Fourth, given that SMC3ac also promotes Sororin recruitment, we next asked whether RSMC and SMC3ac act in the same pathway. ESCO1 and ESCO2 acetyltransferases were KD via siRNA (Fig. 4A). Meanwhile, RSMC$^{+/-}$ hardly changed the protein level of ESCO2, the major SMC3 acetyltransferase during replication-coupled cohesion establishment (Fig. 4B). The cells with the indicated genotypes were synchronized in S phase (5 h) and collected for chromatin fractionation. Total and chromatin-bound Sororin in RSMC$^{+/-}$ or control cells with or without ESCO1/2 siRNA treatment were analyzed. Consistent with previous reports (Nishiyama et al, 2010), ESCO1/2 KD reduced the SMC3ac levels as well as the chromatin-bound Sororin levels (Fig. 4C, compare lanes 7 to 5, and Fig. 4D). On the other hand, RSMC$^{+/-}$ caused a~40%

decline in chromatin-bound Sororin (Fig. 4D), while only a~10% decline in whole-cell extracts (Fig. EV3H). Meanwhile, SMC3ac levels were not significantly changed by RSMC depletion (Fig. 4C, compare lanes 6–5). These results support that RSMC contributes to Sororin recruitment without significantly affecting SMC3ac. In line with this, an additive loss of chromatin-bound Sororin was observed when RSMC$^{+/-}$ and ESCO1/2 KD were combined (Fig. 4C, lanes 5–8, and 4D).

Fifth, to further confirm these results, we analyzed the carnoy-fixed chromosome spreads from which soluble proteins had been removed by hypotonic pretreatment (Nagasaka et al, 2016; Nishiyama et al, 2010). By immunofluorescence analysis, we observed prominent chromatin-bound Sororin in wild-type cells (Fig. 4E). However, the average Sororin intensity decreased by ~34% in either RSMC$^{+/-}$ or ESCO1/2 KD (Fig. 4E,F). Again, a combination of RSMC$^{+/-}$ and ESCO1/2 KD caused a further decline. Together, these data suggest that RSMC enhances the recruitment of Sororin to cohesin on chromatin during S phase in cooperation with ESCO1/2-dependent SMC3 acetylation.

## PARP1 Regulates RSMC–Sororin via PARylation

A previous proximity labeling proteomic study reported that RSMC might be a putative interactor of PARP1 (Chu et al, 2017). On the other hand, many other studies have established intimate relationships among PARP1, cohesin, and DNA replication (Mondal et al, 2019; O'Neil et al, 2013). Besides DNA breaks, PARP1 also recognizes nicks or flaps in the normal replication intermediates like unligated Okazaki fragments and becomes activated during the S phase (Azarm and Smith, 2020; Hanzlikova et al, 2018; Kumamoto et al, 2021). Thus, we reasoned that RSMC might be regulated by the S-phase PARP1 activity. To test this possibility, we first validated RSMC-PARP1 interaction by CoIP (Fig. 5A). To examine whether this interaction is cell-cycle-regulated and/or PARP-activity-dependent, we next conducted in situ PLA assays at physiologic protein levels using RSMC-3Flag cells synchronized by a DT block before release into S phase. In early S phase, only very weak PLA signals were visible (Fig. 5B, panel 1, C). After cells proceeded into S phase for 5 h, the signals became much stronger (Fig. 5B, panel 3, C). Notably, regardless of the signal intensity, both PARP1-RSMC signals in early or mid-S phase vanished in the presence of a PARP1/2 inhibitor (PARPi), olaparib, demonstrating

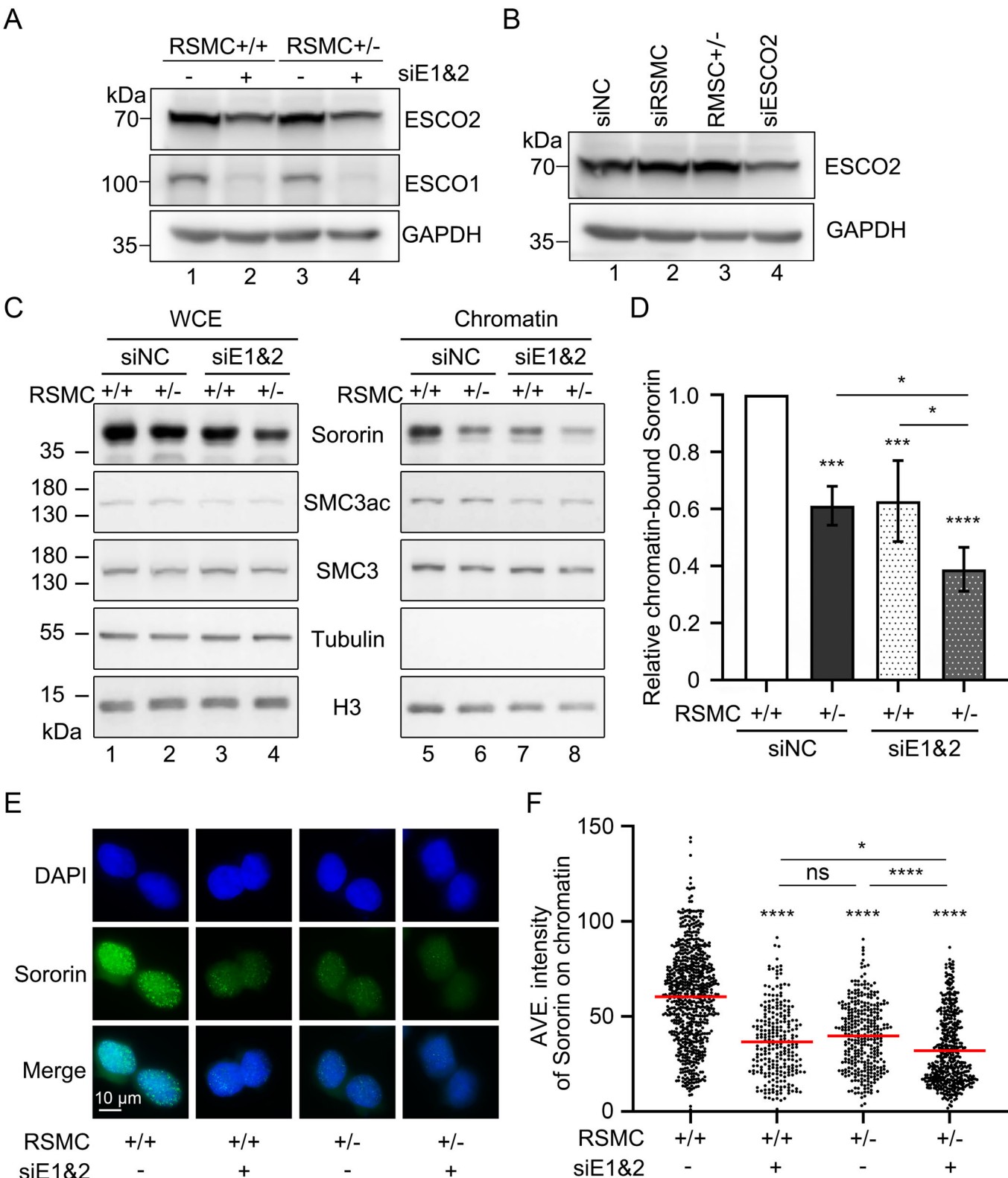

◀ **Figure 4.   RSMC and SMC3ac cooperatively recruit Sororin to chromatin.**

(A) The efficiency of ESCO1 and ESCO2 knockdown. RSMC$^{+/+}$ or RSMC$^{+/-}$ cells transfected with ESCO1 and ESCO2 siRNAs were analyzed by immunoblotting. (B) The protein level of ESCO2 is unaffected by RSMC depletion. RSMC$^{+/-}$ or RSMC$^{+/+}$ cells transfected with siRNAs were subjected to immunoblotting with the anti-ESCO2 antibody. GAPDH as loading control. (C, D) Chromatin fractions were prepared from RSMC$^{+/+}$ or RSMC$^{+/-}$ cells treated with ESCO1/2 RNAi. Cells were harvested after 5 h release from the double-thymidine blocks. (C) Immunoblotting analyses of whole-cell extracts (left) and chromatin (right) using antibodies against indicated proteins. (D) Quantification of chromatin-bound Sororin from four independent assays by Quantity One software. The relative Sororin/H3 ratio was calculated and normalized to siNC RSMC$^{+/+}$. Data represent mean ± SEM. The statistical significance was calculated via one-way ANOVA with Tukey's post hoc test. RSMC$^{+/+}$ (siNC) vs RSMC$^{+/-}$ (siNC) ***$P = 0.0002$, RSMC$^{+/+}$ (siNC) vs RSMC$^{+/+}$ (siE1&2) ***$P = 0.0003$, RSMC$^{+/+}$ (siNC) vs RSMC$^{+/-}$ (siE1&2) ****$P = 2.1839e-6$, RSMC$^{+/-}$ (siNC) vs RSMC$^{+/-}$ (siE1&2) *$p = 0.0168$, RSMC$^{+/+}$ (siE1&2) vs RSMC$^{+/-}$ (siE1&2) *$P = 0.0106$. (E, F) Immunofluorescence analysis of chromatin-bound Sororin was prepared from RSMC$^{+/+}$ or RSMC$^{+/-}$ cells treated with ESCO1/2 RNAi. Cells were harvested after 5 h release from the double-thymidine blocks, and chromatin was pre-extracted before staining. (E) Sororin was stained using its specific antibodies. Nuclei were counterstained using DAPI. Scale bar: 10 μm. (F) The average intensity of Sororin's signals on chromatin was quantified over 200 cells from four independent biological assays. Data represent mean ± SEM. The statistical significance was calculated via one-way ANOVA with Tukey's post hoc test. RSMC$^{+/+}$ (si−) vs RSMC$^{+/+}$ (si+) ****$P = 5.3070e-12$, RSMC$^{+/+}$ (si−) vs RSMC$^{+/-}$ (si−) ****$P = 5.3070e-12$, RSMC$^{+/+}$ (si−) vs RSMC$^{+/-}$ (si+) ****$P = 5.3070e-12$, RSMC$^{+/+}$ (si+) vs RSMC$^{+/-}$ (si−) $P = 0.2739$, RSMC$^{+/-}$ (si−) vs RSMC$^{+/-}$ (si+) ****$P = 2.3473e-6$, RSMC$^{+/+}$ (si+) vs RSMC$^{+/-}$ (si+) *$P = 0.0257$. Source data are available online for this figure.

the specificity of our PLA assays. These results indicate that the interaction between PARP1 and RSMC is PARP activity-dependent and thereby cell-cycle-regulated.

PARP1 usually catalyzes the ADP-ribose unit from NAD$^+$ to specific amino acid residues (e.g., E, D, K, R, and S) on its substrates to form a long and branched negatively charged poly(ADP-ribose) (PAR) chain (Kamaletdinova et al, 2019; Wei and Yu, 2016). We next performed in vitro PARylation reactions using purified recombinant RSMC and PARP1. Besides self-PARylation, PARP1 catalyzed RSMC PARylation (Fig. 5D, lane 3; Fig. EV4A, lane 3; Fig. EV4B, lane 1), which was completely inhibited by PARPi olaparib (Fig. EV4A, lane 4). These suggest that RSMC might be a substrate of PARP1. To confirm this, we then detected RSMC PARylation in vivo. GFP-RSMC was trapped in cell lysates. The PARylated RSMC band was very weak, but it could be significantly reinforced by peroxide treatment (Fig. 5E, compare lanes 4–3). In silico analyses using ADPredict (Lo Monte et al, 2018) and DeepSADPr (Sha et al, 2022) tools predicted that potential 16 PARylated sites of RSMC are enriched coincidentally within the two dispersed Sororin-interacting motifs (SIM) (Fig. EV2C). Indeed, neither RSMCΔSIM nor RSMC-16A mutant proteins could be PARylated anymore (Fig. 5D, lane 2; Fig. 5E, lane 6; Fig. EV4B, lane 2). These in vitro and in vivo results indicate that RSMC's SIM can be targeted by PARP1. The coincidence of the PARylation sites with SIM prompted us to test whether PARylation affects RSMC–Sororin interaction. After preincubation with PARP1 in the presence or absence of NAD$^+$, GST-RSMC was then mixed with purified recombinant Sororin. More Sororin was co-purified with RSMC pretreated by PARP1 in the presence of NAD$^+$ (Fig. 5F,G, compare lanes 3–4, and Fig. EV4C). On the other hand, like RSMCΔSIM, RSMC-16A lost Sororin-binding and cohesion functions as shown in Fig. 2. These results indicate that RSMC is PARylated at its interface with Sororin, which in turn enhances RSMC–Sororin association.

How does S-phase PARylation of RSMC enhance RSMC–Sororin's binding? We predicted that there are three putative PAR-binding motifs in Sororin (Fig. EV4D), which may enable its PAR-binding. Indeed, this was confirmed by in vitro PAR chain binding assays (Fig. EV4D, lane 3). When all of these conserved residues were mutated to alanine, Sororin-12A largely lost both PAR-binding (Fig. EV4D, lane 2) and RSMC association (Fig. EV4E–G). Consistent with these biochemical characteristics, Sororin-12A showed a largely diminished capability to rescue

cohesion defects resulting from depletion of Sororin or RSMC (Figs. 5H and EV4H). Next, we repeated PLA assays of RSMC–Sororin to address whether their interaction is regulated by PARP1 in vivo. The RSMC–Sororin PLA foci were prominently increased from early S phase (0 h) to late S phase (5 h) (Fig. 5I,J). Importantly, these signals were completely abolished by PARPi treatment. The cell-cycle-regulated PLA signals of RSMC–Sororin phenocopied those of PARP1-RSMC (Fig. 5I,B). Collectively, these data suggest that PARP1 may PARylate RSMC and enhance its binding to Sororin during S phase.

Further supporting this notion, like RSMC depletion, olaparib treatment also led to a ~40% decrease of chromatin-bound Sororin in S phase without a significant SMC3ac change (Fig. 6A, lane 10 and Fig. 6B). A similar decline was observed after a very short period (1 h) treatment of emetine (EME) (Fig. 6A, lane 11), which had been shown to selectively inhibit the formation of Okazaki fragments (i.e., the source of S phase PARP activity) (Burhans et al, 1991). Combination PARPi with either emetine (Fig. 6A, lane 12) or RSMC$^{+/-}$ (Fig. 6A, lane 9) did not further reduce chromatin-bound Sororin compared with any single perturbation. All these biochemical data were further corroborated by immunofluorescence analysis of chromatin-bound Sororin as shown in Fig. 6C,D, thus allowing us to conclude that DNA replication, PARP1 and RSMC function in an epistatic manner in recruiting Sororin to chromatin. Surprisingly, PARP1 KD caused no apparent cohesion defects (Fig. EV5A), as reported previously (Kukolj et al, 2017). We reasoned that this might be due to insufficient RNAi efficiency, PARP1/2 redundancy and/or low PARP activity requirement for RSMC. Therefore, we further knocked down PARP2 in the primary enzyme PARP1 KO (PARP1$^{-/-}$) background. Interestingly, such combinational depletion of PARP1/2 caused a significant cohesion defect close to olaparib treatment and RSMC depletion (Figs. 1F, 6E,F, and EV5B). More importantly, olaparib-induced cohesion defect could be effectively rescued by overexpression of either Sororin or its partner RSMC (Figs. 6E,F and EV5C). However, the PAR-binding-deficient mutant Sororin-12A and non-PARylatable mutant RSMC-16A failed to do so. These data strongly support that PARP activity is necessary for sister chromatid cohesion, and RSMC represents the predominant target of PARP1/2 in this process. Mechanistically, PARP inhibition reduced chromatin-bound Sororin. This reduction was additive with ESCO1/2 KD (Fig. 6G,H), suggesting that PARP-RSMC and ESCO1/2-SMC3ac pathways independently recruit Sororin. Supporting this, additive

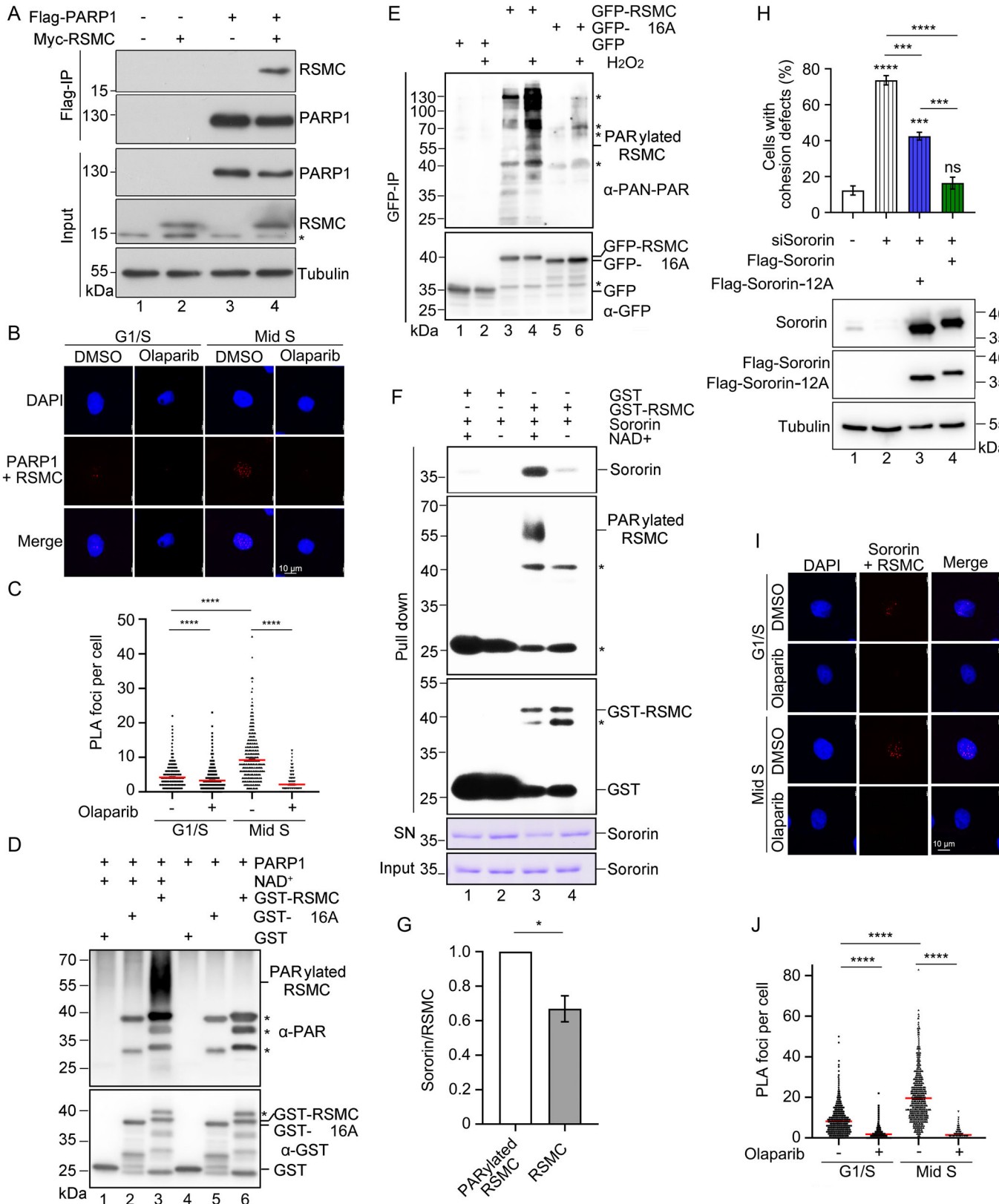

◀ **Figure 5. The S phase PARP1 targets RSMC and enhances RSMC–Sororin interaction.**

(A) The interaction between Flag-PARP1 and Myc-RSMC is shown by CoIP. HEK293T cells were transfected with indicated plasmids for 48 h. Cell lysates were immunoprecipitated with anti-Flag M2 agarose followed by immunoblotting with antibodies against Flag or Myc. An asterisk indicates a nonspecific signal. (B, C) In situ interaction between RSMC and PARP1 detected by PLA. RSMC-3Flag cells were synchronized in early (0 h post double-thymidine release) or mid-S (5 h) phases. 10 μM olaparib was added 1 h before release. Fixed cells were subjected to PLA using anti-Flag and anti-PARP1 antibodies. Scale bar: 10 μm. (C) Quantification of PLA foci number. Data from three independent assays ($n > 800$ cells) are shown as mean ± SEM. Statistical significance was calculated using one-way ANOVA with Tukey's post hoc test. In G1/S, Olaparib(−) vs Olaparib(+) ****$P = 2.7041e-5$; In Mid S, Olaparib(−) vs Olaparib(+) ****$P = 7.1195e-9$; G1/S Olaparib(−) vs Mid S Olaparib(−) ****$P = 7.1195e-9$. (D) PARP1 PARylates RSMC but not RSMC-16A in vitro. In vitro PARylation assays were performed using purified GST-RSMC WT or 16 A in the presence or absence of recombinant $NAD^+$. PARylated RSMC was detected using an anti-PAR antibody. The specificity of RSMC's PARylation signal was validated as shown in Fig. EV5A. (E) RSMC is PARylated in vivo. HEK293T cells expressing GFP-tagged vector, RSMC or RSMC-16A were lysed and subjected to GFP IP, followed by immunoblotting with anti-PAN PAR and anti-GFP antibodies. (F, G) PARylation enhances RSMC's interaction with Sororin. In vitro, PARylation of RSMC was conducted in the absence or presence of $NAD^+$. After removing PARP1 and $NAD^+$, Sororin was incubated with immobilized RSMC. Input (20%) and unbound supernatants (20%) were subjected to CBB staining. The bead-bound proteins were checked by immunoblotting (3% to detect GST-bound RSMC, and 30% each to check RSMC PARylation and its associated Sororin). SN means supernatants. (G) Quantification of the pulldown fraction of Sororin by PARylated RMSC (lane 3) or non-PARylated RSMC (lane 4). Mean ± SEM from three independent assays were shown. Statistical significance was calculated using Student's $t$ test. *$P = 0.0119$. (H) Sororin-12A is defective in cohesion. Cells expressing Flag-tagged vector, Sororin WT, or Sororin-12A were transfected with Sororin siRNA. Cohesion defects were quantified. Immunoblots of Flag, Sororin, and Tubulin validated Sororin expression. Data from three independent assays are shown as mean ± SEM. Significance determined by one-way ANOVA with Tukey's post hoc test. siSororin(−) vs siSororin(+) ****$P = 1.0334e-6$, siSororin(−) vs siSororin(+)+12 A ***$P = 0.0002$, siSororin(−) vs siSororin(+)+Sororin $P = 0.7104$, siSororin(+) vs siSororin(+)+12 A ***$P = 0.0002$, siSororin(−) vs siSororin(+)+Sororin ****$P = 1.7747e-6$, siSororin(+)+12 A vs siSororin(+)+Sororin ***$P = 0.0006$. (I, J) The interaction between RSMC and Sororin increased during the S phase. (I) In situ interaction between RSMC and Sororin was detected by PLA. RSMC-3Flag cells were synchronized in early S (0 h post double-thymidine release) or mid-S (5 h) phases and treated with olaparib. Fixed cells were then subjected to PLA assay with anti-Flag and anti-Sororin antibodies. Scale bar: 10 μm. (J) PLA foci numbers were quantified from three independent assays ($n > 900$ cells). Mean ± SEM is shown. Statistical significance was calculated using one-way ANOVA with Tukey's post hoc test. In G1/S, Olaparib(−) vs Olaparib(+) ****$P = 4.2095e-8$; In Mid S, Olaparib(−) vs Olaparib(+) ****$P = 4.2095e-8$; G1/S Olaparib(−) vs Mid S Olaparib(−) ****$P = 4.2095e-8$. Source data are available online for this figure.

cohesion defects were observed when ESCO1/2 KD was combined with PARP inhibition (Fig. EV5C). In addition, overexpression of the PAR-binding-deficient mutant Sororin-12A rescued cohesion defects in ESCO1/2 KD cells but failed to do so in PARPi-treated cells (Fig. EV5C). Together with the results described in Fig. 4, these data argue that replication-coupled PARP1/2 activities enhance the Sororin cofactor function of RSMC and thereby cohesion establishment in parallel with the ESCO1/2-mediated SMC3 acetylation pathway.

## RSMC boosts Sororin's anti-Wapl activity

Sororin participates in cohesion establishment by antagonizing the cohesin-releasing activity of PDS5-Wapl (Nishiyama et al, 2010). The direct association of RSMC with Sororin encourages us to test whether it acts through anti-Wapl by two different approaches. First, the mitotic cohesion defects in RSMC-depleted cells could be fully rescued by combinational depletion of Wapl (Fig. 7A,B), suggesting that akin to Sororin, RSMC likely contributes to the cohesion through an anti-Wapl mechanism.

Second, to evaluate the Wapl-PDS5 binding kinetics more quantitatively, we adopted microscale thermophoresis (MST) to measure their dissociation constant (Kd) in vitro. Recombinant PDS5B protein was purified and labeled by Cy5. Twofold serial dilutions of Wapl (with an initial concentration of 116 μM) were mixed with PDS5B-Cy5 and incubated for 10 min at room temperature before MST analysis. The binding curve was shown in Fig. 7C (black curve), and the Kd of Wapl-PDS5B (black) was calculated to be about 4.7 μM (Fig. 7D), close to previous studies (~6 μM) measured by isothermal titration calorimetry (ITC) (Liang et al, 2018). Meanwhile, Kd of Sororin-PDS5B (cyan) was about 0.54 μM (Fig. 7E). The addition of Sororin increased Kd of Wapl-PDS5B to 25.3 μM (Fig. 7C,D, purple), in agreement with its well-established anti-Wapl activity (Nishiyama et al, 2010). RSMC alone posed a subtle effect on Wapl-PDS5B binding (orange), indicating

that RSMC has no anti-Wapl activity by itself. However, when RSMC was added together with Sororin in a 1:1 molar ratio, Kd of Wapl-PDS5B was further increased up to 50.1 μM (Fig. 7C,D, red), indicating that RSMC has a potent anti-Wapl stimulatory activity. This stimulatory activity depended on RSMC–Sororin association because it completely disappeared in the interaction-defective mutant (RSMC-16A and RSMCΔSIM in Fig. 7D, green and blue, and Sororin-12A in Fig. EV5D,E, yellow). Collectively, these in vitro and in vivo data suggest that in addition to Sororin recruitment, RSMC also stimulates Sororin's anti-Wapl activity. Based on these findings, we propose that two fork-associated reactions, PARP1-mediated RSMC PARylation reported here and ESCO1/2-catalyzed SMC3 acetylation, cooperatively recruit Sororin to establish sister chromatid cohesion in human cells (Fig. 7F).

# Discussion

We uncover a dual replication-coupled mechanism—PARP1-mediated RSMC PARylation and ESCO1/2-driven SMC3 acetylation—that ensures Sororin's role in cohesion. Sororin, as an essential establishment factor, is recruited to the cohesin complex in a strict DNA replication-dependent manner (Lafont et al, 2010; Nishiyama et al, 2010). Previous studies have established that Sororin recruitment partially relies on SMC3 acetylation (Lafont et al, 2010; Nishiyama et al, 2010), which is coupled to DNA replication through linking cohesin acetyltransferases ESCO1/2 with several replisome components including MCM, PCNA and CRL4[MMS22L] (Ivanov et al, 2018; Skibbens et al, 1999; Sun et al, 2019; Yoshimura et al, 2021; Zhang et al, 2008). During normal DNA replication, nicks and flaps arising during Okazaki processing can trigger both the endogenous PARP1/2 activity and Eco1 acetyltransferase activity (Azarm and Smith, 2020; Hanzlikova and Caldecott, 2019; Hanzlikova et al, 2018; Kumamoto et al, 2021) (Minamino et al, 2023).

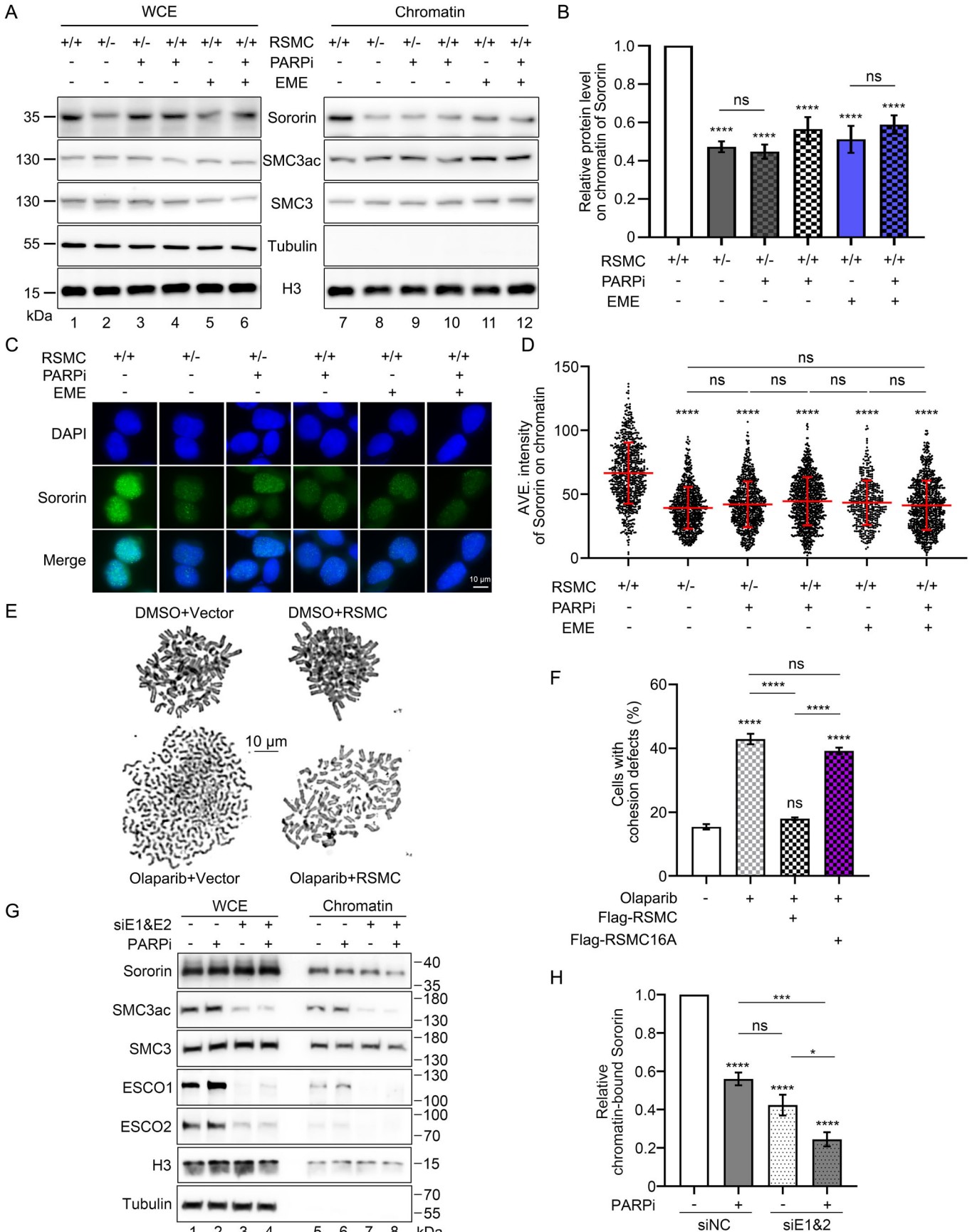

**Figure 6. Replication-coupled PARP and RSMC cooperate with ESCO1/2 to recruit Sororin.**

(A–D) Inhibition of PARP1/2 (olaparib) or DNA replication (emetine, EME) impedes Sororin recruitment on chromatin. (A) Chromatin fractionation was performed as described in Fig. 4C. (B) Quantitative data are presented as means ± SEM from five independent repeats. $P$ value of RSMC$^{+/+}$ vs others (from left to right), ****$P = 3.2014$e-6, ****$P = 1.4982$e-6, ****$P = 2.3967$e-5, ****$P = 4.2127$e-6, ****$P = 5.3302$e-5; RSMC$^{+/-}$ vs RSMC$^{+/-}$(+PARPi) $P = 0.9993$, RSMC$^{+/+}$(+ EME) vs RSMC$^{+/+}$(+ EME&PARPi) $P = 0.8598$. (C) Immunofluorescence analysis was performed as described in Fig. 4E. (D) Quantitative data were from three independent assays. One-way ANOVA with Tukey's post hoc test was used for the comparisons. $P$ value from left to right, from bottom to top: ****$P = 3.5116$e-8, ****$P = 3.5116$e-8, ****$P = 3.5116$e-8, ****$P = 3.5116$e-8, ****$P = 3.5116$e-8, $P = 0.0578$, $P > 0.9999$, $P = 0.9465$, $P = 0.4497$, $P = 0.3004$. (E, F) RSMC-16A fails to rescue cohesion defects caused by PARP inhibition. Cells transfected with Flag-RSMC-16A were treated with or without 10 µM olaparib for 48 h. Mitotic cells were collected for chromosome spreads. Sister chromatid cohesion defects were quantified in >300 mitotic cells across three independent biological replicates. Data represent mean ± SEM. Statistical significance was determined by one-way ANOVA with Tukey's post hoc test. $P$ value of Olaparib(−) vs others (from left to right): ****$P = 4.0419$e-7, $P = 0.3914$, ****$P = 1.3348$e-6; Olaparib(+) vs others (from left to right): ****$P = 9.2022$e-7, $P = 0.1477$; Olaparib(+)+Flag-RSMC vs Olaparib(+)+Flag-RSMC-16A ****$P = 3.1850$e-6. (G, H) PARP1 cooperates with ESCO1/2 to recruit Sororin. (G) Chromatin fractionation of HeLa cells treated with ESCO1/2 RNAi ± olaparib. Cells were harvested 5 h after release from the double-thymidine block. Immunoblots of whole-cell extracts (left) and chromatin fractions (right) are shown. (H) Quantification of chromatin-bound Sororin normalized to histone H3. Data represent mean ± SEM from four independent experiments. Statistical significance was determined by one-way ANOVA with Tukey's post hoc test. siNC(−PARPi) vs siNC(+PARPi) ****$P = 1.1595$e-5, siNC(−PARPi) vs siE1&2(−PARPi) ****$P = 6.4937$e-7, siNC(−PARPi) vs siE1&2(+PARPi) ****$P = 3.0904$e-8, siNC(+PARPi) vs siE1&2(-PARPi) $P = 0.0917$, siNC(+PARPi) vs siE1&2(+PARPi) ***$P = 0.003$, siE1&2(−PARPi) vs siE1&2(+PARPi) *$P = 0.0224$. Source data are available online for this figure.

Under unperturbed conditions, DNA replication-associated PARP activity is responsible for recruiting the single-strand break repair (SSBR) machinery that serves as an alternative unligated Okazaki fragment processing pathway (Azarm and Smith, 2020; Hanzlikova et al, 2017; Hanzlikova et al, 2018). Inhibition of this backup Okazaki fragment processing pathway leads to replication gaps underlying the synthetic lethality between PARPi and BRCA mutations (Cong et al, 2021). Here, we reveal another crucial role of S-phase PARP activity to recruit RSMC and subsequently Sororin in DNA replication-coupled cohesion. Together with this lagging strand-associated pathway, other key replisome components in leading or lagging strand such as PCNA and its loader RFC-Ctf18, MCM, Ctf4 cooperate for efficient Sororin recruitment through SMC3 acetylation. Besides DNA replication, PARP1 interacts with many cohesin factors physically and functionally (Jungmichel et al, 2013; Liu et al, 2018; Mondal et al, 2019; Mosler et al, 2022; Padella et al, 2022; Tothova et al, 2021). Hypomorph mutations or knockdown of many cohesin genes (SMC1, SMC3, RAD21/SCC1, STAG2) are synthetic lethal with PARPi olaparib which are attributed to the overlapping role of cohesin and PARP1/2 in preventing replication fork collapse (Liu et al, 2018; McLellan et al, 2012; Padella et al, 2022; Tothova et al, 2021). Fork collapse eventually causes DNA breaks that must be fixed largely via the complementary repair functions of PARP1/2 and cohesin (Mondal et al, 2019; Ronson et al, 2018; Wu et al, 2023; Zhou et al, 2023). Intriguingly, the cohesion state is required for bringing two sister chromatids together to promote homologous searching (Piazza et al, 2021). Collectively, all these studies highlight the intertwined intimate complementary relationships among PARP1, cohesin and DNA replication.

Besides acting as a primary S phase PARP target and subsequently a Sororin cofactor during sister chromatid cohesion, RSMC has been recently reported to bind 28S rRNA and play a critical role in ribosome biogenesis (Zhang et al, 2023b). Interestingly, PARP1 is also enriched in the nucleolus (Desnoyers et al, 1996; Fakan et al, 1988), modulating ribosomal DNA (rDNA) silencing, transcription, pre-rRNA processing, and pre-ribosome assembly (Huang and Kraus, 2022; Kim et al, 2020). Therefore, it will be of interest to investigate whether RSMC and PARP1 also interplay in this process.

In addition to RSMC, several other microproteins have been recently reported as vital regulators of cell cycle processes. For example, SHPRH-146aa limits the PCNA protein level and thereby glioblastoma cell proliferation by protecting full-length SHPRH, which is the ubiquitin ligase E3 of PCNA (Zhang et al, 2018a). FBXW7-185aa controls the c-Myc protein level by indirectly elevating the ubiquitylation of c-Myc (Yang et al, 2018). PINT-87aa, encoded by a p53-induced lncRNA (LINC-PINT), restricts transcription elongation of c-Myc and cyclin D1 genes by competing with RNA PolII for PAF1c association (Zhang et al, 2018b). These findings illuminate the physiologic significance as well as pathogenic implications of microproteins, an emerging class of proteins from the "dark proteome" (Jiang et al, 2021; Kustatscher et al, 2022a, b; Orr et al, 2020).

## Methods

### Reagents and tools table

| Reagent/resource | Reference or source | Identifier or catalog number |
|---|---|---|
| **Experimental models** | | |
| HEK-293 cells (*H. sapiens*) | Cong Liu | N/A |
| HEK293T *RSMC* heterozygous knockout (RSMC$^{+/-}$) | This study | N/A |
| HEK293T Flag-Sororin | This study | N/A |
| HEK293F (*H. sapiens*) | Dongyi Xu | N/A |
| HeLa (*H. sapiens*) | Xingzhi Xu | N/A |
| dCad9- GFP$_{14×}$-integrated HeLa | Xu et al (2020) (Nucleic Acids Research) | Prof. Dr. Baohui Chen, Zhejiang University, China |
| HeLa RSMC-3Flag | This study | N/A |
| HeLa PARP1 KO | Xingzhi Xu | N/A |
| **Recombinant DNA** | | |
| pRK5-Flag | This study | N/A |
| pRK5-Myc | This study | N/A |
| pEGFP-C1 | This study | N/A |
| pET28a | This study | N/A |
| pGEX-6p-1 | This study | N/A |
| pFast-Bac | This study | N/A |
| pX330 | Addgene | #42230 |
| pX458 | Addgene | #48138 |
| pLenti-Flag-Sororin | This study | N/A |
| pMD.2 G | This study | N/A |

| Reagent/resource | Reference or source | Identifier or catalog number |
|---|---|---|
| psPAX2 | This study | N/A |
| pcDNA3.0-Flag-PARP1 | This study | N/A |
| pRK5-Flag-Sororin | This study | N/A |
| pRK5-Flag-Sororin-12A | This study | N/A |
| pRK5-Flag-RSMC | This study | N/A |
| pRK5-Flag-RSMC-16A | This study | N/A |
| pRK5-Myc-RSMC | This study | N/A |
| pRK5-Myc-RAD21 | This study | N/A |
| pEGFP-RSMC | This study | N/A |
| pEGFP-RSMC-16A | This study | N/A |
| pEGFP-RSMC-ΔSIM | This study | N/A |
| pET28a-Flag-Sororin | This study | N/A |
| pET28a-Flag-Sororin-12A | This study | N/A |
| pGEX-6p-1-RSMC | This study | N/A |
| pGEX-6p-1-RSMC-16A | This study | N/A |
| pGEX-6p-1-RSMC-ΔSIM | This study | N/A |
| pGEX-6p-1-Sororin | This study | N/A |
| pGEX-6p-1-Wapl | This study | N/A |
| pFast-Bac-PDS5B | This study | N/A |
| pX330-RSMC-N-gRNA1 | This study | N/A |
| pX330-RSMC-N-gRNA2 | This study | N/A |
| pX458-RSMC-C-gRNA1 | This study | N/A |
| pX458-RSMC-C-gRNA1 | This study | N/A |
| **Antibodies** | | |
| Mouse anti-Flag | Sigma-Aldrich | F3165 |
| Rabbit anti-Myc | Proteintech | 16286-1-AP |
| Rabbit anti-GST | Beyotime | AF2299 |
| Mouse anti-GAPDH | Proteintech | 60004-1-Ig |
| Mouse anti-α-Tubulin | Proteintech | 66031-1-Ig |
| Rabbit anti-GFP | Proteintech | 50430-2-AP |
| Mouse anti-PAR | Trevigen | 4335-MC-100 |
| Mouse anti-Pan-PAR | Merck | MABE1016 |
| Rabbit anti-ESCO1 | Proteintech | 29821-1-AP |
| Rabbit anti-ESCO2 | Abcam | ab86003 |
| Rabbit anti-SMC3 | Abcam | ab275963 |
| Mouse anti-Acetyl-SMC3 (K105/106) | Merck | MABE1073 |
| Rabbit anti-Sororin (CDCA5) | Abcam | ab192237 |
| Rabbit anti-Wapl | Proteintech | 16370-1-AP |
| Rabbit anti-H3 | Proteintech | 17168-1-AP |
| Goat anti-mouse Alexa Fluor 488 | Jackson ImmunoResearch | AB_2338046 |
| Goat anti-rabbit Alexa Fluor 594 | Jackson ImmunoResearch | AB_2338059 |
| Goat anti-rabbit HRP | Sigma-Aldrich | A6154 |
| Goat anti-mouse HRP | Sigma-Aldrich | A4416 |
| **Chemicals, enzymes, and other reagents** | | |
| DMEM | Gibco | 11995500BT |
| Fetal bovine serum (FBS) | ABW | AB-FBS0500 |
| SMM 293-TI | Sino Biological | M293TI |
| SIM SF medium | Sino Biological | MSF1 |
| Lipofectamine 3000 | Invitrogen | L3000015 |
| Polyethylenimine (PEI) | Yeasen | 40816ES02 |
| Polybrene | Santa Cruz Biotechnology | 28728-55-4 |
| Giemsa | Amresco | 51811-82-6 |

| Reagent/resource | Reference or source | Identifier or catalog number |
|---|---|---|
| Olaparib | Selleck | S1060 |
| ProLong Gold antifade reagent | Invitrogen | P36934 |
| Emetine | Tocris | 7342 |
| ANTI-FLAG® M2 Affinity Gel | Sigma-Aldrich | A2220 |
| Immobilon-P-PVDF-Membrane | Millipore | IPVH00010 |
| Monolith™ NT.115 Standard Treated Capillaries (silica capillaries) | Nano Temper | MO-K002 |
| Recombinant Human PARP-1 Protein | Sino Biological | 11040-H08B |
| Gallotannin | Sigma-Aldrich | V900190 |
| Poly(ADP-ribose) Polymer/pADPr | R&D systems | 4336-100-01 |
| **Oligonucleotides and sequence-based reagents** | | |
| RSMC-EcoRI-F | This study | CCGGAATTCATGG GAGCTCCGGG GGGAAAG |
| RSMC-SalI-R | This study | ACGCGTCGACTTAGCT CTCATCTTCAAGGTC |
| RSMC-XbaI-F1 | This study | GATGACAAGGGATCC TCTAGAATGGGAGC TCCGGGGGGAAA |
| RSMC-HindIII-R1 | This study | TGGGCCATGGCGG CCAAGCTTTTAGCT CTCATCTTCAAGGT |
| RSMC-XhoI-R | This study | AGTCACGATGCG GCCGCTCGAGT TAGCTCTCATCTTCAAG |
| Sororin-BamHI-F1 | This study | ACGACGATGA CAAGGGATCCCATG TCTGGGAGGCGAACG |
| Sororin-HindIII-R1 | This study | GGGCCATGGCGGC CAAGCTTTCATTC AACCAGGAGATC |
| Sororin -EcoRI-F | This study | GGGGCCCCTGGG ATCCCCGGAATTCA TGTCTGGGAGGCGAACG |
| Sororin -XhoI-R | This study | GTCAGTCACGATG CGGCCGCTCGAGT CATTCAACCAGGAGATC |
| PDS5B-FseI-F | This study | GTTCCAGGGGC CCGGCCGGCCAAT GGCTCATTCAAAGAC TAGGACCAATG |
| PDS5B-AscI-R | This study | GCAGGCTCTA GAGGCGCGCCT CATCGCCGTTCC CTTTTTAGCACTTCGC |
| Wapl-BamHI-F | This study | CAAATGGGTCGC GGATCCATGACA TCCAGATTTGGG |
| Wapl-SalI-R | This study | CGCAAGCTTGT CGACCTAGCA ATGTTCCAA |
| ChIP-F1 | This study | AACCAAAAGCCA CTTAAACTCGT |
| ChIP-R1 | This study | CAGTAGGGG GCGCTCATATC |
| ChIP-F2 | This study | AGAGTTGTCTCTG GCAAACGG |
| ChIP-R2 | This study | CCAGGTCACC TTGCGTTATTG |
| C11ORF98-RT-F | This study | AAGTACGGACC GTGAACTGG |
| C11ORF98-RT-R | This study | CTTCTTCAGC TCCGTTCGGG |
| Sororin-RT-F | This study | GGCCATGAATG CCGAGTTTG |
| Sororin-RT-R | This study | AAGGCAGACA GTCCTCATGC |

| Reagent/resource | Reference or source | Identifier or catalog number |
|---|---|---|
| GAPDH-RT-F | This study | AGAAGGCTGGGGCTCATTTG |
| GAPDH-RT-R | This study | AGGGGCCATCCACAGTCTTC |
| siNC | This study | 5'-UUCUCCGAACGUGUCACGU-3' |
| siRSMC-1 | This study | 5'-CGGAAGUACGGACCGUGAACU-3' |
| siRSMC-2 | This study | 5'-GGAGUGGAAUCGCGACUAUGG-3' |
| siRSMC-3 | This study | 5'-GAAAGAUCAACCGGCCCCGAA-3' |
| siSororin-1 | This study | 5'-UGGAGGAGCUCGAGACGGA-3' |
| siSororin-2 | This study | 5'-GCCUAGGUGUCCUUGAGCU-3' |
| siESCO1-1 | This study | 5'-CCAGUGUUGAAAGACAAAUACUUCA-3' |
| siESCO1-2 | This study | 5'-GGACAAAGCUACAUGAUAG-3' |
| siESCO2-1 | This study | 5'-GACCCAACACCAGAUGGCAAGUUAU-3' |
| siESCO2-2 | This study | 5'-ACAGAAGAGUUUAACUGCUAAGUAU-3' |
| siWapl: | This study | 5'-CGGACUACCCUUAGCACAA-3' |
| siPARP1 | This study | 5'-GAGUCAAGAGAUGAAGGAAA-3' |
| siPARP2 | This study | 5'-AAGAUAGAGCGUGAAGGCG-3' |
| RSMC-N gRNA1 | This study | 5'-GGAAGUACGGACCGTGAAC-3' |
| RSMC-N gRNA2 | This study | 5'-GGCCGGTTGATCTTTCCCCC-3' |
| RSMC-C gRNA1 | This study | 5'-GCCCCCAGGATGTAGAAATGA-3' |
| RSMC-C-gRNA2 | This study | 5'-GCTTCATTTCTACATCCTGGG-3' |
| **Software** | | |
| GraphPad Prism 8.0 | https://www.graphpad.com | |
| ImageJ | https://imagej.net/ij/ | |
| Kaluza | https://www.mybeckman.cn/flow-cytometry/software/kaluza | |
| NanoTemper Analysis V2.3 | https://shop.nanotempertech.com/software/analysis-software/ | |
| **Other** | | |
| AxyPrep DNA Purification Kit | Axygen | AP-GX-250 |
| RNA extract kit | Magen | R4011-03 |
| HiScript II Reverse Transcriptase kit | Vazyme | R312-01 |
| Pierce Silver Stain Kit | Sigma-Aldrich | PROTSIL1 |
| Duolink PLA kit | Sigma-Aldrich | DUO92101 |
| ClonExpress MultiS Kit | Vazyme | C113-01 |
| GRAMM protein docking Web Server | https://gramm.compbio.ku.edu/ | |

## Cell lines and cell culture

HEK293T, HEK293T *RSMC* heterozygous knockout (RSMC$^{+/-}$), HEK293T Flag-Sororin, HeLa, dCad9- GFP$_{14\times}$-integrated HeLa, HeLa RSMC-3Flag, HeLa PARP1 WT and KO (provided by X Xu) cells were grown in DMEM (Gibco,11995500BT) supplemented with 10% FBS (ABW, AB-FBS0500) and antibiotics. HEK293F cells were cultured in SMM 293-TI (Sino Biological, M293TI) media supplemented with 1% FBS, 100 U/mL penicillin, and 100 µg/mL streptomycin. SF9 insect cells were grown in suspension in SIM SF medium (Sino Biological, MSF1).

## Plasmid construction, transfection, RNA interference (RNAi)

For overexpression studies, Sororin and RSMC cDNA were cloned into the pRK5-Flag, Myc, and pEGFP-C1 mammalian expression vectors, respectively. For recombinant protein purification, Flag-Sororin was cloned into the pET28a, while Wapl and RSMC were cloned into pGEX-6p-1 bacterial expression vectors. PDS5B cDNA was cloned into a baculoviral pFast-Bac vector. RSMC mutants were generated by PCR: RSMCΔSIM (RSMCΔ46-60Δ99-112-10A, 10 A = R30A, R31A, R33A, R35A, S65A, E87A, E89A, S98A, D113A, E115A), and RSMC-16A with additional lysine-to-alanine substitutions (K52A, K53A, K103A, R104A, K106A, K107A) introduced into the RSMC10A background. Sororin-12A mutants (K24, R27, R28, R31, K56, L60, K61, R62, K214, K218, K219, K22) were generated by PCR. All constructs were verified by sequencing. The primers:

RSMC-EcoRI-F: 5'-CCGGAATTCATGGGAGCTCCGGGGGGAAAG-3'

RSMC-SalI-R: 5'-ACGCGTCGACTTAGCTCTCATCTTCAAGGTC-3'

RSMC-XbaI-F1: 5'-GATGACAAGGGATCCTCTAGAATGGGAGCTCCGGGGGGAAA-3'

RSMC-HindIII-R1: 5'-TGGGCCATGGCGGCCAAGCTTTTAGCTCTCATCTTCAAGGT-3'

RSMC-XhoI-R: 5'-AGTCACGATGCGGCCGCTCGAGTTAGCTCTCATCTTCAAG-3'

Sororin-BamHI-F1: 5'-ACGACGATGACAAGGGATCCATGTCTGGGAGGCGAACG-3'

Sororin-HindIII-R1: 5'-GGGCCATGGCGGCCAAGCTTTCATTCAACCAGGAGATC-3'

Sororin-EcoRI-F: 5'-GGGGCCCCTGGGATCCCCGGAATTCATGTCTGGGAGGCGAACG-3'

Sororin-XhoI-R: 5'-GTCAGTCACGATGCGGCCGCTCGAGT-CATTCAACCAGGAGATC-3'

PDS5B-FseI-F: 5'-GTTCCAGGGGCCCGGCCGGCCAATGGCTCATTCAAAGACTAGGACCAATG-3'

PDS5B-AscI-R: 5'-GCAGGCTCTAGAGGCGCGCCTCATCGCCGTTCCCTTTTAGCACTTCGC-3'

Wapl-BamHI-F: 5'-CAAATGGGTCGCGGATCCATGACATC-CAGATTTGGG-3'

Wapl-SalI-R: 5'-CGCAAGCTTGTCGACCTAGCAATGTTCCAA-3'.

For RNAi (knockdown, KD) experiments, Sororin siRNAs were described previously (Schmitz et al, 2007). RSMC siRNAs were designed using the Designer of Small Interfering RNA (DSIR, http://biodev.extra.cea.fr/DSIR/DSIR.html). All siRNAs were synthesized by Sangon Biotech and are detailed below:

siNC: 5'-UUCUCCGAACGUGUCACGU-3'

siRSMC-1: 5'-CGGAAGUACGGACCGUGAACU-3'

siRSMC-2: 5'-GGAGUGGAAUCGCGACUAUGG-3'

siRSMC-3: 5'-GAAAGAUCAACCGGCCCCGAA-3'

siSororin-1: 5'-UGGAGGAGCUCGAGACGGA-3'

siSororin-2: 5'-GCCUAGGUGUCCUUGAGCU-3'

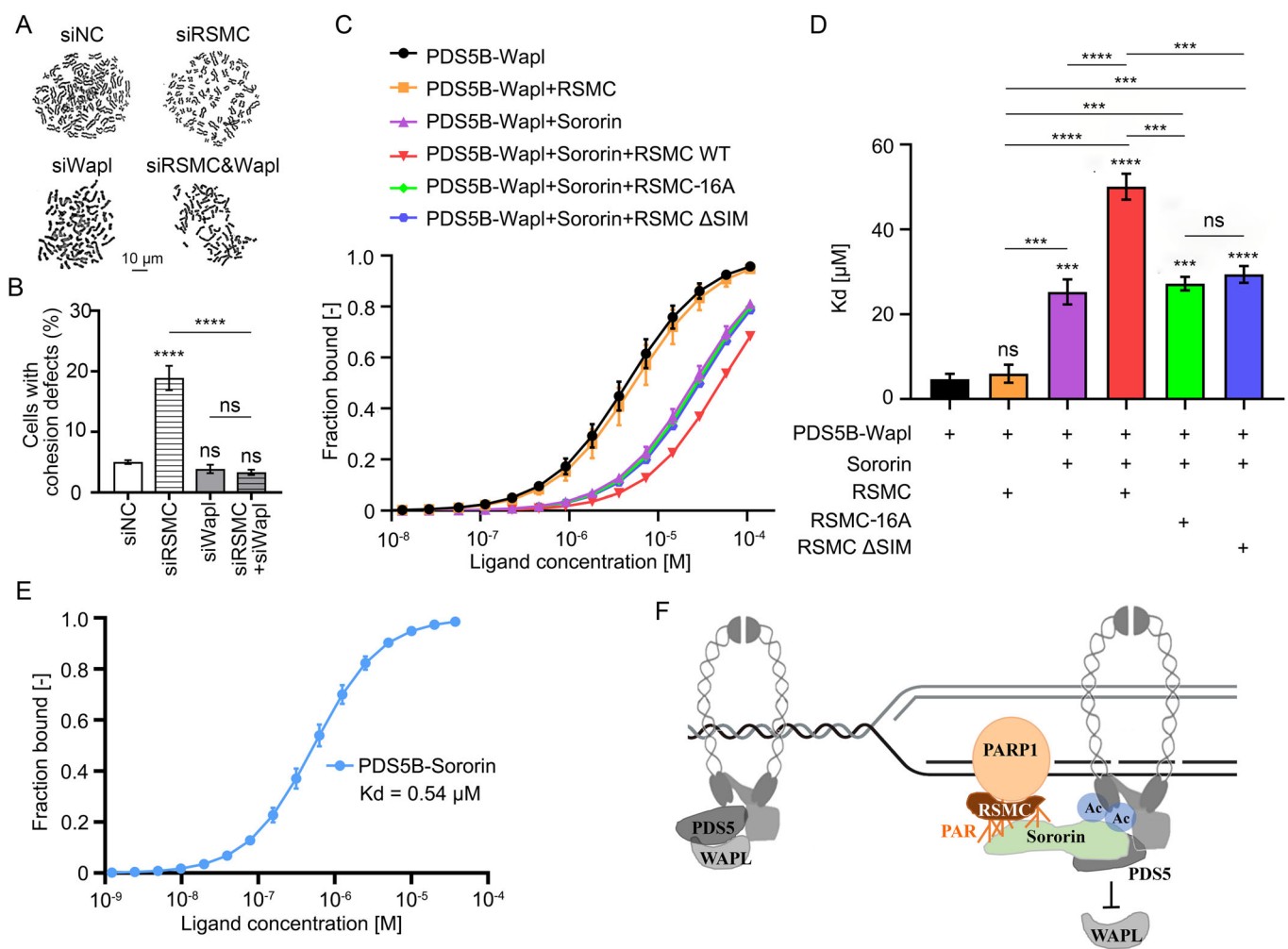

**Figure 7. RSMC stimulates the anti-Wapl activity of Sororin.**

(**A, B**) Wapl KD can fully rescue the cohesion defects caused by RSMC depletion. Cells were transfected with siRNA specifically targeted to RSMC and Wapl. Chromosome spreads analyses were performed. Bar = 10 µm. Data are presented as mean ± SEM from three independent assays. The statistical significance was calculated via one-way ANOVA with Tukey's post hoc test. siNC vs siRSMC ****$P$ = 9.0586e-5, siNC vs siWapl $P$ = 0.8788, siNC vs siRSMC+siWapl $P$ = 0.7031, siRSMC vs siRSMC+siWapl ****$P$ = 3.8718e-5, siWapl vs siRSMC+siWapl $P$ = 0.9836. (**C, D**) Quantification of binding affinity between fluorescently labeled PDS5B and Wapl by competitive microscale thermophoresis (MST). PDS5B was labeled with Cy5 and kept at a constant concentration at 34 nM. Wapl concentrations were serially diluted in a twofold gradient, starting from the highest concentration of 116 nM. The binding curve (**C**) and the calculated dissociation constant (Kd) (**D**) for Wapl-PDS5B interactions are shown with or without the following proteins: none (black), RSMC (17 nM, orange), Sororin (17 nM, purple), Sororin and RSMC (17 nM each, red), Sororin and RSMC-16A (17 nM each, green), Sororin and RSMCΔSIM (17 nM each, cyan). Data are presented as mean ± SEM from three independent assays. Statistical significance was evaluated using one-way ANOVA with Tukey's post hoc test followed by Tukey's post hoc test. $P$ value of PDS5B-Wapl vs others (from left to right), $P$ = 0.9984, ***$P$ = 0.004, ****$P$ = 8.5085e-8, ***$P$ = 1.5291e-4, ****$P$ = 6.1405e-5; $P$ value of PDS5B-Wapl+RSMC vs others (from left to right), ***$P$ = 0.006, ****$P$ = 1.1771e-7, ***$P$ = 0.0003, ***$P$ = 0.0001; PDS5B-Wapl+Sororin vs PDS5B-Wapl+Sororin+RSMC ****$P$ = 5.7977e-5; PDS5B-Wapl+Sororin+RSMC vs PDS5B-Wapl+Sororin+RSMC-16A ***$P$ = 0.0001; PDS5B-Wapl+Sororin+RSMC vs PDS5B-Wapl+Sororin+RSMC ΔSIM ***$P$ = 0.0003; PDS5B-Wapl+Sororin+RSMC-16A vs PDS5B-Wapl+Sororin+RSMC-ΔSIM $P$ = 0.9804. (**E**) Quantification of binding affinity between PDS5B and Sororin by MST. MST was performed as described above. Sororin concentrations were serially diluted in a twofold gradient, starting from the highest concentration of 40 µM. Data are presented as mean ± SEM from four independent assays. (**F**) A parallel Sororin recruitment model on the fork. During the normal S phase, in front of the fork, cohesin encircles chromatin dynamically due to the anti-cohesion activity of PDS5-Wapl. On the moving fork, PARP1 is activated by nicks and flaps in replication intermediates (unligated Okazaki fragments). Then, PARP1 targets an alternative ORF-encoded microprotein named RSMC in this study. PARylation of RSMC reinforces RSMC–Sororin interaction, which contributes to the recruitment of Sororin together with the well-established ESCO1/2-catalyzed SMC3 acetylation pathway. In addition, RSMC can stimulate the anti-Wapl activity of Sororin. In brief, two fork-associated activities, PARP1-mediated RSMC PARylation and ESCO1/2-mediated SMC3 acetylation, cooperatively recruit Sororin to establish sister chromatid cohesion in human cells. ESCO1/2 and their associated fork components including MCM, PCNA and CRL4[MMS22L] are omitted for brevity. Source data are available online for this figure.

siESCO1-1: 5'-CCAGUGUUGAAAGACAAAUACUUCA-3'
siESCO1-2: 5'-GGACAAAGCUACAUGAUAG-3'
siESCO2-1: 5'-GACCCAACACCAGAUGGCAAGUUAU-3'
siESCO2-2: 5'-ACAGAAGAGUUUAACUGCUAAGUAU-3'
siWapl: 5'-CGGACUACCCUUAGCACAA-3'

siPARP1: 5'-GAGUCAAGAGUGAAGGAAA-3'
siPARP2: 5'-AAGAUAGAGCGUGAAGGCG-3'.

All siRNAs were transfected using Lipofectamine 3000 (Lipo3000, L3000015, Invitrogen) according to the manufacturer's instructions. DNA plasmid transfection was carried out using PEI

(Polyethylenimine MW40000, Cat No. 40816ES02, Yeasen, Shanghai, China).

## Construction of cell lines

HEK293T RSMC$^{+/-}$ heterozygous knockout (RSMC$^{+/-}$) and HeLa RSMC-3Flag (RSMC-3Flag) cell lines were generated via CRISPR-Cas9 genome editing. For RSMC$^{+/-}$, two sgRNAs were designed to flank the RSMC start codon (ATG) and cloned pX330 CRISPR vector (Addgene, #42230). To preserve transcriptional integrity of the *LBHD1* gene, sgRNAs were positioned within the *LBHD1* intronic region. HEK293T cells co-transfected with two sgRNA plasmids were cultured for 48 h, then diluted into 96-well plates for isolating single colonies. *RSMC$^{+/-}$* clones were first screened by genomic PCR. PCR products of candidate clones were further cloned into the pRK5 vector, followed by sequencing to confirm the knockout of RSMC at the single-allele level.

For the HeLa RSMC-3Flag cell line, CRISPR-Cas9-mediated homology-directed repair (HDR) was employed to insert the 3×Flag tag at the RSMC C-terminus. A homologous recombination donor (RSMC-3Flag and G418 cassette) was generated using the ClonExpress MultiS Kit (Vazyme C113-01). Two gRNAs targeting the C-terminus of genomic RSMC were cloned into pX458 vector (Addgene #48138). HeLa cells were co-transfected with pX458-gRNAs and the donor plasmid via Lipo3000, followed with G418 selection (800 μg/mL). Single colonies were validated by genomic PCR, sequencing, and immunoblots.

RSMC-N gRNA1:5'-GGAAGTACGGACCGTGAAC-3'
RSMC-N gRNA2:5'-GGCCGGTTGATCTTTCCCCC-3'
RSMC-C gRNA1: 5'-GCCCCCAGGATGTAGAAATGA-3'
RSMC-C-gRNA2: 5'-GCTTCATTTCTACATCCTGGG-3'.

For the construction of the HEK293T Flag-Sororin stable cell line, pLenti-Flag-Sororin was co-transfected into HEK293T cells alongside two helper plasmids: pMD.2 G and psPAX2. Viral supernatants were collected 48 h post-transfection, centrifuged at 500×g for 10 min at 4 °C, and filtered through a 0.45-μm membrane. Prior to lentiviral transduction, HEK293T cells were seeded in 6 cm dishes and cultured to ~80% confluence. Cells were incubated with virus-containing supernatants supplemented with 8 μg/mL Polybrene (Santa Cruz Biotechnology) for 24 h. Following incubation, the medium was replaced with fresh virus-containing medium for an additional 24 h. Transduced cells were then selected using 5 μg/mL blasticidin for 48 h and subjected to immunoblotting to confirm Sororin protein expression.

## Cell synchronization and flow cytometry analysis

Cells were synchronized at the G1/early S-phase boundary using a double-thymidine (DT) block protocol: cells were cultured with 2 mM thymidine for 16 h, then released to thymidine-free medium for 8 h, and with 2 mM thymidine for another 16 h. For flow cytometry analysis, cells were harvested by trypsinization at the indicated time points after release, washed with PBS, and fixed in ice-cold 70% ethanol at −20 °C overnight. The fixed cells were treated with 20 μg/mL RNase A and 10 μg/mL propidium iodide in PBS at 37 °C for 30 min. Data were collected by CytoFLEX S flow cytometer (Beckman counter) and were analyzed using Kaluza Analysis software.

## Chromosome spreads

HEK293T cells were treated with nocodazole (100 ng/mL) for 4 h to enrich mitotic cells, then were collected by mitotic shake-off, resuspended in 75 mM KCl hypotonic buffer for 5 min (37 °C). Cell pellets were collected by centrifugation at 300×g for 5 min, then were fixed by Carnoy's solution (methanol: glacial acetic acid =3:1, v/v) for 20 min at RT and at 4 °C for overnight. After fixation, cells were dropped onto ice-cold glass slides, air-dried, followed with Giemsa (Amresco, 51811-82-6, 10% in PBS) staining. Images were captured using a Leica DM6000 or Zeiss MetaSystems microscope equipped with a ×100 oil objective (Sun et al, 2019). To inhibit PARPs, cells were pretreated with 10 μM olaparib (Selleck, S1060) for 48 h prior to harvest. HeLa PARP1 WT or KO cells were first transfected with siRNAs targeting PARP2 or control siRNAs. Six hours after transfection, olaparib was added to the culture. Mitotic cells were then harvested 48 h following olaparib treatment.

## Chromatin labeling

HeLa cells stably expressing dCas9-GFP$_{14×}$ were transfected with siRNAs of control, RSMC, and Sororin. Cells were synchronized at the G1/S boundary via a double-thymidine block. sgRNAs targeting *LMNA* loci (Xu et al, 2020) were transfected after the first release. At 4 h post-second thymidine release (mid-S phase), cells were fixed with 4% paraformaldehyde (PFA), and nuclei were counterstained with DAPI. Confocal imaging was performed using a Leica SP8 laser-scanning microscope (Leica Microsystems) with a 100× oil-immersion objective (NA 1.4). Z-stacks (0.3 μm intervals) were captured. Sister chromatid pairs ("doublets") with diameters of 0.3–1.2 μm were quantified as described (Stanyte et al, 2018).

## Proximity ligation assay (PLA)

The PLA was performed using the Duolink PLA kit (Sigma-Aldrich) according to the manufacturer's protocol. In brief, RSMC-3Flag cells were pre-extracted (0.1% Triton X-100, 5 min), fixed (4% PFA, 15 min), and permeabilized (0.1% Triton X-100, 10 min). After blocking (37 °C, 1 h), cells were incubated with primary antibodies (anti-Sororin (1:100), anti-PARP1 (1:200), and anti-Flag (1:200)) overnight at 4 °C. After washing 2 × 5 min with Duolink® Wash Buffer A (RT), PLUS and MINUS PLA probes were hybridized (1 h, 37 °C). Unbound probes were removed by washing 2 × 5 min (Wash Buffer A), followed by ligation (37 °C, 30 min). Ligated products were amplified via amplification (90 min, 37 °C) and washed twice (2 × 10 min with high-stringency Wash Buffer B). Nuclei were counterstained with DAPI and mounted with ProLong Gold antifade reagent (Invitrogen, P36934). For PARP inhibition (PARPi), cells were pretreated with 10 μM olaparib 1 h prior to the second release. G1/S and middle (Mid) S phase RSMC-3Flag cells were collected at 0 h and 5 h after release from the DT block. Images were captured by a confocal microscope (Leica SP8) equipped with a ×40 oil objective.

## Immunofluorescence (IF)

To stain endogenous RSMC and Sororin, RSMC-3Flag cells on coverslips were fixed with 4% PFA and permeabilized with 0.1%

Triton X-100 in PBS for 5 min. Cells were blocked with 1% BSA in PBS for 30 min at RT, then incubated with primary antibodies (anti-Flag (1:200) and anti-Sororin (1:100)) at 37 °C for 1 h. After washing with PBST (0.05% Triton X-100), secondary antibodies (anti-mouse Alexa Fluor 488 (1:500) and anti-rabbit Alexa Fluor 594 (1:500)) were stained at 37 °C for 1 h, and nuclei were stained with DAPI for 5 min. Mounting and imaging were performed as described in PLA.

## Immunofluorescence staining of chromatin-bound Sororin

Immunofluorescence staining of chromatin-bound Sororin was modified from Nishiyama et al, 2010. In brief, S-phase cells were released from a double-thymidine (DT) block. Post-trypsinization, cells were hypotonically treated with 75 mM KCl for 5 min and stopped by 1× PBS quick wash, then pre-extracted with 0.1% Triton X-100 for 2 min, then washed once by 1× PBS, followed by 4% PFA fixation for 10 min. Cell pellets were washed with PBS and fixed in Carnoy's solution for 10 min before being dropped on the cover slides. Afterward, IF staining of Sororin was conducted as described above. For RNAi, cells were transfected with target-specific siRNAs prior to cell synchronization. For drug treatment, 10 µM olaparib was added 1 h before the second release and 2 µM Emetine (Tocris, 7342) was added 1 h before cell collection.

## Immunoprecipitation-coupled mass spectrometry (IP-MS)

HEK293F cells were transfected with Flag-Sororin for 72 h, then were lysed in lysis buffer (25 mM Tris pH 7.5, 150 mM NaCl, 1% [v/v] NP-40, 5 mM EDTA, 5% [v/v] glycerol, 10 mM β-glycerophosphate, 5 mM NaF, 1 mM DTT, 0.1 mM $Na_3VO_4$, and protease inhibitor cocktail). Flag-Sororin was immunoprecipitated with anti-Flag M2 agarose (Sigma, A2220) for 4 h at 4 °C. Beads were washed 5 times with the lysis buffer and eluted by 2 µg/µL Flag-peptides. 1% eluates were resolved by SDS-PAGE and stained with Pierce Silver Stain Kit (Sigma, PROTSIL1). Remaining eluates were acetone-precipitated and analyzed by mass spectrometry (Q Exactive Hybrid Quadrupole-Orbitrap Mass Spectrometer, Thermo Fisher) to identify Sororin-interacting proteins.

## Protein purification and in vitro pull-down

6×His-Flag-tagged Sororin was expressed and purified using the prokaryotic expression systems according to standard protocols. In brief, 6×His-Flag-Sororin was cloned into pET28a. Transformed E. coli BL21 (DE3) cells were induced with 0.04 mM IPTG at 25 °C for 3 h. Cells were pelleted (10000 g, 8 min) and resuspended in lysis buffer (50 mM $Na_2HPO_4$-$NaH_2PO_4$, 150 mM NaCl, 0.05% Triton X-100, pH 8.0). Cell lysates were incubated with $Ni^{2+}$ Sepharose for 2 h at 4 °C. Beads washed five times with lysis buffer supplemented with 20 mM imidazole. Protein eluted in elution buffer (50 mM $Na_2HPO_4$-$NaH_2PO_4$, pH 8.0, 300 mM NaCl, and 300 mM imidazole).

For GST-tagged protein purification, Sororin, Wapl, RSMC, RSMC10A, RSMC-16A, and RSMCΔSIM were cloned into pGEX6p-1. RSMC and its mutant were induced by 0.02 mM IPTG at 37 °C for 3 h, and GST-Sororin/GST-Wapl was induced by 0.04 mM IPTG at 25 °C for 3 h. Cells lysed in GST lysis buffer (50 mM Tris pH 8.0, 150 mM NaCl, 0.05% Triton X-100, 2 mM DTT). Lysate incubated with Glutathione Sepharose 4B (GE Healthcare, 17075601). Post-washing, proteins eluted with 10 mM reduced glutathione in buffer (50 mM Tris pH 8.0, 150 mM NaCl). For Sororin/Wapl proteins, GST tags were removed by on-column digestion with PreScission Protease (4 °C, 16 h).

N-terminal 6×His-3×Flag-PDS5B cloned into pFastBac1 (Ouyang et al, 2016). SF9 cells infected with recombinant baculovirus for 72 h. Cells resuspended in lysis buffer (50 mM Tris pH 7.5, 150 mM KCl, 0.1% Triton X-100, 20 mM imidazole, 1 mM PMSF, 10 mM β-mercaptoethanol). After centrifugation at 20,000× g for 1 h, supernatants were incubated with $Ni^{2+}$ Sepharose and applied to two successive washes with buffer A (50 mM Tris pH 7.5, 1.2 M KCl, 20 mM imidazole) and buffer B (50 mM Tris pH 7.5, 150 mM NaCl). Proteins were eluted with the elution buffer (50 mM Tris pH 7.5, 150 mM NaCl, 500 mM imidazole). All proteins were dialyzed into storage buffer (50 mM Tris pH 7.5, 150 mM NaCl, and 10% [v/v] glycerol) and stored at −80 °C.

For in vitro pull-down assays, 1 µg purified 6×His-Flag-tagged Sororin was incubated with anti-Flag M2 agarose in 250 µL binding buffer (50 mM Tris pH 8.0, 150 mM NaCl, 0.05% Triton X-100) at 4 °C for 2 h. After pelleting beads and discarding unbound supernatant, 1 µg GST-tagged RSMC and its mutants were added to the bead-immobilized Sororin in fresh binding buffer and incubated for 2 h. Beads were washed three times and resuspended with 100 µL loading buffer and boiled for 10 min before Coomassie brilliant blue (CBB) staining.

For Flag-PARP1 purification, HEK293T cells were transfected with Flag-PARP1 for 72 h, then were lysed in lysis buffer (25 mM Tris pH 7.5, 150 mM NaCl, 1% [v/v] NP-40, 5 mM EDTA, 5% [v/v] glycerol, 10 mM β-glycerophosphate, 5 mM NaF, 1 mM DTT, 0.1 mM $Na_3VO_4$, and protease inhibitor cocktail). Flag-PARP1 was immunoprecipitated with anti-Flag M2 agarose (Sigma, A2220) for 2 h at 4 °C. Beads were washed five times with the lysis buffer and eluted by 1 µg/µL Flag-peptides.

## Chromatin fractionation

Cells were washed with PBS and lysed in CSK buffer (10 mM PIPES pH 7.0, 100 mM NaCl, 1 mM $MgCl_2$, 1 mM EDTA, 300 mM sucrose, 10 mM NaF, 20 mM β-glycerophosphate, 100 mM $Na_3VO_4$, 0.5% Triton X-100, and 1× protease inhibitor cocktail). Lysates were incubated on ice for 20 min with vortexing every 5 min. After collecting whole-cell extracts (WCE), the remaining lysate was centrifuged (2000× g, 5 min) to isolate chromatin-enriched pellet (chromatin) and supernatant (SN). The chromatin pellet was washed twice and resuspended in CSK buffer. All fractions were analyzed by immunoblotting. For drug treatment, 10 µM olaparib was added 1 h before the second DT release, and 2 µM emetine was added 1 h before cell collection. Emetine was employed for short periods to minimize side effects other than inhibition of Okazaki fragment synthesis (Burhans et al, 1991).

## Co-immunoprecipitation (CoIP)

For immunoprecipitation assays, cells were resuspended in lysis buffer (50 mM Tris, pH 8.0, 150 mM NaCl, 50 mM EDTA, 1% NP-40, 10% glycerol) supplemented with 50 µg/mL PMSF and 1× protease inhibitor cocktail (Cat No.20109ES05, Yeasen). Lysates were

centrifuged at 20,000× g for 20 min. The lysates were incubated with anti-Flag M2 agarose (Sigma, A2220) or protein G beads immobilized with anti-SMC3 antibody (1 μg) at 4 °C for 2 h. Subsequently, the beads were washed three times with the lysis buffer. Bound protein eluted by boiling in the loading buffer at 95 °C for 10 min.

## Immunoblotting

To obtain whole-cell extracts for immunoblotting, cells were resuspended with RIPA buffer (50 mM Tris pH 8.0, 250 mM NaCl, 1% Triton X-100, 0.5% sodium deoxycholate, 0.05% SDS, 1 mM DTT) on ice for 20 min, followed by centrifugation at 20,000× g for 20 min. Supernatants stored as whole-cell protein extracts. All samples were run on SDS-PAGE and transferred to PVDF membranes (Millipore, IPVH00010). Immunoblotting was performed with indicated primary antibodies diluted in the antibody dilution buffer and incubated at room temperature for 1 h. The peroxidase-conjugated goat anti-rabbit (Sigma-Aldrich, A6154) or peroxidase-conjugated goat anti-mouse (Sigma-Aldrich, A4416) second antibodies were diluted 1:10,000 and incubated 1 h at RT. Signals were detected using eECL western blot kit.

Flag, Sigma-Aldrich, F3165; Myc,Proteintech, 16286-1-AP; GST, Beyotime, AF2299; GAPDH, Proteintech, 60004-1-Ig; α-Tubulin, Proteintech, 66031-1-Ig; GFP, Proteintech, 50430-2-AP; anti-PAR, Trevigen, 4335-MC-100; anti-Pan-PAR, Merck, MABE1016; ESCO1, Proteintech, 29821-1-AP; ESCO2, Abcam, ab86003; SMC3, Abcam, ab275963; Acetyl-SMC3 (K105/106), Merck, MABE1073; Sororin (CDCA5), Abcam, ab192237; Wapl, Proteintech, 16370-1-AP; H3, Proteintech, 17168-1-AP.

## Chromatin immunoprecipitation (ChIP)

Cells were synchronized at the G1/S phase using a double-thymidine block. Samples were collected at 0 h (G1/S phase), 5 h (S phase), and 10 h (G2 phase) after release from the block. To enrich G2-phase cells, 10 μM RO3306 was added at the 5-h time point and maintained for 5 h before collection. ChIP was performed as previously described (Ladurner et al, 2016) with minor modifications. Briefly, HeLa cells expressing Flag-tagged proteins were crosslinked with 1.42% formaldehyde (15 min, RT) and quenched with 125 mM glycine (5 min, RT). Cells were lysed in ChIP lysis buffer (50 mM Tris–HCl pH 8.0, 10 mM EDTA, 1% SDS, and protease inhibitors), sonicated to obtain 200–400 bp chromatin fragments, and incubated with anti-Flag M2 agarose (pre-coated with 0.1 mg/mL BSA and sheared salmon sperm DNA) at 4 °C overnight. Sequential washes were performed twice for each wash buffer: A (20 mM Tris–HCl pH 8.0, 2 mM EDTA pH 8.0, 1% Triton X-100, 150 mM NaCl, 0.1% SDS, 1 mM PMSF), B (20 mM Tris–HCl pH 8.0, 2 mM EDTA pH 8.0, 1% Triton X-100, 500 mM NaCl, 0.1% SDS, 1 mM PMSF), C (10 mM Tris–HCl pH 8.0, 2 mM EDTA pH 8.0, 250 mM LiCl, 0.5% NP-40, 0.5% deoxycholate), and TE buffer. After washing steps, DNA was eluted twice with elution buffer (25 mM Tris–HCl pH 7.5, 5 mM EDTA pH 8.0, 0.5% SDS, 65 °C, 20 min), treated with 10 mg/mL RNase-A (65 °C, 1 h) and with 20 mg/mL proteinase K (65 °C, 5 h), then was purified using the AxyPrep DNA Purification Kit (Axygen, AP-GX-250). The products were analyzed by qPCR on a qTOWER3G touch real-time PCR detection System (Analytik Jena) using Taq Pro Universal SYBR qPCR Master Mix (Vazyme,

Q712-02). Primers were: F1: AACCAAAAGCCACTTAAACTCGT; R1: CAGTAGGGGGCGCTCATATC; F2: AGAGTTGTCTCTGG-CAAACGG; R2: CCAGGTCACCTTGCGTTATTG.

## Reverse transcription-PCR and real-time PCR

Total mRNA was extracted from cells using the RNA extract kit (Magen, R4011-03). 3 μg of RNA was reverse-transcribed into cDNA using the HiScript II Reverse Transcriptase kit (Vazyme, R312-01). The mRNA of C11ORF98 (F: AAGTACGGACCGT-GAACTGG; R: CTTCTTCAGCTCCGTTCGGG) and Sororin (F: GGCCATGAATGCCGAGTTTG; R: AAGGCAGACAGTCCTCAT GC) were quantified and normalized to GAPDH (F: AGAAGGC TGGGGCTCATTTG; R: AGGGGCCATCCACAGTCTTC).

## Prediction of protein complex structures

The protein structures of RSMC and Sororin were obtained from the AlphaFold2 online Database. The interaction between RSMC and Sororin was predicted by the GRAMM protein docking Web Server (Tovchigrechko and Vakser, 2006). The resulting model was presented with PyMOL.

## Competitive microscale thermophoresis (MST)

The MST analysis was performed as described previously (Wienken et al, 2010). Purified His-Flag-PDS5B was fluorescently labeled with Cy5. In PDS5B-Wapl interaction assays, Cy5-labeled PDS5B was maintained at 34 nM and the concentrations of Wapl were twofold diluted (116–3.54 nM). After 20 min incubation, samples were loaded into silica capillaries (Nano Temper, MO-K002). Measurements were performed at 25 °C in buffer containing 50 mM Tris pH 7.4, 150 mM NaCl, 10% glycerol, and 0.05% Tween-20 using a Monolith NT.115 system (NanoTemper Technologies) at a constant LED power of 20% and MST power of 40%. The assays were repeated at least three times for each affinity measurement. Data were analyzed using the NanoTemper Analysis V2.3 provided by the manufacturer. In competitive interaction assays, 34 nM Cy5-labeled-PDS5B and gradient-diluted concentrations of Wapl were combined to enable interaction at RT for 10 min, Sororin and RSMC were then added to the mixture before the measurements.

## In vitro PARylation assays

For the in vitro PARylation assay, purified GST-RSMC or GST-RSMC-16A was bound to Glutathione Sepharose 4B beads and incubated with the PARP1 enzyme (Sino Biological, 11040-H08B) for 1 h at 37 °C in reaction buffer (50 mM Tris–HCl pH 8.0, 150 mM NaCl, 10 mM MgCl$_2$, 10% glycerol, 1 mM DTT) supplemented with or without 200 μM NAD$^+$, followed with three times wash with reaction buffer supplied with 300 mM NaCl. The reaction was stopped by adding the SDS loading buffer before immunoblotting.

## In vitro PARylation followed by pull-down assays

To check PARylated RSMC binding to Sororin, RSMC PARylation was performed as described above. After washing away free PARP1 and NAD$^+$, 1 μg Sororin was added and incubated at 4 °C for

another 1 h, then the bound proteins were washed three times and boiled for 10 min at 95 °C before immunoblotting with the indicated antibodies.

## PAR-binding assays

The PAR-binding assay was performed as described (Peng et al, 2021) with minor modifications. Briefly, 6×His-Flag-Sororin or Sororin PAR mutants were incubated with Ni-NTA agarose for 1 h at 4 °C. After three times washing, beads were incubated with 20 nM PAR in NETN buffer (20 mM Tris–HCl pH 8.0, 0.15 M NaCl, 1 mM EDTA, 0.5% NP-40, and protease inhibitor cocktail) for 1 h. Then the bound proteins were washed 3 times and stopped by loading buffer boiled for 10 min at 95 °C, and analyzed by dot-blotting onto PVDF membranes with an anti-PAR antibody.

## In vivo PARylation

In vivo PARylation was performed as described (Jungmichel et al, 2013) with minor modifications. HEK293T cells transfected with GFP-tagged expression plasmids were treated with 10 μM Gallotannin (Sigma, V900190) for 2 h, followed by 1 mM $H_2O_2$ for 10 min to activate PARPs. Cells were resuspended in lysis buffer (50 mM Tris pH 7.5, 400 mM NaCl, 1 mM EDTA, 1% NP-40, 0.1% Na-deoxycholate, and protease inhibitor cocktail supplemented with 2 mM Na-orthovanadate, 5 mM NaF, 5 mM glycerol-2-phosphate, 10 μM Gallotannin, and 50 μM olaparib). Lysates were diluted to final concentrations of 200 mM NaCl, 0.5% NP-40, and 0.05% Na-deoxycholate, and incubated with GFP-binding protein (GBP) beads for 2 h at 4 °C. Beads were washed three times with lysis buffer, and proteins were eluted by boiling in 2× loading buffer for 5 min before immunoblotting.

## Statistical analysis

Statistical analyses were performed using unpaired Student *t* test, one-way, and two-way ANOVA in the GraphPad Prism 8.0 software. The cellular phenotypes were analyzed blindly.

# Data availability

This study includes no data deposited in external repositories.

The source data of this paper are collected in the following database record: biostudies:S-SCDT-10_1038-S44318-025-00641-8.

# Peer review information

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

## Acknowledgements

This study was supported by National Natural Science Foundation of China [32300074 to JZ; 32101039 to WH; 32161133015 to HL]; Shenzhen Science and Technology Program [JCYJ20240813143001003 to WH; JCYJ20240813143701003 to HL; RCBS20231211090516017 to JZ; KQTD20240729102249044 to WH]; Medical Scientific Research Foundation of Guangdong Province [A2024342 to WH]; China Postdoctoral Science Foundation [2024M752102 to JZ]; SZU Top Ranking Project [860/00000210 to HL]; SZU Top Ranking Project "B&RsacFMS" [868/0000080208 to HL and WH]. We're grateful to Drs. Baohui Chen (Zhejiang University, China), Cong Liu (Sichuan University), Bin Peng (Shenzhen University, China) for generously sharing materials; to all members of the Lou laboratory for discussion; to the Instrumental Analysis Center of Shenzhen University for the assistance with fluorescence imaging analysis, and to the Instrumental Platform of State Key Laboratory of Plant Environmental Resilience at China Agricultural University for assistance with MST experiments.

## Author contributions

**Meiqian Jiang**: Data curation; Formal analysis; Validation; Investigation; Visualization; Methodology; Writing—original draft; Writing—review and editing. **Jiaxin Zhang**: Data curation; Formal analysis; Funding acquisition; Validation; Investigation; Visualization; Methodology; Writing—original draft; Writing—review and editing. **Jiankun He**: Data curation; Investigation; Methodology. **Yu Miao**: Resources; Methodology. **Linhui Wang**: Data curation;

Investigation. **Haitao Zhong**: Data curation; Investigation. **Yingying Gong**: Data curation; Investigation. **Zhen Li**: Data curation; Formal analysis; Investigation; Methodology. **Li-Lin Du**: Resources; Methodology. **Xingzhi Xu**: Resources; Methodology. **Chunlai Chen**: Resources; Methodology. **Alibek Ydyrys**: Funding acquisition. **Yisui Xia**: Writing—review and editing. **Qinhong Cao**: Investigation; Methodology. **Huiqiang Lou**: Conceptualization; Resources; Formal analysis; Supervision; Funding acquisition; Validation; Visualization; Methodology; Writing—original draft; Project administration; Writing—review and editing. **Wenya Hou**: Conceptualization; Resources; Data curation; Formal analysis; Supervision; Funding acquisition; Validation; Investigation; Visualization; Methodology; Writing—original draft; Project administration; Writing—review and editing.

Source data underlying figure panels in this paper may have individual authorship assigned. Where available, figure panel/source data authorship is listed in the following database record: biostudies:S-SCDT-10_1038-S44318-025-00641-8.

## Disclosure and competing interests statement

The authors declare no competing interests.

# Expanded View Figures

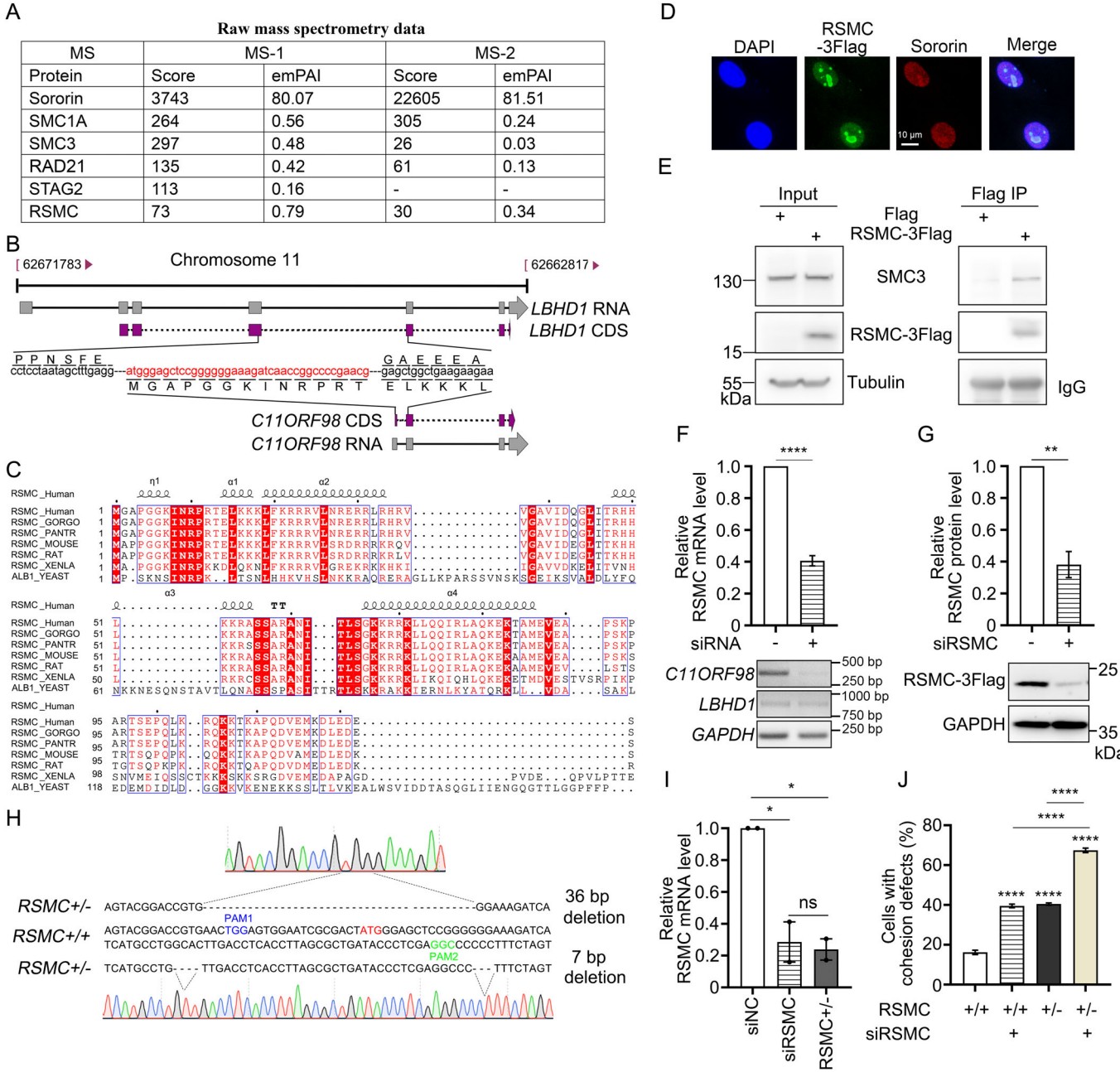

◀

**Figure EV1.  (related to Fig. 1). Identification of RSMC as a Sororin cofactor.**

(A) Raw mass spectrometry data of Flag-Sororin immunoprecipitates (related to Fig. 1A). (B) The *C11ORF98* gene is an alternative open reading frame of the *LBHD1* gene. (C) RSMC is a conserved microprotein in eukaryotes. Sequence alignment of RSMC orthologs from the indicated species from T-coffee with the default setting and modified by ESPript 3.0. (D) Subcellular localization of RSMC. HeLa cells harboring endogenous RSMC-3Flag were synchronized at early S phase via double-thymidine block. After release into S phase for 6 h, cells were fixed and immunostained with the indicated antibodies. Nuclei were counterstained using DAPI. Scale Bar = 10 μm. (E) RSMC also interacts with the cohesin subunit SMC3. Co-immunoprecipitation (CoIP) assays were performed using RSMC-3Flag cells. Lysates were immunoprecipitated with anti-Flag M2 agarose, followed by immunoblotting with anti-SMC3 antibody to detect co-precipitated SMC3. 3% of samples were loaded as input. (F, G) Verification of the efficiency of RSMC (*C11ORF98*) KD by RNAi. (F) The mRNA level of RSMC was checked by qRT-PCR and compared to the *GAPDH* control. Note that *RSMC* mRNA was efficiently knocked down whereas the *LBHD1* mRNA level was not affected, Quantitative data are presented as mean ± SEM from four independent assays. The statistical significance was calculated via Student's *t* test. ****$P = 1.622e\text{-}6$. (G) The protein level of RSMC was measured by immunoblotting. RSMC-3Flag HeLa cells were transfected with RSMC siRNA for 48 h, then cells were collected and subjected to immunoblotting analysis. Quantitative data are presented as mean ± SEM from three independent assays. The statistical significance was calculated via Student's *t* test. **$P = 0.0017$ (G). (H) Verification of CRISPR-Cas9-edited RSMC$^{+/-}$ clone by genomic sequencing. On-target deletion mutations were around the start codon of RSMC. (I) Real-time PCRs revealed that the RSMC$^{+/-}$ and RSMC siRNAs exhibited similar reduction of RSMC's mRNA levels. Quantitative data are presented as mean ± SEM from two independent assays. The statistical analyses were performed by one-way ANOVA with Tukey's post hoc test. siNC *vs* siRSMC *$P = 0.0179$, siNC *vs* RSMC$^{+/-}$ *$P = 0.0149$, siRSMC *vs* RSMC$^{+/-}$ $P = 0.9172$. (J) Combination of RSMC KD with RSMC$^{+/-}$ caused an additive cohesion defect. HEK293T WT and RSMC$^{+/-}$ cells were transfected with RSMC siRNA. After 48 h, mitotic cells were collected and subjected to chromosome spreads analysis. More than 300 mitotic cells from three independent assays were scored. Mean ± SEM are shown. The statistical analyses were performed by One-way ANOVA with Tukey's post hoc test. RSMC$^{+/+}$ *vs* RSMC$^{+/+}$+siRSMC ****$P = 6.0835e\text{-}7$, RSMC$^{+/+}$ *vs* RSMC$^{+/-}$ ****$P = 4.2887e\text{-}7$, RSMC$^{+/+}$ *vs* RSMC$^{+/-}$+siRSMC ****$P = 2.4757e\text{-}9$, RSMC$^{+/+}$+siRSMC *vs* RSMC$^{+/-}$+siRSMC ****$P = 1.1824e\text{-}7$, RSMC$^{+/-}$ *vs* RSMC$^{+/-}$+siRSMC ****$P = 1.6412e\text{-}7$. Source data are available online for this figure.

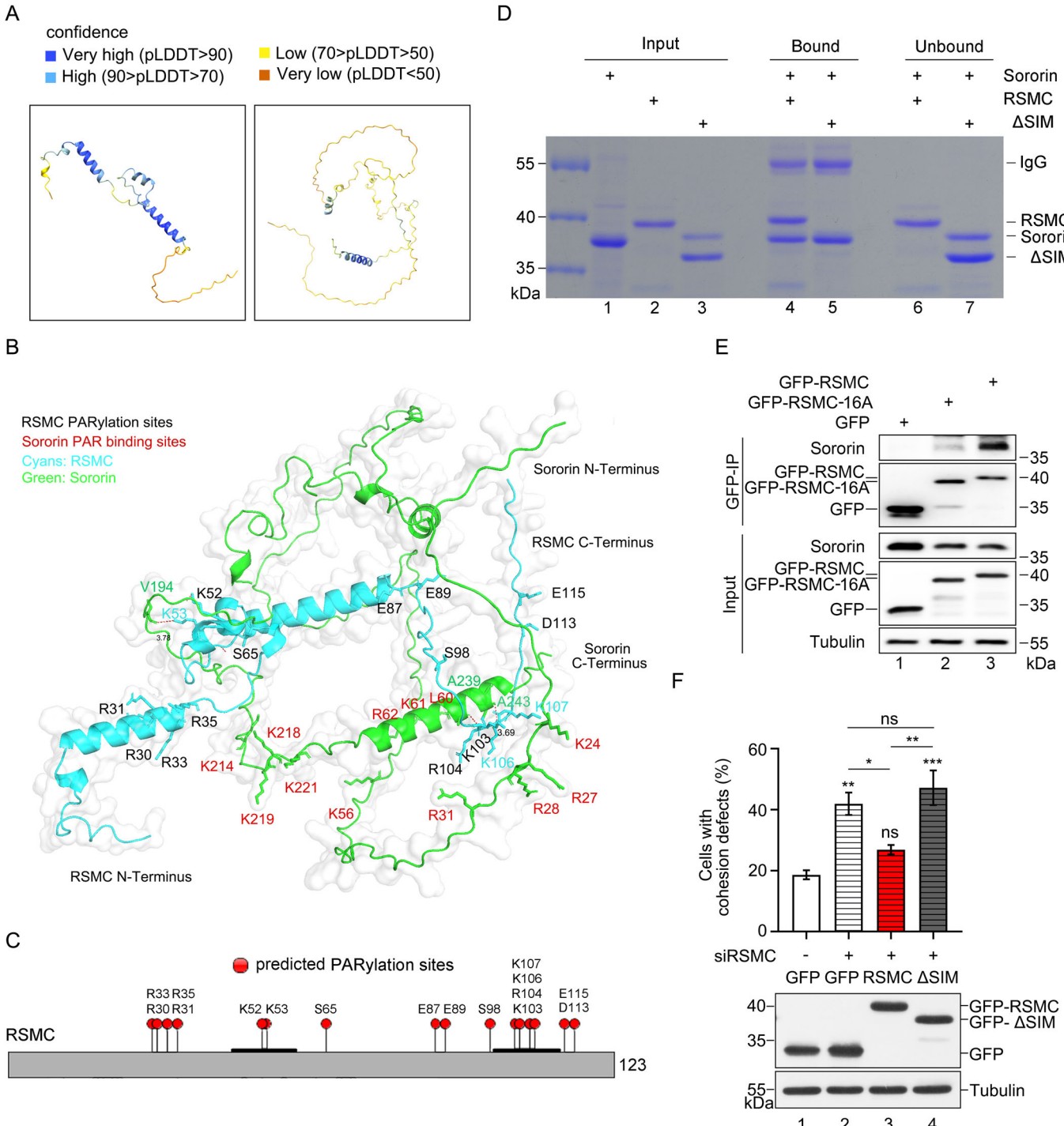

**A**

confidence

■ Very high (pLDDT>90)
■ High (90>pLDDT>70)
□ Low (70>pLDDT>50)
□ Very low (pLDDT<50)

**B**

RSMC PARylation sites
Sororin PAR binding sites
Cyans: RSMC
Green: Sororin

**C**

● predicted PARylation sites

RSMC

**D**

**E**

**F**

◄

**Figure EV2.   (related to Fig. 2). Mapping RSMC–Sororin interfaces.**

(A) The RSMC and Sororin's structure were predicted by the α-Fold2 database. (B) The putative interacting interfaces between RSMC (cyan) and Sororin (green) predicted by GRAMM-X (https://gramm.compbio.ku.edu/). The predicted basal RSMC/Sororin interaction via hydrogen bonds was marked in the corresponding protein's color (RSMC K53, K106, K107, and Sororin V194, A239, A243). The predicted 16 PARylation sites of RSMC were labeled (K53, K106, K107, R30, R31, R33, R35, K52, S65, E87, E89, S98, K103, R104, D113, E115, in black). And the PAR-binding motifs of Sororin (related to Fig. EV5D) contain 12 amino acids (K24, R27, R28, R31, K56, L60, K61, R62, K214, K218, K219, K221, in red). (C) A schematic diagram of RSMC illustrating the RSMC–Sororin interfaces including black lines labeled a.a.46-60 and a.a.99-112, as well as their adjacent polar side chain residues (red dots) likely undergoing PARylation predicted by ADPredict. (https://www.adpredict.net/). (D) Mapping the interaction domains of RSMC with Sororin by pull-down assays. 6×His-Flag-Sororin was immobilized on anti-Flag M2 agarose. Then the GST-RSMC or its mutant proteins RSMCΔ46-60Δ99-112-10A (RSMCΔSIM) were incubated with immobilized Sororin. Sororin-bound proteins were separated by SDS-PAGE and detected by CBB. 30% of samples were loaded as input. (E) Compared to RSMC WT, RSMC-16A shows an attenuated interaction with Sororin. HEK293T cells were transfected with GFP-RSMC WT or GFP-RSMC-16A for 48 h. Lysates were immunoprecipitated with GBP beads. 3% of samples were loaded as input. (F) SIMs of RSMC are indispensable for its cohesion function. HEK293T cells transfected with RSMC siRNA and then with GFP, GFP-RSMC and GFP-RSMCΔSIM, respectively. Chromosome spreads analysis was conducted with four independent biological repeats as described in Fig. 1E. Data are presented as means ± SEM. The statistical significance was calculated via one-way ANOVA with Tukey's post hoc test. The expression of the indicated proteins was shown by immunoblotting. siRSMC(−)+GFP *vs* siRSMC(+)+GFP **$P = 0.0028$, siRSMC(−)+GFP *vs* siRSMC(+)+RSMC $P = 0.3929$, siRSMC(−)+GFP *vs* siRSMC(+)+ΔSIM ***$P = 0.0005$, siRSMC(+)+GFP *vs* siRSMC(+)+RSMC *$P = 0.0478$, siRSMC(+)+RSMC *vs* siRSMC(+)+ΔSIM **$P = 0.0077$, siRSMC(+)+GFP *vs* siRSMC(+)+ΔSIM $P = 0.7272$. Source data are available online for this figure.

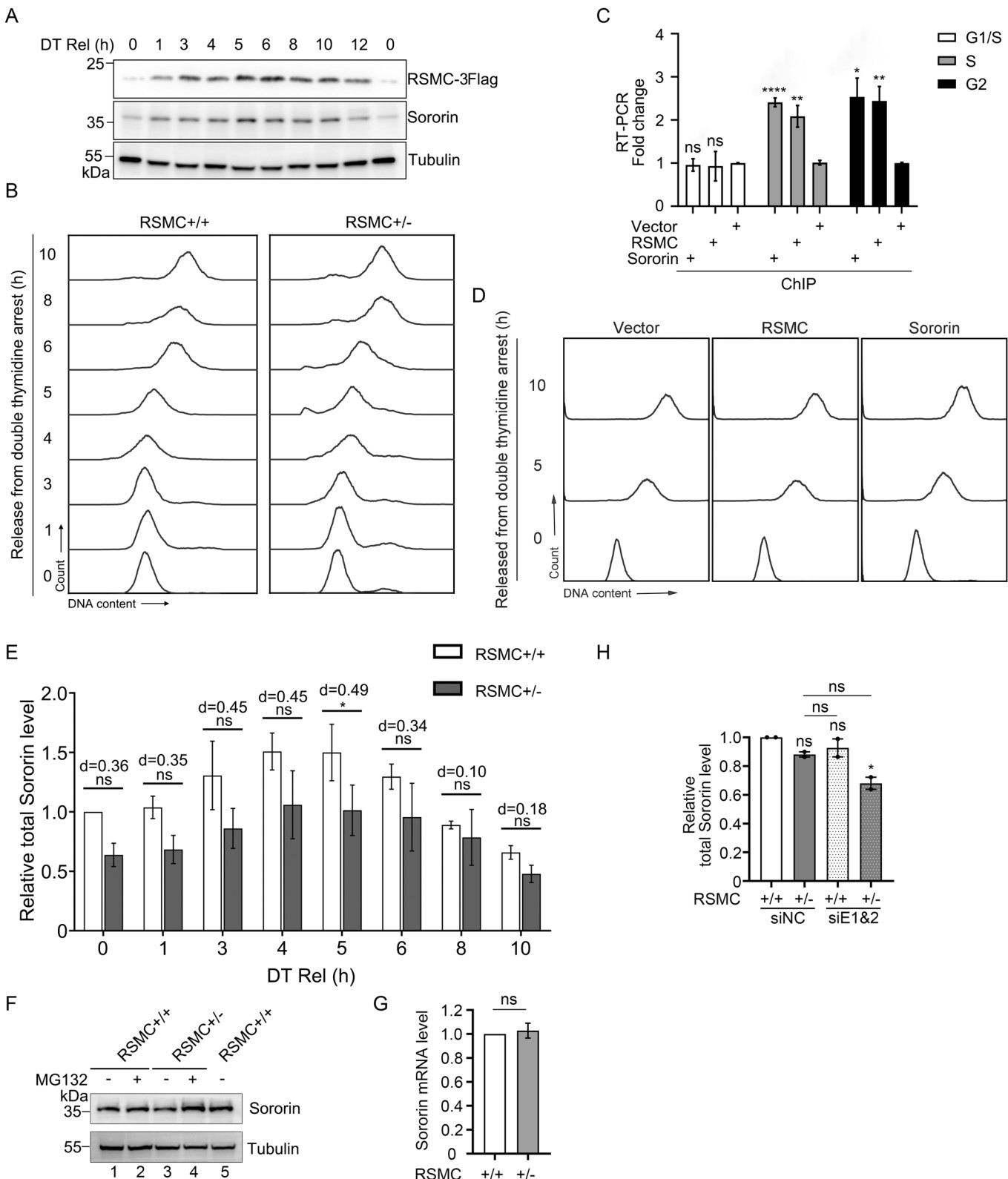

◀  **Figure EV3.   (related to Figs. 3 and 4). The cell-cycle-regulated RSMC protein levels and its contribution to Sororin stability.**

(A) RSMC and Sororin exhibit similar fluctuations in protein levels throughout the cell cycle. RSMC-3Flag cells were harvested at the indicated time points following release from a double-thymidine block (early S phase). Immunoblotting analyses were performed using antibodies against the indicated proteins. (B) RSMC$^{+/-}$ shows a subtle effect on the cell cycle progression. HEK293T RSMC$^{+/+}$ and RSMC$^{+/-}$ cells were synchronized in early S phase by a double-thymidine block and harvested at the indicated time points. Cell samples were fixed and stained with PI and analyzed by flow cytometry (related to Fig. 3D). (C) Like Sororin, RSMC binds to specific cohesin-associated regions (CARs). HeLa cells transfected with 9Flag, 9Flag-RSMC, or Flag-Sororin were subjected to ChIP with anti-Flag M2 agarose. qPCR analysis was conducted using primers targeting known Sororin/cohesin-associated CARs (CARs, Human Chr8:134214868-134215746, related to Fig. 3C). Data are presented as means ± SEM from three independent assays. The statistical significance was calculated via two-way ANOVA with Tukey's post hoc test. $P$ value from left to right, $p = 0.9550$, $P = 0.9731$, ****$P = 5.2387e-8$, **$P = 0.0084$, *$P = 0.0202$, **$P = 0.0068$. (D) Verification of the cell cycle progression. HeLa cells transfected with 9Flag, 9Flag-RSMC, or Flag-Sororin were synchronized in early S phase by a double-thymidine block and harvested at the indicated time points. Cell samples were fixed and stained with PI and analyzed by flow cytometry (related to Fig. 3C). (E) RSMC$^{+/-}$ reduces the total Sororin protein levels during S phase. The relative Sororin/H3 ratio of the whole-cell lysates was calculated from three independent assays. Data are presented as means ± SEM. The statistical significance was calculated via two-way ANOVA with Sidak's multiple comparisons test. The "d" represents the difference value of chromatin-bound Sororin in RSMC$^{+/+}$ and RSMC$^{+/-}$ (related to Fig. 3D). $P$ value from left to right, $P = 0.1825$, $P = 0.2004$, $p = 0.0620$, $P = 0.0597$, *$P = 0.0359$, $P = 0.2369$, $P = 0.9948$, $P = 0.8797$. (F) The decreased Sororin in RSMC$^{+/-}$ was recovered by the proteasome inhibitor MG132 treatment for 5 h. (G) Sororin mRNA levels were not changed in RSMC$^{+/-}$. Data are presented as means ± SEM from three independent assays. The statistical significance was calculated via Student's *t* test. $P = 0.6681$. (H) Quantification of Sororin in whole-cell extracts. The relative Sororin/H3 ratio of WCE was calculated from two independent assays. Data are presented as means ± SEM and were normalized to siNC RSMC$^{+/+}$. The statistical significance was calculated via one-way ANOVA followed by Tukey's post hoc test. $P$ value of RSMC$^{+/+}$(siNC) *vs* others (from left to right), $P = 0.2731$, $P = 0.5911$, *$P = 0.0146$; $P$ value of RSMC$^{+/-}$(siNC) *vs* others (from left to right), $P = 0.8363$, $P = 0.0693$; RSMC$^{+/+}$(siE1&2) *vs* RSMC$^{+/-}$(siE1&2) *$P = 0.0359$. (related to Fig. 4C,D). Source data are available online for this figure.

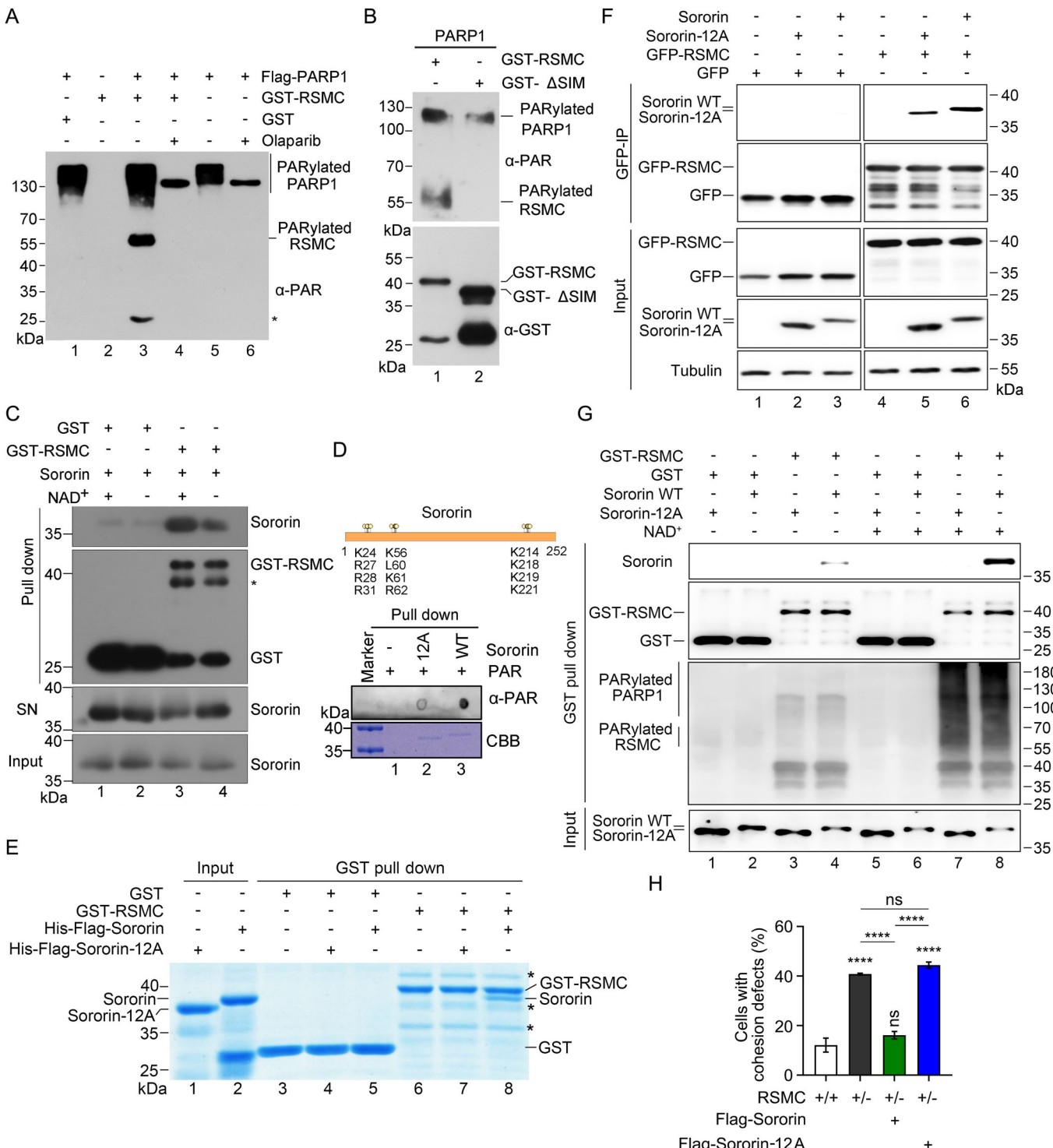

◀ **Figure EV4.** (related to Fig. 5). **PARP1-mediated PARylation of RSMC enhances RSMC–Sororin interaction.**

(A) The in vitro PARylation assay was performed using purified GST or GST-RSMC and Flag-PARP1 in the absence or presence of olaparib (20 μM) in the reaction buffer at 37 °C for 1 h. Immunoblotting was carried out with an anti-PAR monoclonal antibody. (B) RSMC is PARylated by PARP1 at SIMs. The in vitro PARylation assay was performed using purified GST-RSMC or GST-RSMCΔSIM and PARP1 in the reaction buffer at 37 °C for 1 h. Immunoblotting was carried out with an anti-PAR monoclonal antibody. (C) PARylation enhances RSMC's binding to Sororin. In vitro PARylation of RSMC was conducted as above in the absence or presence of $NAD^+$. After the removal of PARP1 and $NAD^+$, Sororin was incubated with immobilized RSMC. The resins were then washed three times, resuspended with 1×loading buffer and subjected to immunoblotting analyses with the indicated antibodies. (D) Diagram of Sororin with three PAR-binding motifs containing 12 residues and Sororin binds to PAR chains in vitro via 3 PAR-binding motifs. PAR (20 nM) was incubated with Sororin or Sororin-12A immobilized on $Ni^{2+}$ and subjected to dot blot assays. (E) Sororin-12A is defective in binding RSMC in vitro. Immobilized GST-RSMC or GST (control) on glutathione beads was incubated with purified 6×His-Flag-Sororin WT or 12 A mutant. After washing, bound proteins were eluted and analyzed by SDS-PAGE followed by CBB staining. (F) Sororin-12A weakens its association with RSMC in cells. Cells were transfected with GFP, GFP-RSMC, Flag-Sororin WT, or Flag-Sororin-12A. Lysates were subjected to GFP-IP before immunoblotting with anti-Flag and anti-GFP antibodies. (G) Sororin-12A hardly binds PARylated RSMC in vitro. Immobilized RSMC was PARylated in the presence of $NAD^+$, then PARP1 and $NAD^+$ were washed away. Afterwards, PARylated RSMC was incubated with either Sororin WT or 12 A mutant. Bound proteins were analyzed by immunoblotting using anti-Sororin, anti-GST and anti-PAR antibodies. (H) Sororin-12A is defective in cohesion. Flag-tagged vector, Sororin WT, or Sororin-12A was transfected into HEK293T WT or $RSMC^{+/-}$ cells. Cohesion defects were quantified. Data are presented as means ± SEM from three independent assays. The statistical significance was calculated via one-way ANOVA followed by Tukey's post hoc test. *P* value of $RSMC^{+/+}$ vs others (from left to right), \*\*\*\**P* = 1.2454e-5, *P* = 0.4180, \*\*\*\**P* = 5.0813e-6; *P* value of $RSMC^{+/-}$ vs others (from left to right), \*\*\*\**P* = 3.8307e-5, *P* = 0.4946; $RSMC^{+/-}$+Flag-Sororin vs $RSMC^{+/-}$+Flag-Sororin-12A \*\*\*\**P* = 1.3754e-5. Source data are available online for this figure.

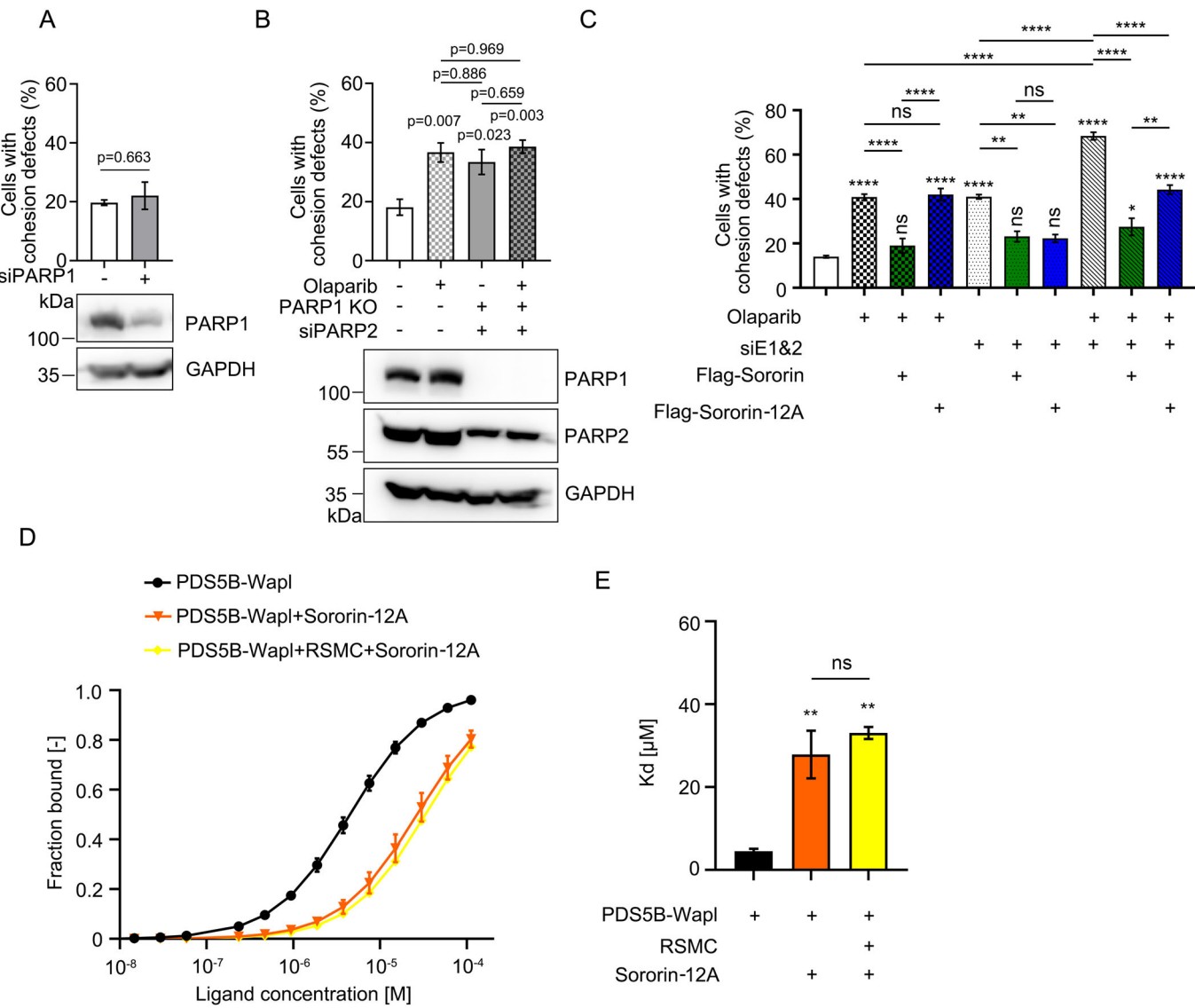

**Figure EV5.** (related to Figs. 6 and 7). PARPs cooperate with ESCO1/2 for Sororin functioning in cohesion.

(A) siPARP1 does not affect sister chromatid cohesion. Mitotic cells were collected after 48 h PARP1 RNAi and subjected to chromosome spread analyses. Sister chromatid cohesion defects were quantified from three independent assays. Immunoblots of PARP1 and GAPDH were shown below each column to confirm knockdown efficiency. Data represent mean ± SEM. The statistical significance was calculated via Student's t test. P = 0.3984. (B) PARP activity is required for cohesion. PARP1 KO HeLa cells were transfected with PARP2 or control siRNA with or without 10 μM olaparib treatment. Mitotic cells were analyzed by chromosome spreads after 48 h. More than 300 mitotic cells were scored from four independent assays. Immunoblots of PARP1, PARP2 and GAPDH were shown below each column to confirm knockout and knockdown efficiency. Mean ± SEM are shown. The statistical significance was calculated via one-way ANOVA followed by Tukey's post hoc test. P value of WT vs others (from left to right), **P = 0.0065, *P = 0.0231, **P = 0.0031; P value of Olaparib(+) vs others (from left to right), P = 0.8862, P = 0.9699; PARP1 KO+siPARP2 vs PARP1 KO +siPARP2+Olaparib P = 0.6589. (C) PARPs function redundantly with ESCO1/2 in cohesion. HeLa cells transfected with ESCO1/2 or control siRNA treated ± 10 μM olaparib. Mitotic cells were analyzed by chromosome spreads after 48 h. Over 300 mitotic cells were scored from three independent assays. Mean ± SEM are shown. The statistical analyses were performed by One-way ANOVA followed by Tukey's post hoc test. P value of WT vs others (from left to right), ****P = 4.3539e-6, P = 0.8503, ****P = 2.3505e-6, ****P = 4.0305e-6, P = 0.3061, P = 0.4265, ****P = 5.6290e-11, *P = 0.0142, ****P = 7.6187e-7; Olaparib(+) vs Olaparib(+)+Flag-Sororin ****P = 7.2240e-5, Olaparib(+) vs Olaparib(+)+Flag-Sororin-12A P > 0.9999, Olaparib(+) vs Olaparib(+)+siE1&2 ****P = 3.0100e-6; Olaparib(+)+Flag-Sororin vs Olaparib(+)+Flag-Sororin-12A****P = 3.6410e-5; siE1&2 vs siE1&2+Flag-Sororin **P = 0.0028, siE1&2 vs siE1&2+Flag-Sororin-12A**P = 0.0017, siE1&2 vs siE1&2+ Olaparib(+) ****P = 3.2485e-6; siE1&2+Flag-Sororin vs siE1&2+Flag-Sororin-12A P > 0.9999; siE1&2+Olaparib(+) vs siE1&2+Olaparib(+)+Flag-Sororin****P = 6.6151e-9, siE1&2+Olaparib(+) vs siE1&2+Olaparib(+)+Flag-Sororin-12A ****P = 1.9114e-5; siE1&2+Olaparib(+)+Flag-Sororin vs siE1&2+Olaparib(+)+Flag-Sororin **P = 0.0017. (D, E) Quantification of Wapl-PDS5B interactions by competitive microscale thermophoresis (MST). Cy5-labeled PDS5B (34 nM constant concentration) was titrated with 2-fold serially diluted Wapl (starting at 116 nM). (D) Binding curves and (E) calculated dissociation constants (Kd) under: control (black), 17 nM Sororin-12A (orange) and 17 nM each Sororin-12A and RSMC (yellow). Data represent mean ± SEM from three independent experiments. Statistical significance was determined by one-way ANOVA followed by Tukey's post hoc test. PDS5B-Wapl vs. PDS5B-Wapl+Sororin-12A **P = 0.0071, PDS5B-Wapl vs. PDS5B-Wapl+Sororin-12A + RSMC **P = 0.0026, PDS5B-Wapl+Sororin-12A vs. PDS5B-Wapl+Sororin-12A + RSMC P = 0.5637. Source data are available online for this figure.

