## [Peer Review File · The EMBO Journal]

S-phase PARylation of microprotein RSMC enhances the function of Sororin in sister chromatid cohesion

Meiqian Jiang, Jiabin Zhang, Jiankun He, Yu Miao, Linhui Wang, Haitao Zhong, Yingying Gong, Zhen Li, Li-Lin Du, Xingzhi Xu, Chunlai Chen, Alibek Ydyrys, Yisui Xia, Qinhong Cao, Huiqiang Lou, and Wenya Hou

Corresponding author(s): Wenya Hou (wenya.hou@szu.edu.cn) , Huiqiang Lou (lou@cau.edu.cn)

Review Timeline:

Submission Date:	10th Mar 25
Editorial Decision:	25th Apr 25
Revision Received:	23rd Jul 25
Editorial Decision:	30th Sep 25
Revision Received:	16th Oct 25
Accepted:	4th Nov 25

Editor: Hartmut Vodermaier

Transaction Report:

Dr. Wenya Hou
Shenzhen University
Shenzhen 518060
China

25th Apr 2025

Re: EMBOJ-2025-120713
S phase PARylation of microprotein RSMC enhances Sororin's function in sister chromatid cohesion

Dear Dr. Hou,

Thank you for submitting your manuscript on RSMC as a PARylation-regulated new Sororin interactor to The EMBO Journal. We sent it to three expert referees, who have now provided the feedback copied below. As you will see, all reviewers consider your findings - if substantiated - interesting and potentially important for the understanding of cohesion regulation. However, especially referees 1 and 2 also raise several substantive concerns, and ask for more decisive evidence able to rule out possible alternative explanations for the findings. Key concerns include the need for more direct evidence that RSMC PARylation facilitates Sororin recruitment in cells, and that RSMC is not affecting Sororin via translation; possible confounding effects of chromatin-trapped PARP enzymes; and the need for acute RSMC depletion to better define its specific roles. Other important referee points relate to insufficient experimental descriptions, partly moderate effects, and unconvincing cell staining and immunoblot data.

At this point, I feel that the nature and extent of these issues make it unclear whether they may be satisfactorily addressable through a regular major revision. However, in light of the potential overall interest acknowledged by all reviewers, I would nevertheless be willing to give you a chance to revise the manuscript in response to the referees' criticisms. It is clear that this may require substantial further time and efforts, but should you be able to validate and improve the present analyses along the lines suggested by the referees, we would remain interested in pursuing this study further for publication. Since it is our policy to allow only a single round of major revision, I would very much encourage you to contact me with a revision plan and preliminary point-by-point response already during the early stages of your revision work, so that we could discuss if and how the main points could be resolved. Of course, we would also be open to extension of the default three-months revision period if needed; our 'scooping protection' (meaning that competing work appearing elsewhere in the meantime will not affect our considerations of your study) would of course remain valid also throughout such an extension.

Detailed information on preparing, formatting and uploading a revised manuscript can be found below and in our Guide to Authors. Thank you again for the opportunity to consider this work for The EMBO Journal, and I look forward to hearing from you in due time.

With kind regards,

Hartmut Vodermaier

- size of the scale bars that are mandatory for all micrograph panels
- the statistical test used to generate error bars and P-values
- the type error bars (e.g., S.E.M., S.D.)

- the number (n) and nature (biological or technical replicate) of independent experiments underlying each data point
- Figures may not include error bars for experiments with $n < 3$; scatter plots showing individual data points should be used instead.

9) To facilitate reproducibility and cross-laboratory adoption of methodologies, please structure the Materials & Methods section as outlined in our guide to authors, including a completed Reagents and Tools Table that can be downloaded from our author guidelines as well (<https://www.embopress.org/page/journal/14602075/authorguide#structuredmethods>).

10) Digital image enhancement is acceptable practice, as long as it accurately represents the original data and conforms to community standards. If a figure has been subjected to significant electronic manipulation, this must be clearly noted in the figure legend and/or the 'Materials and Methods' section. The editors reserve the right to request original versions of figures and the original images that were used to assemble the figure. Finally, we generally encourage uploading of numerical as well as gel/blot image source data; for details see: embopress.org/page/journal/14602075/authorguide#sourcedata

Revision to The EMBO Journal should be submitted online within 90 days, unless an extension has been requested and approved by the editor; please click on the link below to submit the revision online before 24th Jul 2025:

Link Not Available

If you choose to alternatively have this study further considered by another EMBO Press publication, please use the following hyperlink to directly transfer the manuscript, optionally with inclusion of referee reports and identities:

Link Not Available

Referee #1:

Sororin, a cohesin-binding protein, is essential for sister chromatid cohesion establishment during DNA replication. Sororin is recruited to the cohesin complex via Pds5 and Escp1/2-dependent acetylation of Smc3. Although it has been established that acetylation of Smc3 is required for Sororin recruitment to cohesin (i.e. to chromatin), it has also been suggested that Smc3 acetylation is not sufficient and that an unknown mechanism dependent on DNA replication is required for full activation of Sororin recruitment.

In this study, Jiang and colleagues identified a novel Sororin-interacting protein, RSMC, in human HEK293T cells by IP-MS. In HEK293 cells, overexpressed RSMC or overexpressed Sororin can associate with endogenous Sororin or RSMC, respectively. PLA assay shows that RSMC and Sororin or cohesin are in close proximity to each other in human cells. Knockdown (RNAi) or

heterozygously expression of RSMC (RSMC +/-) exhibits mild cohesion defects. The PARylation-deficient RSMC 16A mutant cannot restore cohesion defect in RSMC RNAi cells. Finally, RSMC WT but not 16A facilitates Wapl-Pds5B dissociation in vitro. From these lines of evidence, the authors conclude that PARP1-dependent PARylation of RSMC strengthens its association with Sororin, thereby enhancing Sororin's chromatin recruitment by antagonizing Wapl's cohesin dissociation activity. Although the authors' findings that RSMC interacts with Sororin and that RSMC has an impact on sister chromatid cohesion in human cells are novel and interesting, the direct evidence showing that PARylation of RSMC facilitates Sororin recruitment to chromatin is insufficient, especially in vivo. Therefore, the relationship between PARylation and Sororin-dependent sister chromatid cohesion in this study is still unclear.

Another major concern in this manuscript is that all the presenting data could not rule out the possibility that RSMC facilitate Sororin translation because (a) protein level of Sororin is decreased in RSMC +/- cells (Figures S3D and S3E), (b) Sororin overexpression rescues the cohesion defect (Figure 2D), (c) mRNA level of Sororin does not change (Figure S3F), and (d) RSMC have recently been reported to bind 28S rRNA and play a critical role in ribosome biogenesis (Zhang et al., 2023). To exclude this possibility, it is necessary to test whether the Sororin mutant, which is unable to bind to RSMC, restores the cohesion defect in RSMC +/- cells (see below @ Major point (2) and (4)). Unless the authors clarify these points, it would be impossible to conclude that PARylation of RSMC facilitates Sororin recruitment to chromatin. For these reasons, I would recommend revising the manuscript.

Major points:

- 1) The authors showed that partial depletion of RSMC caused partial cohesion defect in human somatic cells and that RSMC-WT, but not the PARylation-deficient RSMC-16A mutant, could rescue the defect. However, as mentioned above, one of the major concerns is that direct evidence that PARylation of RSMC is required for Sororin binding to RSMC and/or chromatin is insufficient. To more directly test the importance of RSMC PARylation, the authors should perform an IP experiment (rather than PLA) in HEK293T cells if RSMC WT but not 16A associates with Sororin.
- 2) Related to the previous point, if binding between Sororin and PARylated RSMC is important for cohesion, then Sororin mutant that does not bind to RSMC should show the similar cohesion defect. Does Sororin WT/12A associate with PARylated RSMC both in vitro and in vivo? Furthermore, when endogenous Sororin is substituted with Sororin 12A, does Sororin 12A restore the cohesion defect? More importantly, is cohesion defect shown under PARP inhibition or in RSMC +/- cells restored by expression of either Sororin WT or 12A? These points are also important to rule out the possibility that RSMC facilitate Sororin translation, that is another big concern in this manuscript.
- 3) Although it's shown that non-PARylatable RSMC 16A is less bound to Sororin in vitro (Figure 2B), where even RSMC-WT is not PARylated, there is no direct evidence that PARylated RSMC are better associated with Sororin compared to non-PARylated RSMC in vitro or in vivo. The authors should test this.
- 4) It is unclear which part of Sororin is responsible for the association with RSMC from AlphaFold prediction (Figure S2B). Although the authors identified 12 K/R residues in Sororin that are required for PAR binding in vitro, are these 12 residues also important for association with RSMC even without PARylation?
- 5) In both RSMC RNAi and RSMC +/- cells, the cohesion defect is relatively mild compared to Sororin KD. This could be due to knockdown efficiency. Can the authors generate RSMC -/- cells or perform RSMC RNAi in RSMC +/- cells to completely deplete RSMC from the cells? This might result in more severe cohesion defects.
- 6) In Figures 2A and 2B, RSMC in its unPARylated state binds to Sororin in vitro, whereas RSMC 16A or Δ SIM cannot. This suggests that PARylation is not required for RSMC association with Sororin in vitro. The authors should explain how these 16 K/R residues on RSMC contribute to Sororin binding and how PARylation of these residues facilitates the association with Sororin.
- 7) In Figure 7F, RSMC facilitates Sororin activity to dissociate Wapl from Pds5B. Does PARylated RSMC further promote the Sororin activity of Wapl dissociation?
- 8) Related to the previous point, does PARylated RSMC facilitate Pds5-Sororin binding? If so, what about the binding between Pds5 and Sororin-12A?
- 9) It seems that RSMC levels increase during S phase. Is this DNA replication dependent? In other words, is RSMC accumulation suppressed by replication inhibition? In addition, if PARP1 activity is also replication-dependent, then PARylated RSMC should also be decreased after replication inhibition. Is this the case?

Minor points:

- 10) In PLA experiments (Figures 1C, 5B, and 5H), the images are too dark to recognize the red colors. Please improve the figures.
- 11) In line 433, what is the meaning of GFP14x ?
- 12) In Figure S4A lane 3, why is the PARylated RSMC is sharply banded? Normally PARylation looks smeary like Figures 5D and 5E.

13) What does "SN" mean in Figure 5F? Please explain in the legend.

Referee #2:

This manuscript uncovers a mechanism of Sororin recruitment to cohesin during DNA replication to establish cohesion that works in parallel with cohesin acetylation. The microprotein RSMC (123 aa) is identified in Sororin immunoprecipitates and its reduction is shown to cause cohesion defects that are epistatic to reduction of Sororin and rescued by overexpression of Sororin. The microprotein contains multiple parylation sites and its parylation is important for the interaction with Sororin. Addition of olaparib results in cohesion defects that can be overcome by RSMC overexpression. The authors conclude that RSMC parylation by PARP1 during DNA replication enhances the interaction of RSMC with Sororin and thereby promotes both its recruitment and its anti-WAPL activity.

The results shown are interesting and relevant, as they provide a mechanism to ensure that cohesin complexes are made cohesive during DNA replication. Overall, the in vitro and in vivo data are clear and support the conclusions. However, some additional experiments are important to strengthen the manuscript conclusions in general and to clarify the role of PARP1 in particular.

Major concerns:

-Olaparib works by preventing parylation but also trapping PARP on chromatin (e.g., Kanev 2024) and so it is difficult to dissect whether its effect is a consequence of steric hindrance to replication forks or on parylation of RSMC. A previous publication that is commented in Discussion, Kukolj 2017 showed that cohesion defects caused by Olaparib were the consequence of PARP entrapment, as defects were rescued by knockdown of PARP1/2 (in the presence of the drug). The authors should also perform PARP1/2 KD in the presence and absence of Olaparib to support their claim that the major effect of Olaparib is to prevent proper interaction of RSMC and Sororin and to further support the idea that PARP1 is important to promote (through RSMC) Sororin recruitment during DNA replication.

- On the same line, what is the effect of adding Flag-RSMC_16A in the presence of Olaparib?

- The authors propose a double role of RSMC in (1) recruitment of Sororin and (2) to increase its anti-WAPL activity. It would be important to generate an RSMC degron and test if RSMC is required for maintenance of cohesion, as shown previously for Sororin, by checking cohesion defects when RSMC is degraded in G2 AFTER cohesion establishment (Ladurner 2016). This would support the idea that RSMC interaction with Sororin is maintained after initial recruitment at the fork via PARP. Alternatively, the IP shown in Figure 3A in early versus mid S phase, could be also shown for G2.

-The degron (if it works efficiently) would also allow to have a better idea of the importance of RSMC since the effects of siRNA and heterozygous KO are relatively mild.

The same problem affects the experiments with Esco1/2 depletion, as Esco2 KD by siRNA is not very efficient. When better depletions are achieved, recruitment of Sororin is much diminished and cohesion defects are much increased (see Minamino 2015 or Nishiyama 2010). The question is whether RSMC binding is absolutely required for cohesion establishment, as is cohesin acetylation.

-In Figure 6, the figure legend says "Replication-coupled PARP1 and RSMC cooperate with ESCO1/2 to recruit Sororin" but there is nothing in the figure related with ESCO1/2. To show that these two mechanisms contribute "separately" to ensure Sororin recruitment concomitant with DNA replication, it would be important to compare the additive effect of Olaparib with Esco1/2 KD. Also, can overexpression of Sororin rescue these defects?

-I am unconvinced of the reliability of the Sororin stainings shown in Figure 4E or 6C. First, I cannot see much in the images provided. Second, I do not understand the methodology: "carnoy-fixed chromosome spreads from which soluble proteins had been removed by hypotonic pretreatment (Nagasaka et al., 2016; Nishiyama et al., 2010)". I do not think that these references are correct. Hypotonic treatment followed by Carnoy fixation and spreading on coverlips is used for mitotic chromosome spreads like those shown in Figure 1E. In order to remove soluble proteins, pre-extraction should be performed with detergent before fixation.

-Legend of Figure 7(C,D) "RSMC deficiency leads to reduced chromatin associated Wapl" is the opposite of what it is said in main text. Also, the western blot is not convincing, please show one of the other two replicates. More importantly, this result contradicts a previous one by Nishiyama 2010 showing both in HeLa cells and Xenopus egg extracts that depletion of Sororin does affect the amount of WAPL on chromatin.

Additional issues:

-The ChIP experiment is very limited. It would be nice to use more than one site and also to compare different times in the cell cycle: G1 (before DT release), S phase (5h) and G2 (10 h but combined with RO 3306 to prevent entry in mitosis). This G2 condition would serve also for the colP Sororin-RSMC proposed above.

-Figure legend of 7C,D "RSMC deficiency leads to reduced chromatin associated Wapl is a mistake. Also, the western blot is not convincing, please show one of the other two replicates. More importantly, this result is at odds with previous results by Nishiyama 2010 showing both in HeLa cells and Xenopus egg extracts that depletion of Sororin does not affect the amount of WAPL on chromatin.

-Line 68 "SMC3 is acetylated mainly by ESCO2 (Chan et al, 2012; Rolef Ben-Shahar et al, 2008; Rowland et al, 2009; Unal et al, 2008; Zhang et al, 2008), which is recruited by several replication fork components"

All the references indicated above refer to yeast Eco1. In mammalian cells, both Esco1 and Esco2 can acetylate cohesin: even though Esco2 appears to have a more prominent role in cohesin acetylation in S phase upon replication fork passage, both Esco1 and Esco2 must be depleted in order to see clear cohesion defects (Alomer 2017 PNAS, or Minamino 2015 Curr Biol)

Referee #3:

This study investigates a previously underappreciated microprotein, RSMC (encoded by the alternative ORF C11ORF98), and its role in establishing sister chromatid cohesion during S phase. The authors demonstrate that RSMC directly interacts with Sororin, a key regulator required for the establishment and maintenance of sister chromatid cohesion. Importantly, they show that PARP1-mediated PARylation of RSMC during S phase reinforces this interaction. Using a combination of methods including immunofluorescence (IF), immunoprecipitation (IP), proximity ligation assay (PLA), and chromatin fractionation, the authors show that the PARP-RSMC axis facilitates Sororin recruitment in cooperation with DNA replication-coupled, ESCO1/2-catalyzed SMC3 acetylation. Overall, the study is well designed, and the conclusions are supported by multiple lines of evidence. The findings and the proposed model are of broad interest to the field of biology. I have the following comments that need to be addressed:

Major Points:

- (1) Figure 1: In panel B, the percent of input should be shown to demonstrate the relative co-immunoprecipitation between Flag-RSMC and endogenous Sororin; In panel C, the signal for endogenous RSMC-GFP and its colocalization with Sororin appears weak. The authors do not specify the cell cycle stage examined. This weak signal may be due to low protein levels in G1-phase cells.
- (2) Figure 2: The authors generated two interaction-defective mutants, RSMC Δ SIM and RSMC-16A. It would be useful to know whether these mutants behave similarly in other assays, such as the MST shown in Figure 7E. In addition, the percent of input should be shown in panels A and B.
- (3) Figure 3: RSMC appears to affect both the total and chromatin-bound levels of Sororin. Do the authors observe changes at the mRNA and/or protein level of Sororin?
- (4) In most experiments, only the interaction between RSMC and Sororin is shown. It is recommended that the authors also assess the interaction between RSMC and other core Cohesin subunits.
- (5) The efficiency of RSMC knockdown or the RSMC \pm cell lines is evaluated solely by qPCR at the mRNA level. It would be preferable to assess the endogenous protein levels as well.
- (6) Experimental details are often insufficient; for example, the authors should clarify how gel loading was normalized, how quantification was performed relative to an internal standard, and the number of biological replicates used for each assay. Providing these details would help clarify the results and strengthen the conclusions.
- (7) Figure 4: In panel C, is it possible that the low intensity of the Sororin band in line 8 is due to the low level of Sororin in lane 4?
- (8) Figure 5: Would it be possible to show PARylation of endogenous RSMC in cells?
- (9) Figure 6: Emetine, which is also a protein synthesis inhibitor, is used in the experiments, complicating the interpretation of the data. The use of a more specific inhibitor would be preferable.
- (10) PARP is involved in multiple cellular processes, with a well-documented role in single-strand DNA break repair. Did the authors examine whether PARP also recruits RSMC and Sororin at other types of DNA nicks, such as those found in damaged DNA?
- (11) The authors propose that PARP1 binds to Okazaki fragments on the lagging strand. However, factors involved in leading strand synthesis, such as PCNA and Ctf18, are also implicated in cohesion establishment. The manuscript would benefit from a discussion of the roles of both leading and lagging strand synthesis in this process.
- (12) Figure 7: It would be better to discuss how could RSMC promote Sororin activity in antagonizing the Pds5B-Wapl interaction.

Minor Points:

- (1) Some items are not presented consistently throughout the text (e.g., capitalization of subtitles, bar formats, and the nomenclature used for RSMC $\pm\pm$ versus RSMC \pm).
- (2) The level of detail in the figure legends is inconsistent. For instance, Figure 5F includes many details that are not provided for other figures. Consider moving some of this information to the Materials and Methods section.

Dear Editors,

Thanks to your careful consideration and all highly constructive comments from the reviewers, which helped us to design more than 20 new experiments and thus improve the entire manuscript significantly. Here is the summary of the main changes followed by a point-to-point response.

General Response to the key concerns:

1. More direct evidence that RSMC PARylation facilitates Sororin recruitment in cells.

In addition to RSMC-16A (all PARylation sites mutated to alanine), we have generated the RSMC-binding deficient mutant Sororin-12A (PAR-binding motifs mutated to alanine). So, we can provide new direct evidence from PARylation writer, substrate and reader:

- 1) PARylated RSMC barely binds Sororin-12A *in vitro* (Figure EV4G, lane 7).
- 2) Co-immunoprecipitation (CoIP) analyses revealed that RSMC binds Sororin-12A significantly weaker than Sororin WT (Figure EV4F, lane 5).
- 3) Functionally, Sororin-12A overexpression failed to rescue cohesion defects in Sororin KD (Figure 5H), RSMC^{+/-} (Figure EV4H), or PARPi-treated cells (Figure EV5C). However, Sororin-12A overexpression can rescue cohesion defects caused by ESCO1&2 RNAi (Figure EV5C).
- 4) CoIP results showed that RSMC WT associates Sororin more strongly than the non-PARylatable mutant RSMC-16A (Figure EV2E).
- 5) RSMC-16A overexpression failed to rescue cohesion defects caused by PARPi (Figure 6F).
- 6) In silico structural prediction implicates a dual-mode interaction between RSMC-Sororin. First, RSMC (K53, K106, and K107 residues) mediates a basal interaction with Sororin (V194, A239, and A243) via intermolecular hydrogen bonds (Figure EV2B). This basal RSMC-Sororin interaction allows detection by *in vitro* pulldown using excessive purified recombinant proteins. PARylation of the RSMC's KR-rich sites (R30, R31, R33, R35, K52, K53, S65, E87, E89, S98, K103, R104, K106, K107, D113, E115) enhances the negative charge of RSMC, thereby strengthening its association with the K/R-rich PAR binding motifs (K24, R27, R28, R31, K56, L60, K61, R62, K214, K218, K219, K22) of Sororin. The PAR-enhanced interaction can be recognized mainly through *in vivo* or *in situ* assays.

After many attempts with various conditions, we were unable to detect endogenous PARylated RSMC. PARP activity triggered by the unligated Okazaki fragments is very transient (Hanzlikova et al., 2018, Mol Cell). The endogenous PARylation levels of RSMC may be too low to be detected using commercially

available anti-PAR antibodies.

2. Possible confounding effects of chromatin-trapped PARP enzymes.

- 1) In our hand, PARP1 (the primary PARP enzyme) KD caused no apparent cohesion defects in chromosome spread analyses (Figure EV5A), which is consistent with Kukolj 2017.
- 2) We reasoned that this might be due to insufficient RNAi efficiency, PARP1/2 redundancy and/or low PARP activity requirement for RSMC. Therefore, we further knocked down PARP2 in the primary enzyme PARP1 KO (PARP1^{-/-}) background. Then, we observed cohesion defects similar to those induced by PARPi and RSMC depletion (Figure EV5B).
- 3) Combination of PARP1^{-/-} and PARP2 KD failed to rescue the cohesion defects caused by olaparib (Figure EV5B), arguing against a major contribution of chromatin-trapping of PARP enzymes.

These data are consistent with a series of in vitro, in vivo and in situ biochemical/cell biological results and genetic results described in Point 1. We'd like to emphasize two critical pieces of functional evidence:

- 4) Olaparib-induced cohesion defect could be effectively rescued by overexpression of either Sororin or its partner RSMC. However, the PAR-binding-deficient mutant Sororin-12A (Figure EV5C) and non-PARYlatable mutant RSMC-16A failed to do so (Figure 6E, F).
- 5) Meanwhile, Sororin-12A overexpression can rescue cohesion defects caused by ESCO1/2 KD (Figure EV5C).

3. RSMC is not affecting Sororin via translation.

- 1) The total and chromatin-bound protein levels of Sororin decrease by about average 20% and 50% in RSMC^{+/-} (Figure 4C, D and Figure EV3G; Figure 3D-F and Figure EV3D), suggesting RSMC regulates both protein and chromatin recruitment levels of Sororin.
- 2) The total protein levels of Sororin in RSMC^{+/-} could be restored by treatment of proteasome inhibitor MG132 (Figure EV3E). Meanwhile, the mRNA levels of Sororin were not significantly changed after RSMC depletion (Figure EV3F). These results suggest that RSMC might stabilize Sororin presumably through interaction.
- 3) As already described in General Response 1, we have added orthogonal evidence from a separation-of-function mutant from the Sororin side, Sororin-12A, at a wildtype protein level, is defective in both RSMC binding (Figure EV4E-G) and cohesion establishment (Figure 5H).

4. Acute RSMC depletion to better define its specific roles.

- 1) Our attempts to generate homozygous RSMC^{-/-} knockout cells across multiple cell lines (HEK293T, HCT116 and HepG2) were unsuccessful. We obtained about 15% RSMC^{+/-} mutants from hundreds of clones screened, but none of them are homozygous.
- 2) Alternatively, we conducted siRNA against RSMC in RSMC^{+/-} cells as suggested. This led to additive cohesion defects (~67%, Figure EV1J), close to depletion of the essential factors as cohesin subunits or Sororin (~80%, Figure 1F). Such severe cohesion defects may not allow cell survival.
- 3) To overcome the lack of RSMC antibodies, we labeled RSMC with a 3Flag tag at its genomic loci. This allowed us to detect the RNAi efficiency of RSMC to be about 60% (Figure EV1G).
- 4) We also attempted to develop conditional RSMC degradation systems (using both PROTAC and AID). However, neither system has worked yet.

Referee #1:

Sororin, a cohesin-binding protein, is essential for sister chromatid cohesion establishment during DNA replication. Sororin is recruited to the cohesin complex via Pds5 and Esco1/2-dependent acetylation of Smc3. Although it has been established that acetylation of Smc3 is required for Sororin recruitment to cohesin (i.e. to chromatin), it has also been suggested that Smc3 acetylation is not sufficient and that an unknown mechanism dependent on DNA replication is required for full activation of Sororin recruitment.

In this study, Jiang and colleagues identified a novel Sororin-interacting protein, RSMC, in human HEK293T cells by IP-MS. In HEK293 cells, overexpressed RSMC or overexpressed Sororin can associate with endogenous Sororin or RSMC, respectively. PLA assay shows that RSMC and Sororin or cohesin are in close proximity to each other in human cells. Knockdown (RNAi) or heterozygously expression of RSMC (RSMC^{+/-}) exhibits mild cohesion defects. The PARylation-deficient RSMC 16A mutant cannot restore cohesion defect in RSMC RNAi cells. Finally, RSMC WT but not 16A facilitates Wapl-Pds5B dissociation in vitro. From these lines of evidence, the authors conclude that PARP1-dependent PARylation of RSMC strengthens its association with Sororin, thereby enhancing Sororin's chromatin recruitment by antagonizing Wapl's cohesin dissociation activity. Although the authors' findings that RSMC interacts with Sororin and that RSMC has an impact on sister chromatid cohesion in human cells are novel and interesting, the direct evidence showing that PARylation of RSMC facilitates Sororin recruitment to chromatin is insufficient, especially in vivo. Therefore, the relationship between PARylation and Sororin-dependent sister chromatid cohesion in this study is still unclear.

Another major concern in this manuscript is that all the presenting data could not rule out the possibility that RSMC facilitate Sororin translation because (a) protein level of Sororin is decreased in RSMC^{+/-} cells (Figures S3D and S3E), (b) Sororin overexpression rescues the cohesion defect (Figure 2D), (c) mRNA level of Sororin does not change (Figure EV3F), and (d) RSMC have recently been reported to bind 28S rRNA and play a critical role in ribosome biogenesis (Zhang et al., 2023). To exclude this possibility, it is necessary to test whether the Sororin mutant, which is unable to bind to RSMC, restores the cohesion defect in RSMC^{+/-} cells (see below @ Major point (2) and (4)). Unless the authors clarify these points, it would be impossible to conclude that PARylation of RSMC facilitates Sororin recruitment to chromatin. For these reasons, I would recommend revising the manuscript.

Major points:

- 1) The authors showed that partial depletion of RSMC caused partial cohesion defect in human somatic cells and that RSMC-WT, but not the PARylation-deficient RSMC-16A mutant, could rescue the defect. However, as mentioned above, one of the major concerns is that direct evidence that PARylation of RSMC is required for Sororin binding to RSMC and/or chromatin is insufficient. To more directly test the importance of RSMC PARylation, the authors should perform an IP experiment

(rather than PLA) in HEK293T cells if RSMC WT but not 16A associates with Sororin.

A: Refer to general response 1.

2) Related to the previous point, if binding between Sororin and PARylated RSMC is important for cohesion, then Sororin mutant that does not bind to RSMC should show the similar cohesion defect. Does Sororin WT/12A associate with PARylated RSMC both in vitro and in vivo? Furthermore, when endogenous Sororin is substituted with Sororin 12A, does Sororin 12A restore the cohesion defect? More importantly, is cohesion defect shown under PARP inhibition or in RSMC^{+/-} cells restored by expression of either Sororin WT or 12A? These points are also important to rule out the possibility that RSMC facilitate Sororin translation, that is another big concern in this manuscript.

A: Refer to general response 1 and 3.

3) Although it's shown that non-PARylatable RSMC 16A is less bound to Sororin in vitro (Figure 2B), where even RSMC-WT is not PARylated, there is no direct evidence that PARylated RSMC are better associated with Sororin compared to non-PARylated RSMC in vitro or in vivo. The authors should test this.

A: Refer to general response point 1.

4) It is unclear which part of Sororin is responsible for the association with RSMC from AlphaFold prediction (Figure EV2B). Although the authors identified 12 K/R residues in Sororin that are required for PAR binding in vitro, are these 12 residues also important for association with RSMC even without PARylation?

A: Refer to general response point 1.

5) In both RSMC RNAi and RSMC^{+/-} cells, the cohesion defect is relatively mild compared to Sororin KD. This could be due to knockdown efficiency. Can the authors generate RSMC ^{-/-} cells or perform RSMC RNAi in RSMC ^{+/-} cells to completely deplete RSMC from the cells? This might result in more severe cohesion defects.

A: Refer to general response point 4.

6) In Figures 2A and 2B, RSMC in its unPARylated state binds to Sororin in vitro, whereas RSMC 16A or ΔSIM cannot. This suggests that PARylation is not required for RSMC association with Sororin in vitro. The authors should explain how these 16 K/R residues on RSMC contribute to Sororin binding and how PARylation of these residues facilitates the association with Sororin.

A: Refer to general response point 1.

7) In Figure 7F, RSMC facilitates Sororin activity to dissociate Wapl from Pds5B. Does PARylated RSMC further promote the Sororin activity of Wapl dissociation?

A: We appreciate the reviewer's insightful question. Due to very low PARylation efficiency in vitro, we failed to obtain sufficient PARylated RSMC samples for this experiment.

8) Related to the previous point, does PARylated RSMC facilitate Pds5-Sororin binding? If so, what about the binding between Pds5 and Sororin-12A?

A: Very good point. Like Sororin WT, Sororin-12A alone still retained basal anti-Wapl activity (Figure EV5D-E). This implies that PAR-binding of Sororin is unlikely involved in its interaction with Pds5.

9) It seems that RSMC levels increase during S phase. Is this DNA replication dependent? In other words, is RSMC accumulation suppressed by replication inhibition? In addition, if PARP1 activity is also replication-dependent, then PARylated RSMC should also be decreased after replication inhibition. Is this the case?

A: Good point. We treated cells with hydroxyurea (HU)—which inhibits DNA replication by depleting nucleotide pools without blocking S-phase initiation—still leads to an increase in the RSMC protein level, suggesting that its upregulation is triggered by S-phase entry itself rather than DNA synthesis (Letter Figure 1). In line with this, we found that there are several E2F1-binding motifs in the RSMC promoter (Table 1). E2F1 is a master regulator of the G1/S transition, controlling genes required for DNA replication (e.g., cyclins, DNA polymerases) and histones. The presence of E2F1 sites implies that RSMC transcription can be cell-cycle-regulated. According to S-phase PARP activity, replication inhibition by HU, aphidicolin or other drugs may induce more ssDNA or breaks which complicates the issue.

Letter Figure 1. RSMC accumulation during S phase is independent of DNA replication. HEK293T cells expressing 3Flag-tagged RSMC were synchronized at the G1/S boundary using a double-thymidine block. Cells were released into S phase and treated with DMSO (control) or hydroxyurea (HU, 2 mM) to inhibit DNA replication. Samples were harvested at indicated time points post-release and subjected to immunoblotting with antibodies against Flag (RSMC), and the tubulin

control.

Table 1. E2F1-binding motifs in RSMC promoter region.

Display profiles Filter:

Matrix ID	Name	Score	Relative score	Sequence ID	Start	End	Strand	Predicted sequence
MA0024.2	MA0024.2.E2F1	8.926818	0.9023232	98	1908	1918	-	GGTGCGGGAGG
MA0024.2	MA0024.2.E2F1	7.733048	0.88436323	98	1279	1289	+	GGGGCGGCACG
MA0024.3	MA0024.3.E2F1	11.587598	0.8825689	98	1004	1015	+	ATAGGCGCCATC
MA0024.3	MA0024.3.E2F1	11.476967	0.8807269	98	1004	1015	-	GATGGCGCCTAT
MA0024.2	MA0024.2.E2F1	6.747393	0.8695343	98	1533	1543	+	GCAGCGGAAG
MA0024.2	MA0024.2.E2F1	6.602072	0.86734796	98	1094	1104	-	GAGGCGGCAG
MA0024.2	MA0024.2.E2F1	6.3519964	0.8635857	98	1844	1854	-	TGGCGGCCCGG
MA0024.2	MA0024.2.E2F1	6.153494	0.8605992	98	623	633	+	TAGGCTGGAAG
MA0024.2	MA0024.2.E2F1	5.4829955	0.8505118	98	1302	1312	+	GGCTCGGCAAG

Showing 1 to 9 of 9 entries Previous Next

Minor points:

10) In PLA experiments (Figures 1C, 5B, and 5H), the images are too dark to recognize the red colors. Please improve the figures.

A: Thanks to the reviewer, the original raw figures were too dark. We improved the figures by increasing the contrast of the whole figure panel in the updated manuscript (Figure 1C, 5B, and 5I).

11) In line 433, what is the meaning of GFP14x ?

A: We clarified as: The term "GFP14x" refers to 14 tandem repeats of GFP C-terminal peptide (GFP11), which can bind with N-terminal fragment (GFP1–10) to enhance fluorescence detection sensitivity for low-abundance targets (Cabantous et al., 2005 Nat Biotechnol). The split GFP system used in our assays was gifted by Prof. Baohui Chen's group (Xu et al., 2020).

12) In Figure EV4A lane 3, why is the PARylated RSMC is sharply banded? Normally PARylation looks smeary like Figures 5D and 5E.

A: Yes, the PARylation signals mostly looks smeary, but in Figure EV4A look sharply. In Figure EV4A, Flag-PARP1 was precipitated from cell lysates, which may have relatively higher activity that enables a higher signal/noise ratio. In other figures, PARP1-His6 was a commercially available recombinant protein purified from baculovirus-transfected insect cells (SinoBiological, 11040-H08B).

13) What does "SN" mean in Figure 5F? Please explain in the legend.

A: SN means supernatants. We added it in the legend.

Referee #2:

This manuscript uncovers a mechanism of Sororin recruitment to cohesin during DNA replication to establish cohesion that works in parallel with cohesin acetylation. The microprotein RSMC (123 aa) is identified in Sororin immunoprecipitates and its reduction is shown to cause cohesion defects that are epistatic to reduction of Sororin and rescued by overexpression of Sororin. The microprotein contains multiple parylation sites and its parylation is important for the interaction with Sororin. Addition of olaparib results in cohesion defects that can be overcome by RSMC overexpression. The authors conclude that RSMC parylation by PARP1 during DNA replication enhances the interaction of RSMC with Sororin and thereby promotes both its recruitment and its anti-WAPL activity.

The results shown are interesting and relevant, as they provide a mechanism to ensure that cohesin complexes are made cohesive during DNA replication. Overall, the in vitro and in vivo data are clear and support the conclusions. However, some additional experiments are important to strengthen the manuscript conclusions in general and to clarify the role of PARP1 in particular.

Major concerns:

1) -Olaparib works by preventing parylation but also trapping PARP on chromatin (e.g., Kanev 2024) and so it is difficult to dissect whether its effect is a consequence of steric hindrance to replication forks or on parylation of RSMC. A previous publication that is commented in Discussion, Kukulj 2017 showed that cohesion defects caused by Olaparib were the consequence of PARP entrapment, as defects were rescued by knockdown of PARP1/2 (in the presence of the drug). The authors should also perform PARP1/2 KD in the presence and absence of Olaparib to support their claim that the major effect of Olaparib is to prevent proper interaction of RSMC and Sororin and to further support the idea that PARP1 is important to promote (through RSMC) Sororin recruitment during DNA replication.

A: Refer to general response point 2.

2) - On the same line, what is the effect of adding Flag-RSMC_16A in the presence of Olaparib?

A: Refer to general response point 1, RSMC-16A overexpression failed to rescue cohesion defects in cells treated by olaparib (Figure 6F)

3) The authors propose a double role of RSMC in (1) recruitment of Sororin and (2) to increase its anti-WAPL activity. It would be important to generate an RSMC degron and test if RSMC is required for maintenance of cohesion, as shown previously for Sororin, by checking cohesion defects when RSMC is degraded in G2 AFTER cohesion establishment (Ladurner 2016). This would support the idea that RSMC

interaction with Sororin is maintained after initial recruitment at the fork via PARP. Alternatively, the IP shown in Figure 3A in early versus mid S phase, could be also shown for G2.

A: We thank the reviewer for highlighting this important aspect of cohesion maintenance.

We performed chromatin immunoprecipitation (ChIP) assays as suggested. Both RSMC and Sororin remain localized at cohesin binding sites during G2 phase (Figure 3C and EV3C), suggesting RSMC-Sororin interaction can be maintained in G2.

We agree with the reviewer that an RSMC degron would be helpful to distinguish RSMC's role in cohesion establishment and maintenance. Currently, we are developing conditional RSMC degradation systems (via both PROTAC and AID). However, neither system has worked yet. Should these efforts succeed, we can directly test its possible role in cohesion maintenance in G2 in future.

4) -The degron (if it works efficiently) would also allow to have a better idea of the importance of RSMC since the effects of siRNA and heterozygous KO are relatively mild. The same problem affects the experiments with Esc1/2 depletion, as Esc2 KD by siRNA is not very efficient. When better depletions are achieved, recruitment of Sororin is much diminished and cohesion defects are much increased (see Minamino 2015 or Nishiyama 2010). The question is whether RSMC binding is absolutely required for cohesion establishment, as is cohesin acetylation.

A: Refer to general response point 4.

5) In Figure 6, the figure legend says "Replication-coupled PARP1 and RSMC cooperate with ESCO1/2 to recruit Sororin" but there is nothing in the figure related with ESCO1/2. To show that these two mechanisms contribute "separately" to ensure Sororin recruitment concomitant with DNA replication, it would be important to compare the additive effect of Olaparib with Esc1/2 KD. Also, can overexpression of Sororin rescue these defects?

A: Yes, olaparib and Esc1/2 KD show an additive effect in terms of Sororin recruitment to chromatin (Figure 6G-H) as well as cohesion defects (Figure EV5C). In addition, Sororin overexpression rescued cohesion defects in the absence of ESCO1&2, under PARP inhibition, or both. More interestingly, the PAR/RSMC-binding deficient mutant Sororin-12A overexpression only rescued cohesion defects caused by ESCO1&2 RNAi but not by PARPi (Figure EV5C), supporting that PARPs and ESCO1&2 function in parallel to recruit Sororin.

6) -I am unconvinced of the reliability of the Sororin stainings shown in Figure 4E or 6C. First, I cannot see much in the images provided. Second, I do not understand the methodology: "carnoy-fixed chromosome spreads from which soluble proteins had been removed by hypotonic pretreatment (Nagasaka et al., 2016; Nishiyama et al.,

2010)". I do not think that these references are correct. Hypotonic treatment followed by Carnoy fixation and spreading on coverlips is used for mitotic chromosome spreads like those shown in Figure 1E. In order to remove soluble proteins, pre-extraction should be performed with detergent before fixation.

A: Thanks to the reviewer, the original figures were too dark. We improved the figures by increasing the contrast of the whole figure panel in the updated manuscript as suggested (Figure 4E, and 6C).

We apologize for the wrong citation, and rewrote the methods as follows:

“Immunofluorescence staining of chromatin-bound Sororin was modified from Nishiyama et al., 2010. In brief, S phase cells were released from a double thymidine (DT) block. Post-trypsinization, cells were hypotonically treated with 75 mM KCl for 5 min and stopped by 1xPBS quick wash, then pre-extracted with 0.1% Triton X-100 for 2 min then wash once by 1xPBS, followed by 4% PFA fixation for 10 min. Cell pellets were washed with PBS and fixed in Carnoy’s solution for 10 min before dropping on the cover slides. Afterward, IF staining of Sororin was conducted as described above.”

7) -Legend of Figure 7(C, D) "RSMC deficiency leads to reduced chromatin associated Wapl" is the opposite of what it is said in main text. Also, the western blot is not convincing, please show one of the other two replicates. More importantly, this result contradicts a previous one by Nishiyama 2010 showing both in HeLa cells and Xenopus egg extracts that depletion of Sororin does affect the amount of WAPL on chromatin.

A: Very good point, we repeated this experiment by two people back to back more than 5 times. As shown below, chromatin-associated Wapl levels varied between batches (Letter Figure 2), presumably due to slight differences in the experimental conditions. This is mainly in agreement with the previous notion that after displacement from PDS5, Wapl may still associate with chromatin in a distinct conformation/manner, which may be susceptible to wash conditions. So, we decided to remove this dataset.

Letter Figure 2. Chromatin fractionation of Wapl.

Additional issues:

8) -The ChIP experiment is very limited. It would be nice to use more than one site and also to compare different times in the cell cycle: G1 (before DT release), S phase (5h) and G2 (10 h but combined with RO 3306 to prevent entry in mitosis). This G2 condition would serve also for the coIP Sororin-RSMC proposed above.

A: ChIP analyses showed that both RSMC and Sororin bind two representative CAR sites. Critically, RSMC binding persists at Sororin-enriched CAR sites during G2 phase (Figure 3C and Figure EV3C), supporting the sustained RSMC-Sororin interaction beyond DNA replication.

9) -Figure legend of 7C, D "RSMC deficiency leads to reduced chromatin associated Wapl is a mistake. Also, the western blot is not convincing, please show one of the other two replicates. More importantly, this result is at odds with previous results by Nishiyama 2010 showing both in HeLa cells and Xenopus egg extracts that depletion of Sororin does not affect the amount of WAPL on chromatin.

A: same as Point 7.

10) -Line 68 "SMC3 is acetylated mainly by ESCO2 (Chan et al, 2012; Rolef Ben-Shahar et al, 2008; Rowland et al, 2009; Unal et al, 2008; Zhang et al, 2008), which is recruited by several replication fork components". All the references indicated above refer to yeast Eco1. In mammalian cells, both Esco1 and Esco2 can

acetylate cohesin: even though Esco2 appears to have a more prominent role in cohesin acetylation in S phase upon replication fork passage, both Esco1 and Esco2 must be depleted in order to see clear cohesion defects (Alomer 2017 PNAS, or Minamino 2015 Curr Biol)

A: Good suggestions. We rewrote it and added the citation as follows: SMC3 is acetylated by ESCO1 and ESCO2 (Chan et al, 2012; Rolef Ben-Shahar et al, 2008; Rowland et al, 2009; Unal et al, 2008; Zhang et al, 2008; Alomer et al., 2017, Minamino et al., 2015). The acetylase ESCO2 is recruited by several replication fork components (e.g. proliferating cell nuclear antigen, PCNA, (Skibbens et al, 1999); CRL4-MMS22L (Sun et al, 2019; Zhang et al, 2023a; Zhang et al, 2017) and MCM (Ivanov et al, 2018; Yoshimura et al, 2021).

Referee #3:

This study investigates a previously underappreciated microprotein, RSMC (encoded by the alternative ORF C11ORF98), and its role in establishing sister chromatid cohesion during S phase. The authors demonstrate that RSMC directly interacts with Sororin, a key regulator required for the establishment and maintenance of sister chromatid cohesion. Importantly, they show that PARP1-mediated PARylation of RSMC during S phase reinforces this interaction. Using a combination of methods including immunofluorescence (IF), immunoprecipitation (IP), proximity ligation assay (PLA), and chromatin fractionation, the authors show that the PARP-RSMC axis facilitates Sororin recruitment in cooperation with DNA replication-coupled, ESCO1/2-catalyzed SMC3 acetylation. Overall, the study is well designed, and the conclusions are supported by multiple lines of evidence. The findings and the proposed model are of broad interest to the field of biology. I have the following comments that need to be addressed:

Major Points:

(1) Figure 1: In panel B, the percent of input should be shown to demonstrate the relative co-immunoprecipitation between Flag-RSMC and endogenous Sororin; In panel C, the signal for endogenous RSMC-GFP and its colocalization with Sororin appears weak. The authors do not specify the cell cycle stage examined. This weak signal may be due to low protein levels in G1-phase cells.

A: Thank you for the reminder. Figure 1: In panel B, there are 3% input show in Figure 1 panel B input, we clarified in the updated figure legend.

Good suggestions, we did the staining in S phase cells and the signal is way improved (Figure EV1D).

(2) Figure 2: The authors generated two interaction-defective mutants, RSMC Δ SIM and RSMC-16A. It would be useful to know whether these mutants behave similarly in other assays, such as the MST shown in Figure 7E. In addition, the percent of input

should be shown in panels A and B.

A: Both RSMC Δ SIM and RSMC-16A exhibit comparable loss-of-function phenotypes, showing no detectable anti-Wapl stimulatory activity in parallel assays (MST data now in Figure 7C, D). We added the percent of input in the legend.

(3) Figure 3: RSMC appears to affect both the total and chromatin-bound levels of Sororin. Do the authors observe changes at the mRNA and/or protein level of Sororin?

A: Refer to general response point 3.

(4) In most experiments, only the interaction between RSMC and Sororin is shown. It is recommended that the authors also assess the interaction between RSMC and other core Cohesin subunits.

A: Good suggestion. We added RSMC interactions with core cohesin subunits RAD21 (Figure EV1E), and we originally showed RSMC-SMC3 interaction by PLA assay (Figure 1C, D). These confirmed interactions between RSMC and other core Cohesin subunits.

(5) The efficiency of RSMC knockdown or the RSMC^{+/-} cell lines is evaluated solely by qPCR at the mRNA level. It would be preferable to assess the endogenous protein levels as well.

A: As suggested, we conducted RSMC RNAi into RSMC-3Flag cells (3Flag inserted into C terminal of RSMC) and observed that RSMC can be knocked down about 60% by its siRNAs (Figure EV1G).

(6) Experimental details are often insufficient; for example, the authors should clarify how gel loading was normalized, how quantification was performed relative to an internal standard, and the number of biological replicates used for each assay. Providing these details would help clarify the results and strengthen the conclusions.

A: We thank the reviewer for highlighting the need for methodological clarity. All experimental details - including gel normalization controls, quantification standards, and biological replicate numbers - have been comprehensively added to the figure legends.

(7) Figure 4: In panel C, is it possible that the low intensity of the Sororin band in line 8 is due to the low level of Sororin in lane 4?

A: Refer to general response point 3.

(8) Figure 5: Would it be possible to show PARylation of endogenous RSMC in

cells?

A: Refer to general response point 1.

(9) Figure 6: Emetine, which is also a protein synthesis inhibitor, is used in the experiments, complicating the interpretation of the data. The use of a more specific inhibitor would be preferable.

A: As for general DNA replication inhibitors, many studies have well addressed that all tested DNA replication inhibitors cause cohesion defects. As far as we know, there is still no ideal specific inhibitors of Okazaki fragment synthesis without affecting DNA replication. Emetine was employed for short periods to minimize side effects other than inhibition of Okazaki fragment synthesis (Burhans et al, 1991). Therefore, we chose Emetine (1 h treatment with 2 μ M emetine) with low dose and short time which has been applied in several recent reports (Hanzlikova et al, 2018; Thakar et al, 2020; Xiao et al, 2020; Cong et al, 2021; Yamashita et al, 2022).

(10) PARP is involved in multiple cellular processes, with a well-documented role in single-strand DNA break repair. Did the authors examine whether PARP also recruits RSMC and Sororin at other types of DNA nicks, such as those found in damaged DNA?

A: Yes, we conducted micro-irradiation experiments and showed that both RSMC and Sororin can be recruited to the DSB sites, while olaparib treatment inhibits the recruitment of RSMC to DSB sites. We will investigate their mechanism as a separate story.

(11) The authors propose that PARP1 binds to Okazaki fragments on the lagging strand. However, factors involved in leading strand synthesis, such as PCNA and Ctf18, are also implicated in cohesion establishment. The manuscript would benefit from a discussion of the roles of both leading and lagging strand synthesis in this process.

A: We thank the reviewer for highlighting this important consideration. We have expanded the Discussion section as following: Together with this lagging strand-associated pathway, other key replisome components in leading or lagging strand such as PCNA and its loader RFC-Ctf18, MCM, Ctf4 cooperate for efficient Sororin recruitment through SMC3 acetylation.

(12) Figure 7: It would be better to discussion how could RSMC promotes Sororin activity in antagonizing the Pds5B-Wapl interaction.

A: thanks, we proposed that post PARylation, RSMC binds potently with Sororin, which may cooperate with Sororin's FGF motif to enhance Sororin-PDS5 binding,

thereby increasing its competition against Wapl.

Minor Points:

(1) Some items are not presented consistently throughout the text (e.g., capitalization of subtitles, bar formats, and the nomenclature used for RSMC^{+/+} versus RSMC^{+/-}).

A: Thanks, we corrected it throughout the text.

(2) The level of detail in the figure legends is inconsistent. For instance, Figure 5F includes many details that are not provided for other figures. Consider moving some of this information to the Materials and Methods section.

A: Thanks, we briefed the legend as follows: PARylated RSMC showed increased Sororin interaction. In vitro, PARylation of RSMC was conducted in the absence or presence of NAD⁺. After removing PARP1 and NAD⁺, Sororin was incubated with immobilized RSMC. Input (20%) and unbound supernatants (20%) were subjected to CBB staining to check Sororin. The beads-bound proteins were checked by immunoblotting (3% to detect GST-bound RSMC, and 30% each to check RSMC PARylation and its associated Sororin).SN means supernatants.

Dr. Wenya Hou
Shenzhen University
Shenzhen University School of Medicine
Shenzhen 518060
China

30th Sep 2025

Re: EMBOJ-2025-120713R
S phase PARylation of microprotein RSMC enhances Sororin's function in sister chromatid cohesion

Dear Dr. Hou,

Thank you for submitting your revised manuscript to The EMBO Journal, and my sincere apologies for its delayed re-evaluation. The three original reviewers have in the meantime looked once more into the study, and all acknowledge the improvements to the study during the revisions. However, there are still several remaining concerns, with particularly referee 2 raising issues with missing controls/validations for the cell cycle phase experiments, and with the missing interaction assay of RSMC with endogenous core cohesin subunit(s). Given the potentially far-reaching implications of the conclusions and proposed model, I feel that adding such data (on co-IP of endogenously tagged RSMC with endogenous RAD21 or another subunit of the core cohesin complex) would still be important to strengthen the experimental evidence.

As I mentioned, we normally allow only a single round of major revision - to avoid repeated rounds of partial improvements and to give authors a decisive commitment from our side. However, given that you have already made significant efforts and addressed a majority of issues during the first revision, I would in this case allow an exceptional second round of experimental revision, to allow you to address the remaining open points. With such additional strengthening of the study, we would be happy to eventually accept it for EMBO Journal publication.

When preparing a re-revised manuscript, please also note the following figure issues that came up during our routine image analyses: some panels in Figures EV4F and EV4G are over-contrasted and without any visible background signals, making it difficult to assess the proper representation of these experiments; unfortunately, the issue is also present in the respective source data images. This needs to be clarified by providing unprocessed raw data and possibly re-assembling these figure panels from them.

I am therefore returning the manuscript to you for a second, final round of experimental revision, with the link below for eventual resubmission. Should you have any questions regarding the referee comments or this decision, please do not hesitate to contact me directly.

Yours sincerely,

Hartmut Vodermaier

- size of the scale bars that are mandatory for all micrograph panels
- the statistical test used to generate error bars and P-values
- the type error bars (e.g., S.E.M., S.D.)
- the number (n) and nature (biological or technical replicate) of independent experiments underlying each data point

- Figures may not include error bars for experiments with $n < 3$; scatter plots showing individual data points should be used instead.

9) To facilitate reproducibility and cross-laboratory adoption of methodologies, please structure the Materials & Methods section as outlined in our guide to authors, including a completed Reagents and Tools Table that can be downloaded from our author guidelines as well (<https://www.embopress.org/page/journal/14602075/authorguide#structuredmethods>).

10) Digital image enhancement is acceptable practice, as long as it accurately represents the original data and conforms to community standards. If a figure has been subjected to significant electronic manipulation, this must be clearly noted in the figure legend and/or the 'Materials and Methods' section. The editors reserve the right to request original versions of figures and the original images that were used to assemble the figure. Finally, we generally encourage uploading of numerical as well as gel/blot image source data; for details see: embopress.org/page/journal/14602075/authorguide#sourcedata

In the interest of ensuring the conceptual advance provided by the work, we recommend submitting a revision within 3 months (29th Dec 2025). Please discuss the revision progress ahead of this time with the editor if you require more time to complete the revisions. Use the link below to submit your revision:

Link Not Available

Referee #1:

I have read the revised manuscript and point-by-point response. My major concern regarding the direct molecular evidence for the relationship between Sororin and PARylated RSMC has been clearly addressed by the new Figures EV4F and EV4G. The authors also addressed another concern about the translation issue and demonstrated that Sororin proteins are indeed stabilized in an RSMC-dependent manner (Fig. EV3E), presumably through their binding. Furthermore, the authors showed that Sororin-12A failed to support cohesion in RSMC \pm cells or in Olaparib-treated cells, but not in Esc α 1/2 KD cells (new Figs. EV4H and EV5C). These results strongly suggest that the 12A mutation itself, rather than decreased protein levels, is the cause of the cohesion defects. These major concerns have been sufficiently addressed.

Nevertheless, I have the following two minor concerns.

1) In Figure EV2B, please consider using a color other than yellow, as it is difficult to recognize.

2) In the new Figure EV4G, I notice unnatural black solid lines in lanes 7 and 8 (PARylated PARP1 panel), probably due to overexposure. To avoid the appearance of these solid lines, I would suggest showing a shorter exposure in parallel. Other than these points, I would recommend this study for publication.

Referee #2:

Overall, the manuscript is improved after revision and the added data support the conclusions of the authors regarding a new mechanism to regulate Sororin binding to cohesin and thus its contribution to sister chromatid cohesion.

My criticisms regarding the role of PARP have been addressed satisfactorily. Additional details on methods are lacking, e.g., combination of siRNA for PARP, Olaparib treatment (for 48h) and enrichment of mitotic cells to prepare the spreads; western blots showing PARP1 and PARP2 after KO or KD.

Unfortunately, it has not been possible to obtain an RSMC acute degradation system. The authors have repeated ChIP of Sororin and RSMC at two genomic loci at different cell cycle phases to show that the two proteins colocalize not only during cohesion establishment in S phase, but also later, in G2, supporting a role of their interaction for cohesion maintenance. They have forgotten to add (1) synchronization method (2) flow cytometry data to verify that cells are in the indicated cell cycle phase. Referee 3 asked for co-IP of RSMC with cohesin subunits, not only Sororin. In the added data in Figure EV1E, both cohesin subunit Rad21 and RSMC are not endogenous. Why? Cells expressing Flag-RSMC from the endogenous locus (i.e., not overexpressed) are available and RAD21 is an abundant protein for which reliable antibodies are available. Reciprocal co-IP with endogenous proteins would strongly support the claims and would complement the in vitro interaction data and the experiments with PLA or overexpressed proteins.

Referee #3:

This study investigates the role of the microprotein RSMC in regulating sister chromatid cohesion through its PARylation-dependent interaction with Sororin during S phase. The authors present a compelling model wherein PARP1-mediated PARylation of RSMC enhances its association with Sororin, promoting Sororin's recruitment to chromatin and its anti-Wapl activity. The findings are novel and mechanistically insightful, offering a parallel pathway to ESCO1/2-mediated acetylation for cohesion establishment. The revised article has been strengthened by additional clarifications and data that address my prior concerns. The authors reasonably explain the technical challenges in detecting endogenous PARylated RSMC due to its transient nature and low abundance. However, the indirect evidence provided (e.g., functional rescue by WT but not PARylation-deficient mutants) is compelling. The study is of high significance to the fields of chromosome biology, DNA replication, and post-translational regulation. I therefore support its publication in EMBO J.

Dear Editors,

Thanks to your careful consideration and all highly constructive comments from the reviewers.

We changed all points as suggested: (1) added cell cycle phase validations, (2) added the interaction assay of RSMC with endogenous core cohesin subunit(s) Rad21 and SMC3, (3) changed to a low exposure figure in EV4G, (4) provided unprocessed raw data of all western blots, and (5) added the details on methods. All changes in the text were labeled in dark blue.

Details as follows:

Referee #1:

I have read the revised manuscript and point-by-point response. My major concern regarding the direct molecular evidence for the relationship between Sororin and PARylated RSMC has been clearly addressed by the new Figures EV4F and EV4G. The authors also addressed another concern about the translation issue and demonstrated that Sororin proteins are indeed stabilized in an RSMC-dependent manner (Fig. EV3E), presumably through their binding. Furthermore, the authors showed that Sororin-12A failed to support cohesion in RSMC^{+/-} cells or in Olaparib-treated cells, but not in Esc^{o1/2} KD cells (new Figs. EV4H and EV5C). These results strongly suggest that the 12A mutation itself, rather than decreased protein levels, is the cause of the cohesion defects. These major concerns have been sufficiently addressed.

Nevertheless, I have the following two minor concerns.

1) In Figure EV2B, please consider using a color other than yellow, as it is difficult to recognize.

Re: We changed Yellow into Green color.

2) In the new Figure EV4G, I notice unnatural black solid lines in lanes 7 and 8 (PARylated PARP1 panel), probably due to overexposure. To avoid the appearance of these solid lines, I would suggest **showing a shorter exposure in parallel.**

Other than these points, I would recommend this study for publication.

Re: We exchanged it for a shorter exposure image.

Referee #2:

Overall, the manuscript is improved after revision and the added data support the conclusions of the authors regarding a new mechanism to regulate Sororin binding to cohesin and thus its contribution to sister chromatid cohesion.

My criticisms regarding the role of PARP have been addressed satisfactorily. **Additional details on methods are lacking, e.g., combination of siRNA for PARP, Olaparib treatment (for 48h) and enrichment of mitotic cells to prepare the spreads; western blots showing PARP1 and PARP2 after KO or KD.**

Re: We added the additional details on methods as follows:

Line 525, “HEK293T cells were treated with nocodazole (100 ng/mL) for 4 h to enrich mitotic cells,”

Line 534-536, “HeLa PARP1 WT or KO cells were first transfected with siRNAs targeting PARP2 or control siRNAs. Six hours after transfection, olaparib was added to the culture. Mitotic cells were then harvested 48 hours following olaparib treatment.”

In addition, we added the western blots showing PARP1 and PARP2 after KO or KD in Figure EV5A and EV5B.

Unfortunately, it has not been possible to obtain an RSMC acute degradation system. The authors have repeated ChIP of Sororin and RSMC at two genomic loci at different cell cycle phases to show that the two proteins colocalize not only during cohesion establishment in S phase, but also later, in G2, supporting a role of their interaction for cohesion maintenance. They have forgotten to add **(1) synchronization method (2) flow cytometry data to verify that cells are in the indicated cell cycle phase.**

Re: We added the synchronization method in lines 671-674, “Cells were synchronized at the G1/S phase using a double-thymidine block. Samples were collected at 0 h (G1/S phase), 5 h (S phase), and 10 h (G2 phase) after release from the block. To enrich G2-phase cells, 10 μ M RO3306 was added at the 5-h time point and maintained for 5 hours before collection.”

And added the flow cytometry data as a new Figure EV3D.

Referee 3 asked for co-IP of RSMC with cohesin subunits, not only Sororin. In the added data in Figure EV1E, both cohesin subunit Rad21 and RSMC are not endogenous. Why? Cells expressing Flag-RSMC from the endogenous locus (i.e., not overexpressed) are available and RAD21 is an abundant protein for which reliable antibodies are available. Reciprocal co-IP with endogenous proteins would strongly support the claims and would complement the in vitro interaction data and the experiments with PLA or overexpressed proteins.

Re: We added co-IP with endogenous SMC3 and RSMC proteins as new data in Figure EV1E, replacing the old data. Furthermore, we also tested endogenous CoIP of RAD21 with RSMC, which is attached below.

Letter Figure, Co- IP of endogenous RAD21 and RSMC.

Referee #3:

This study investigates the role of the microprotein RSMC in regulating sister chromatid cohesion through its PARylation-dependent interaction with Sororin during S phase. The authors present a compelling model wherein PARP1-mediated PARylation of RSMC enhances its association with Sororin, promoting Sororin's recruitment to chromatin and its anti-Wapl activity. The findings are novel and mechanistically insightful, offering a parallel pathway to ESCO1/2-mediated acetylation for cohesion establishment. The revised article has been strengthened by additional clarifications and data that address my prior concerns. The authors reasonably explain the technical challenges in detecting endogenous PARylated RSMC due to its transient nature and low abundance. However, the indirect evidence provided (e.g., functional rescue by WT but not PARylation-deficient mutants) is compelling. The study is of high significance to the fields of chromosome biology, DNA replication, and post-translational regulation. I therefore support its publication in EMBO J.

Looking forward to hearing from you.

Best regards,

Wenya Hou, PhD

Shenzhen University